
# GNOM v1.0: An optimized steady-state model of the modern marine neodymium cycle

Benoît Pasquier[1], Sophia K. V. Hines[2,3], Hengdi Liang[1], Yingzhe Wu[2], Seth G. John[1], and Steven L. Goldstein[2,4]

[1]Earth Sciences Department, University of Southern California, Los Angeles, CA, USA
[2]Lamont-Doherty Earth Observatory of Columbia University, Palisades, NY, USA
[3]Department of Marine Chemistry and Geochemistry, Woods Hole Oceanographic Institution, Woods Hole, MA, USA
[4]Department of Earth and Environmental Sciences, Columbia University, Palisades, NY, USA

**Correspondence:** Benoît Pasquier (bp_358@usc.edu), Sophia K. V. Hines (shines@whoi.edu)

**Abstract.**

Spatially distant sources of neodymium (Nd) to the ocean that carry different isotopic signatures ($\varepsilon_{Nd}$) have been shown to trace out major water masses, and have thus been extensively used to study large-scale features of the ocean circulation both past and current. While the global marine Nd cycle is qualitatively well understood, a complete quantitative determination of all its components and mechanisms, such as the magnitude of its sources and the paradoxical conservative behavior of $\varepsilon_{Nd}$,

remains elusive. To make sense of the increasing collection of observational Nd and $\varepsilon_{Nd}$ data, we develop the global neodymium ocean model (GNOM) v1, the first inverse model of the global marine biogeochemical cycle of Nd. The GNOM is embedded in a data-constrained steady-state circulation that affords spectacular computational efficiency, which we leverage to estimate biogeochemical parameters via systematic objective optimization. Owing to its matrix representation, the GNOM model is

additionally amenable to novel diagnostics that allow us to investigate open questions about the Nd cycle with unprecedented accuracy. The GNOM is open-source and freely accessible, is written in Julia, and its code is easily understandable and modifiable for further developments and experiments.

## 1 Introduction

Rare earth elements (REEs) have long been recognized to provide unique insight into ocean circulation and biogeochemical
cycles (e.g., de Baar et al., 1983, 1985; Bertram and Elderfield, 1993; Elderfield, 1988; Elderfield and Greaves, 1982; German et al., 1995; Goldberg et al., 1963; Haley et al., 2014; Høgdahl et al., 1968; Lacan and Jeandel, 2001, 2004; Piepgras and Jacobsen, 1992; Piper, 1974; Sholkovitz and Schneider, 1991; Zheng et al., 2016). Isotopic variations of neodymium (Nd), in particular, have been extensively used as a tracer of ocean circulation, which plays a fundamental role in Earth's climate over a wide range of timescales, from millennia to millions of years (e.g., Adkins, 2013; van de Flierdt et al., 2016; Frank, 2002;
Goldstein and Hemming, 2003; Piepgras and Wasserburg, 1980; Sigman et al., 2010; Tachikawa et al., 2017).

Neodymium is part of a long-lived isotope system. Samarium-147 ($^{147}$Sm) decays to neodymium-143 ($^{143}$Nd) with a half-life of 106 Gyr. The Sm:Nd ratio is largely uniform in most continental rocks and lower than the bulk Earth value (DePaolo and





Wasserburg, 1976; McCulloch and Wasserburg, 1978; Goldstein et al., 1984), therefore the Nd isotope ratio $R = {}^{143}\text{Nd}/{}^{144}\text{Nd}$ is mainly a reflection of the amount of time the Nd in a rock has been a part of the continental crust, with lower values indicating

older ages and longer crustal residence times (DePaolo and Wasserburg, 1976; McCulloch and Wasserburg, 1978; Goldstein and Hemming, 2003; Jeandel et al., 2007; van de Flierdt et al., 2016; Robinson et al., 2021). Because $R$ variations are typically small, Nd isotope signatures are usually defined as

$$\varepsilon_{\text{Nd}} = R/R_{\text{CHUR}} - 1 \tag{1}$$

expressed in parts per ten thousands (‰) (DePaolo and Wasserburg, 1976), where $R$ is the measured ${}^{143}\text{Nd}/{}^{144}\text{Nd}$ ratio and

the chondritic uniform reservoir (CHUR) represents an estimate of the average Nd isotope ratio of chondritic meteorites and the bulk Earth. For this study we use $R_{\text{CHUR}} = 0.512638$ (Jacobsen and Wasserburg, 1980).

Early measurements of $\varepsilon_{\text{Nd}}$ in seawater (Piepgras and Wasserburg, 1980) and ferromanganese oxide crusts (Elderfield et al., 1981; Goldstein and O'Nions, 1981; O'Nions et al., 1978; Piepgras et al., 1979) showed systematic variation across the ocean basins, with the lowest $\varepsilon_{\text{Nd}}$ values in the North Atlantic ($-14$ to $-10$ ‰), the highest values in the Pacific ($-5$ to $0$ ‰), and

intermediate values in the Southern Ocean ($-11$ to $-8$ ‰). The latter value broadly reflects mixing between waters from the North Atlantic, which are influenced by old continental terrains in northern Canada and Greenland, and the Pacific, which is influenced by mantle-derived volcanics (van de Flierdt et al., 2016; Frank, 2002; Garcia-Solsona et al., 2014; Goldstein and Hemming, 2003; Goldstein and O'Nions, 1981; Lambelet et al., 2016; Piepgras and Wasserburg, 1980; Stichel et al., 2012a). These observations led to the recognition that $\varepsilon_{\text{Nd}}$ values could be used to trace mixing between North Atlantic and Pacific

waters over time, thus making $\varepsilon_{\text{Nd}}$ a potentially powerful paleoceanographic tracer.

More recently, the GEOTRACES program was created to better understand the sources and cycling of trace elements and isotopes in the ocean and how they impact broader marine biogeochemical cycles. The GEOTRACES Science Plan identified Nd isotopes as a "key trace element or isotope" that is expected to be measured on all GEOTRACES cruises because of its use as a paleoceanographic proxy. Thanks to this international effort, considerable amounts of new Nd concentration and isotope

data have been generated in recent years, collected notably in the GEOTRACES Intermediate Data product 2017 (IDP17, Schlitzer et al., 2018), which increased the Nd data inventory by about 50 %, as well as post-IDP17 GEOTRACES data yet to be released in future data products.

To gain the most useful and accurate information from these observations, however, it is paramount to understand their modern ocean biogeochemical cycles and tracer budgets. Neodymium and other REEs enter the ocean via rivers, submarine

groundwater discharge, aeolian deposition, pore waters, and/or interaction with sediments (Fig. 1). Once in the ocean, they are redistributed by the ocean circulation, scavenged by particulate matter, and exit the ocean via sedimentation and incorporation into authigenic ferromanganese oxides (Frank, 2002; Byrne and Kim, 1990; Elderfield, 1988; Elderfield et al., 1981; Elderfield and Sholkovitz, 1987; Sholkovitz et al., 1989, 1994, 1992; Haley et al., 2004; Blaser et al., 2016; Du et al., 2016). While most sources of REEs to the ocean have likely been identified, there are still large uncertainties associated with the magnitudes of

these different fluxes due to the inherent challenges of measuring sources that are temporally variable and globally widespread.





**Table 1.** Previous modelling studies.

| Reference | Circulation model | | Nd sources (Mmol yr⁻¹) | | | Residence time[e] (yr) | Isotope model |
|---|---|---|---|---|---|---|---|
| | | | Aeolian | Riverine | Sedimentary | | |
| Bertram and Elderfield (1993) | Box model | 7 boxes | 10 | | | 2900 | $\varepsilon_{Nd}$ only |
| Tachikawa et al. (2003) | | 10 boxes | 60 | | | 480 | $\varepsilon_{Nd}$ only |
| Du et al. (2020) | | 4 boxes | 0.82 | 3.4 | 78 | 350 | $\varepsilon_{Nd}$ only |
| Arsouze et al. (2007) | GCM | ORCA2 | No explicit Nd model | | | | $\varepsilon_{Nd}$ only[b] |
| Arsouze et al. (2009) | | ORCA2 | 0.69 | 1.8 | 55 | 560 | $^{143}$Nd, $^{144}$Nd |
| Arsouze et al. (2010) | | ORCA025[a] | No explicit Nd model | | | | $\varepsilon_{Nd}$ only[b] |
| Rempfer et al. (2011) | | Bern3D | 1.8 | 2.4 | 38 | 690 | $^{143}$Nd, $^{144}$Nd |
| Gu et al. (2019) | | POP2 | 1.5 | 9 | 28 | 750 | $^{143}$Nd, $^{144}$Nd |
| Gu et al. (2020) | | POP2 | 1.5 | 9 | 28 | 750 | $^{143}$Nd, $^{144}$Nd |
| Pöppelmeier et al. (2020) | | Bern3D | 3.5 | 12.3 | 22.8 | 750 | $^{143}$Nd, $^{144}$Nd |
| Jones et al. (2008) | Steady-state | MITgcm2.8 | No explicit Nd model | | | | $\varepsilon_{Nd}$ only[d] |
| Siddall et al. (2008) | | MITgcm2.8 | Surface boundary condition | | | | $\varepsilon_{Nd}$ only[c] |
| Du et al. (2020) | Propagator | TMI | No explicit Nd model | | | | $\varepsilon_{Nd}$ only[c] |

[a]Only the Pacific is modelled. [b]With deep boundary condition. [c]With surface boundary condition. [d]With deep and/or surface boundary conditions. [e]Bulk residence time estimated from the total Nd source magnitude for an ocean volume of $1.32 \times 10^{18}$ m³ and a mean Nd concentration of 22 pM.

Models of the marine Nd cycle, in conjunction with seawater measurements, offer a way to constrain the magnitudes and isotopic compositions of these various inputs to the ocean and identify the most important sources. Four distinct types of models have been used to simulate the modern ocean Nd cycle: simple box models, ocean general circulation models (OGCMs), steady-state circulation models, and boundary propagation models, each with their strengths and weaknesses (Table 1). Some of these models have explicitly tracked the concentrations of each Nd isotope ($^{143}$Nd and $^{144}$Nd, thus allowing for estimation of Nd concentration as well as its isotopic composition), while other models have simply tracked $\varepsilon_{Nd}$ as a single conservative tracer.

"Box models" typically refer to models consisting of 10 or fewer well-mixed boxes that exchange tracer with each other through prescribed mixing and overturning rates. Owing to their small size, box-model simulations are the fastest to run and require very little computational power. Thus, they facilitate parameter optimization and scientific exploration by allowing for quick experimentation. For example, box models have been successfully used to determine that Nd must exchange between seawater and particles in the water column or at the sediment–water interface (Bertram and Elderfield, 1993) and that riverine and aeolian sources are not sufficient to explain regional $\varepsilon_{Nd}$ variability (Tachikawa et al., 2003). However, very low spatial resolution prevents box models from capturing many important features of ocean circulation.

Ocean general circulation models sit on the other end of the spectrum of computational complexity, with better spatial resolution and resolved physics. Their computational costs generally prohibit systematic parameter space exploration or parameter optimization. These models have thus been used primarily to run well-defined experiments that target specific hypotheses, such as the importance of continental margin sources ("boundary exchange") on $\varepsilon_{Nd}$ distributions, either as the sole source of Nd to the ocean (Arsouze et al., 2007, 2010) or an additional source to rivers and aeolian deposition (Arsouze et al., 2009; Rempfer



et al., 2011; Gu et al., 2019, 2020). To our knowledge, only Rempfer et al. (2011) have attempted to optimize a Nd-cycling
model, using the low-resolution Bern3D OGCM and only two parameters at a time.

   More recently, a new class of steady-state models has emerged with unique potential to combine the advantages of OGCMs
with the computational speed of box models. These models do not resolve the physics at run time and, instead, rely on a
prescribed, steady-state circulation. They can thus directly solve for the steady-state solution of the three-dimensional tracer
equations, avoiding costly spin-ups, and drastically reducing simulation times. Thus far, to our knowledge, these models have
only been used to test the top-down hypothesis by propagating a surface boundary condition into the ocean interior. Using
the transport matrix method (TMM, Khatiwala et al., 2005; Khatiwala, 2007), Jones et al. (2008) showed that conservative
mixing and advection from the surface alone cannot reproduce interior $\varepsilon_{Nd}$ observations, while Siddall et al. (2008) showed
that including reversible scavenging captures the observed decoupling between quasi conservative $\varepsilon_{Nd}$ and nutrient-like Nd
concentration ([Nd]) distributions (the "Nd paradox"; Goldstein and Hemming, 2003).

The fourth class of models, which we have termed "boundary propagation models", entirely bypasses expressing fluxes
between model grid cells by connecting interior grid cells directly to the surface, using the total matrix intercomparison method
(TMI, Gebbie and Huybers, 2010). Specifically, boundary propagation models estimate the fractional contribution of each
surface grid cell to each interior grid cell. The TMI method does not incorporate scavenging or external fluxes and can thus
only be used to simulate conservative transport. Indeed, similar to the experiment of Jones et al. (2008) referenced above, Du
et al. (2020) used the TMI to inquire how well interior $\varepsilon_{Nd}$ values can be explained by conservative mixing and advection alone.

   Our goal is to fill the current gap in the marine Nd modelling landscape and leverage the largely unexplored benefits of
steady-state circulation models. Hence here, we present the Global Neodymium Ocean Model (GNOM) v1.0, a mechanistic
model of the modern ocean Nd cycle embedded in a state-of-the-art steady-state estimate of the modern ocean circulation from
the Ocean Circulation Inverse Model version 2 (OCIM v2.0, DeVries and Primeau, 2011; DeVries, 2014; DeVries and Holzer,
2019). The computational efficiency afforded by the model allows us to objectively optimize the model's parameters, making
GNOM v1.0 the first inverse model of the Nd cycle and producing a good match to observations.

   The GNOM v1.0 thus provides the community with a realistic yet computationally affordable tool to model the marine Nd
cycle that we hope will be used to further improve our understanding of Nd cycling in the ocean. The model code and its
optimization script are available publicly on GitHub at https://github.com/MTEL-USC/GNOM. We used the free and open-
source Julia language (Bezanson et al., 2017) and its packages, AIBECS.jl in particular (Pasquier, 2020a; Pasquier et al., under
review), as our main development platform. Owing to its open-source design, simplicity, and computational speed, the GNOM
v1.0 is ideal for Nd cycle investigations. Except for the GEOTRACES dataset which must be downloaded manually, the GNOM
is self-contained and version-controlled, making it easy to reproduce simulations.

   Additionally, the steady-state formulation of GNOM is amenable to novel Green-function-based diagnostics that can provide
important new insights into major open questions on the marine Nd cycle. Here, we introduce new partitions of Atlantic Nd and
$\varepsilon_{Nd}$ (following, e.g., Holzer et al., 2016; Pasquier and Holzer, 2017, 2018; Holzer et al., 2021) that are helpful to disentangling
the neodymium paradox (Siddall et al., 2008). We show that we can accurately partition [Nd] and $\varepsilon_{Nd}$ in the central Atlantic into
contributions from northern- and southern-sourced waters. These preliminary diagnostics already reveal important information.





They help quantify the conservativeness of $\varepsilon_{\text{Nd}}$ along water pathways and unveil the underlying mechanisms by evaluating the
effect of local sources and sinks. Detailed investigations of these diagnostics are out of the scope of this study and will be
carried out in future work. We invite paleoceanographers and modellers alike to use the GNOM v1.0 model, to improve its
implementation, explore its capabilities, and thus contribute to quantitatively answering long-lasting questions on the Nd cycle.

## 2   The GNOM Model

Neodymium concentrations are controlled by the interplay between circulation, external sources, and reversible scavenging
and burial in the sediments (Fig. 1). These components completely define the state of the Nd cycle in our Global Neodymium
Ocean Model (GNOM) v1.0. The three-dimensional partial differential equation for the Nd concentration tracer are discretized
onto the grid of the Ocean Circulation Inverse Model (OCIM v2.0; DeVries and Holzer, 2019), yielding a system of 200160
ordinary differential equations. Reorganizing the discretized three-dimensional arrays into column vectors, the steady-state
tracer equation is recast in matrix form,

$$(\mathbf{T}_{\text{circ}} + \mathbf{T}_{\text{scav}})\boldsymbol{\chi}_{\text{Nd}}^{\text{mod}} = \sum_k \boldsymbol{s}_k, \tag{2}$$

where $\boldsymbol{\chi}_{\text{Nd}}^{\text{mod}}$ is the modelled Nd concentration vector, $\mathbf{T}_{\text{circ}}$ is the OCIM v2.0 advection–diffusion operator or transport matrix,
$\mathbf{T}_{\text{scav}}$ is the reversible-scavenging matrix, and the $\boldsymbol{s}_k$ are the external sources of neodymium. Note that $\boldsymbol{\chi}_{\text{Nd}}^{\text{mod}}$ and $\boldsymbol{s}_k$ are 200160-
element column vectors and that $\mathbf{T}_{\text{circ}}$ and $\mathbf{T}_{\text{circ}}$ are sparse 200160×200160 matrices such that the linear system represented by
Eq. (2) can be solved in a few seconds on a modern laptop via LU factorization and forward and backward substitution (often
referred to as "matrix inversion").

The global $\varepsilon_{\text{Nd}}$ distribution is determined by both the distribution of $^{143}$Nd and $^{144}$Nd. Following, e.g., John et al. (2020),
instead of explicitly simulating two additional tracers, we recover $\varepsilon_{\text{Nd}}$ values by simulating a single additional fictitious tracer
for $R[\text{Nd}]$, which we denote by RNd (and its column vector by $\boldsymbol{\chi}_{\text{RNd}}^{\text{mod}}$). This is equivalent to assuming that $^{144}$Nd:Nd is constant
such that $\boldsymbol{\chi}_{\text{Nd}}^{\text{mod}}$ and $\boldsymbol{\chi}_{\text{RNd}}^{\text{mod}}$ nominally track $\left[^{144}\text{Nd}\right]$ and $\left[^{143}\text{Nd}\right]$, respectively, multiplied by this constant $^{144}$Nd:Nd. Because
there is no evidence of isotopic fractionation during scavenging, in Eq. (2), only the external sources $\boldsymbol{s}_k$ differ between isotopes.
Thus, in practice, $\boldsymbol{\chi}_{\text{RNd}}^{\text{mod}}$ is computed by solving Eq. (2) with the sources replaced by $\boldsymbol{R}_k \boldsymbol{s}_k$ (element-wise multiplication),
where $\boldsymbol{R}_k$ is the vector of the isotopic ratio of Nd injected by source $k$. The modelled $\varepsilon_{\text{Nd}}$ values are then given by the vector
$\varepsilon_{\text{Nd}}^{\text{mod}} = \frac{\boldsymbol{\chi}_{\text{RNd}}^{\text{mod}}/\boldsymbol{\chi}_{\text{Nd}}^{\text{mod}}}{R_{\text{CHUR}}} - 1$ (where all the operations are element-wise).

### 2.1   Ocean circulation

The $\mathbf{T}_{\text{circ}}\,\boldsymbol{\chi}_{\text{Nd}}^{\text{mod}}$ term in Eq. (2) captures the flux divergence of [Nd] as it gets carried along the mean ocean currents of the model
and mixed by subgrid-scale eddies. The advection–diffusion operator $\mathbf{T}_{\text{circ}}$ is represented as a $200160 \times 200160$ sparse matrix
and comes from the output of the OCIM v2.0 (DeVries and Holzer, 2019), which provides a state-of-the-art data-assimilated
steady-state ocean circulation (DeVries and Primeau, 2011; DeVries, 2014; DeVries and Holzer, 2019). Physically, $\mathbf{T}_{\text{circ}}$ can
be interpreted as the equivalent of $\nabla \cdot (\boldsymbol{u} - \mathbf{K}\nabla)$ where $\boldsymbol{u}$ is the climatological mean water velocity field and $\mathbf{K}$ is an eddy-

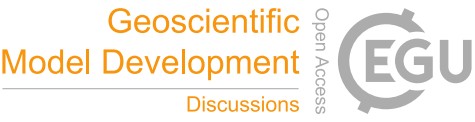

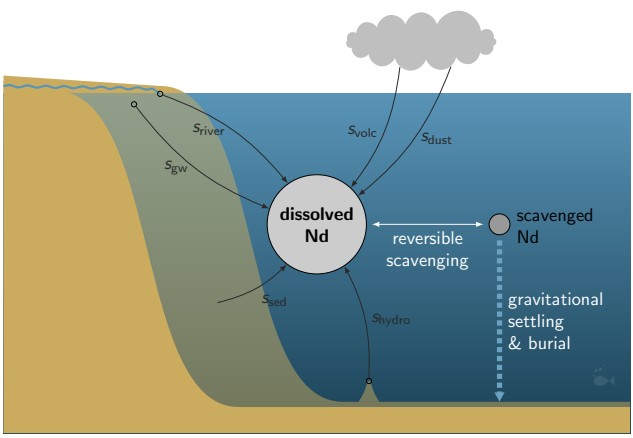

**Figure 1.** Diagram of the Nd-cycle model as implemented in GNOM v1.0. External sources of dissolved Nd are represented by black arrows. Localized sources, rivers, groundwaters, and hydrothermal vents, are indicated by a small circle at the origin of their respective arrows. A fraction of Nd is reversibly scavenged and pumped downwards. A fraction of scavenged Nd that reaches the sediments is buried in the sediments and removed from the system. Nd is also continuously transported by the ocean circulation model (not represented in the schematic).

diffusivity matrix of which the horizontal component is slanted along isopycnals. The spatial resolution of its grid is fixed at a
nominal $2° \times 2°$ in the horizontal and consists of 24 vertical levels of increasing height with depth. We emphasize that the OCIM v2.0 is particularly suited to this type of model because it arguably provides the best available estimate of the current-climate long-term large-scale ocean circulation while it affords spectacular computational efficiency.

## 2.2 External sources

The GNOM v1.0 explicitly represents six sources of Nd into the ocean (Fig. 1): (i) atmospheric mineral dust deposition, $s_{\mathrm{dust}}$,
(ii) atmospheric volcanic ash deposition, $s_{\mathrm{volc}}$ (iii) riverine discharge $s_{\mathrm{river}}$, (iv) groundwater discharge $s_{\mathrm{gw}}$, (v) sedimentary remobilization (including pore water fluxes), $s_{\mathrm{sed}}$, and (vi) hydrothermal-vent release, $s_{\mathrm{hydro}}$. The column vectors $s_k$ summed together constitute the total source of Nd (Eq. (2)). Each source term is detailed in the following sections. Their spatial patterns and isotopic signatures are shown on Fig. 2 and their magnitudes and contributions to the total inventory of Nd are collected in Table 3.

### 2.2.1 Aeolian dust


We assume that atmospheric dust deposition injects Nd in the surface ocean only. That is, soluble Nd from dust is instantly released as dissolved Nd in the top layer of the model grid. Although it can vary with location and mineralogy (Goldstein et al., 1984), for simplicity, we assume a constant dust Nd content of $(\mathrm{Nd:dust}) = 40\,\mu\mathrm{g\,g}^{-1}$. The spatial pattern of the dust source is prescribed by an atmospheric model output (Scanza et al., 2018) and is shown in Fig. 2a.





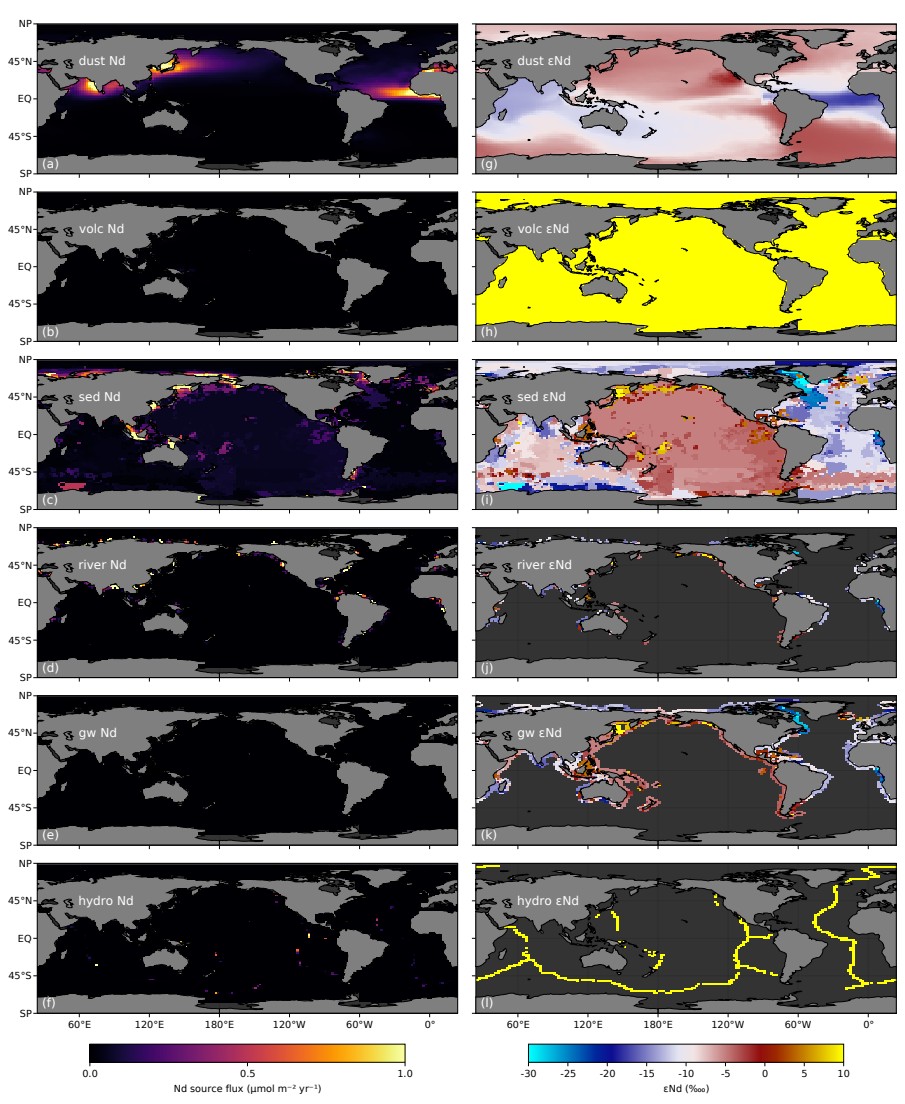

**Figure 2.** Vertically integrated Nd sources and corresponding vertical mean $\varepsilon_{\mathrm{Nd}}$.



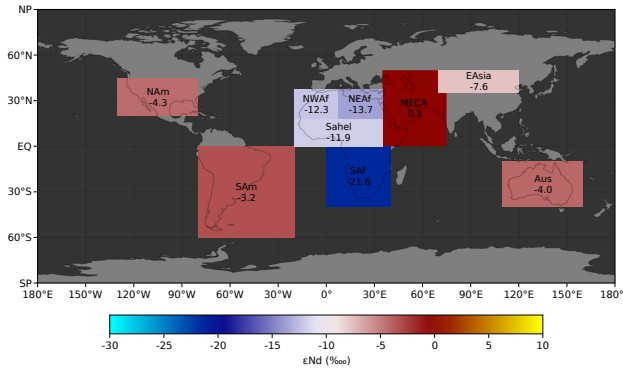

**Figure 3.** Extent of the dust regions of origin (Kok et al., 2021a, b; Adebiyi et al., 2020) and their $\varepsilon_{\text{Nd}}$ values as optimized in GNOM v1.0. See text for region names.

The isotopic signature of atmospheric mineral dust deposited on the ocean surface is not homogeneous (Goldstein et al., 1984). Instead, dust $\varepsilon_{\text{Nd}}$ varies with composition and mineralogy, which derives from its land origin. It is also likely that Nd solubility varies with composition and mineralogy. Thus, the GNOM v1.0 uses nine separate annual mineral dust deposition fields (dataset available from Adebiyi et al., 2020) from nine different regions. These dust deposition fields were generated by Kok et al. (2021a) and Kok et al. (2021b) who partitioned dust emissions according to nine different regions of origin, using the global climate model of Scanza et al. (2018). The nine regions we use are North-Western Africa (NWAf), North-Eastern Africa (NEAf), Southern Sahara and Sahel (Sahel), Middle East and Central Asia (MECA), East Asia (EAsia), North America (NAm), Australia (Aus), South America (SAm), and Southern Africa (SAf). Figure 3 shows the extent of these regions.

We assign a distinct Nd solubility and isotopic signature to each region of origin, controlled by the $2 \times 9$ corresponding parameters (denoted $\beta_r$ and $\varepsilon_r$ for each region $r$; see parameters Table 2). The dust source of Nd into the ocean is hence given by

$$\boldsymbol{s}_{\text{dust}} = \sum_r \beta_r \frac{(\text{Nd:dust}) \, \boldsymbol{\phi}_{\text{dust},r}}{M_{\text{Nd}} \, \Delta z_1} \left( \boldsymbol{z} = z_1 \right) \tag{3}$$

where $\boldsymbol{\phi}_{\text{dust},r}$ is the dust deposition flux from region $r$ taken from the Adebiyi et al. (2020) dataset and rearranged into a 200160-element vector, $\Delta z_1$ and $z_1$ are the height and depth of the top layer of the model grid, $\boldsymbol{z}$ is the 200160-element vector of depths, and $M_{\text{Nd}} = 144.24 \, \text{g mol}^{-1}$ is the molar mass of Nd. (All the operations in Eq. (3) are element wise and $(\boldsymbol{z} = z_1)$ acts like a mask so that $\boldsymbol{s}_{\text{dust}}$ only injects Nd in the top layer of the model grid.)

Each isotopic signature parameter $\varepsilon_r$ uniquely defines the isotopic ratio of each region via $R_r = R_{\text{CHUR}} (\varepsilon_r + 1)$, which is then used to compute the dust source for the RNd tracer via

$$\boldsymbol{R}_{\text{dust}} \boldsymbol{s}_{\text{dust}} = \sum_r R_r \beta_r \frac{(\text{Nd:dust}) \, \boldsymbol{\phi}_{\text{dust},r}}{M_{\text{Nd}} \, \Delta z_1} \left( \boldsymbol{z} = z_1 \right). \tag{4}$$

This allows for the aeolian dust source to carry an elaborate and more realistic isotopic signature than previous models (Fig. 2g). Figure 3 also shows the optimized $\varepsilon_r$ values of each region.





### 2.2.2 Volcanic ash

Despite a smaller atmospheric loading than mineral dust, we include volcanic ash as a separate, potentially important, aeolian source of Nd because of its typically high reactivity and solubility compared to mineral dust. This reactivity partly reflects the high surface area of volcanic ash, the thermodynamic instability of volcanic glass, and its typically mafic composition, which

is more easily weathered than more felsic compositions (Gaillardet et al., 1999; Dessert et al., 2003). We use the geographic pattern of volcanic ash deposition from the work of Chien et al. (2016), which provides estimates of the global deposition fields of dust and soluble iron from different aerosol types (mineral dust, volcanic ash, combustion fire, and so on). Assuming a constant neodymium content identical to dust, the volcanic-ash source of Nd into the ocean is thus given by

$$\boldsymbol{s}_{\mathrm{volc}} = \beta_{\mathrm{volc}} \frac{(\mathrm{Nd:dust})\, \boldsymbol{\phi}_{\mathrm{volc}}}{M_{\mathrm{Nd}}\, \Delta z_1}\, (\boldsymbol{z} = z_1). \tag{5}$$

where $\boldsymbol{\phi}_{\mathrm{volc}}$ is the column vector of the volcanic ash deposition flux from the Chien et al. (2016) dataset and $\beta_{\mathrm{volc}}$ is the Nd solubility in volcanic ash. Similarly to the dust-source formulation, the magnitude of the volcanic-ash source of RNd is controlled by the parameter $\varepsilon_{\mathrm{volc}}$,

$$\boldsymbol{R}_{\mathrm{volc}} \boldsymbol{s}_{\mathrm{volc}} = R_{\mathrm{volc}}\, \beta_{\mathrm{volc}} \frac{(\mathrm{Nd:dust})\, \boldsymbol{\phi}_{\mathrm{volc}}}{M_{\mathrm{Nd}}\, \Delta z_1}\, (\boldsymbol{z} = z_1). \tag{6}$$

where $R_{\mathrm{volc}} = R_{\mathrm{CHUR}}(\varepsilon_{\mathrm{volc}} + 1)$. (Note that $\boldsymbol{R}_{\mathrm{volc}} = R_{\mathrm{volc}}$ everywhere because the volcanic-ash source comprises a single term,

unlike the region-of-origin-partitioned dust source.) The geographical patterns of the volcanic-ash source and its uniform isotopic signature are shown in Fig. 2b and h.

### 2.2.3 Sediments

Sedimentary Nd is likely released via pore waters located in the upper few centimetres below the seafloor (e.g., Elderfield and Sholkovitz, 1987; Sholkovitz et al., 1989; Haley et al., 2004; Lacan and Jeandel, 2005; Wilson et al., 2013; Haley et al.,

2017; Abbott et al., 2015a, b; Du et al., 2016, and references therein). The flux magnitude of this sedimentary release likely depends on sediment composition, reactivity, and other environmental factors (Lacan and Jeandel, 2005; Pearce et al., 2013; Wilson et al., 2013; Blaser et al., 2016, 2020). Oxygenation and organic matter fluxes also potentially play an important role in early diagenesis (Elderfield and Sholkovitz, 1987; Sholkovitz et al., 1989, 1992; Haley et al., 2004; Lacan and Jeandel, 2005; Wilson et al., 2013). At high latitudes, mechanical glacial erosion likely increases sedimentary Nd fluxes by exposing fresh

material and increasing surface area by producing fine particulates (Anderson, 2005; von Blanckenburg and Nägler, 2001), while increased bottom water eddy-kinetic energy likely enhances Nd release (Lacan and Jeandel, 2005; Gardner et al., 2018; Pöppelmeier et al., 2019).

There is no established quantitative flux model for sedimentary Nd release that works on the global scale. Therefore, the GNOM v1.0 implements the sedimentary Nd flux into the ocean as a simple function of depth $z$ and local sedimentary $\varepsilon_{\mathrm{Nd}}$.

The "base" sedimentary Nd flux, $\phi(z)$, is modelled as an exponential function of depth,

$$\phi_{\mathrm{sed}}(z) = (\phi_0 - \phi_\infty) e^{-z/z_0} + \phi_\infty, \tag{7}$$



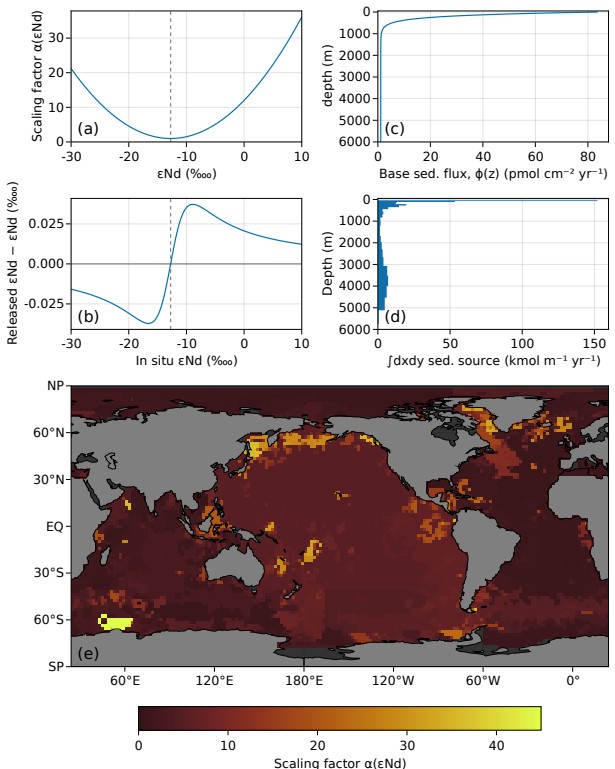

**Figure 4.** (a) Sedimentary source enhancement as a quadratic function of observed $\varepsilon_{\mathrm{Nd}}$. (b) Difference between effectively released $\varepsilon_{\mathrm{Nd}}$ and in situ $\varepsilon_{\mathrm{Nd}}$ as a function of in situ $\varepsilon_{\mathrm{Nd}}$. (c) Sedimentary source flux profile as a function of depth. (d) Horizontally integrated sedimentary source. (e) Map of sedimentary reactivity / scaling factor $\alpha(\varepsilon_{\mathrm{sed}})$.

where $\phi_0$, $\phi_\infty$, and $z_0$ are optimizable parameters. The rationale here is that for $\phi_\infty \approx 0$, the flux profile resembles that of eddy kinetic energy and particulate organic matter fluxes, which are larger near the surface and coastal areas and smaller in the deepest parts of the ocean. However, the optimization only enforces weak direct constraints on the parameters such that this

parameterization of sedimentary flux allows for many different profile shapes (for example similar to the linearly increasing flux profile of Du et al., 2020, Fig. 1C). The optimized base Nd flux profile as a function of depth is shown in Fig. 4c.

The base sedimentary flux is further scaled by a reactivity factor, $\alpha$, which controls the effective sedimentary Nd release. Sediment reactivity is modelled via a simple parameterized quadratic function of local sedimentary $\varepsilon_{\mathrm{Nd}}$,

$$\alpha(\varepsilon_{\mathrm{Nd}}) = a\left(\frac{\varepsilon_{\mathrm{Nd}} - c}{\varepsilon_{10}}\right)^2 + 1, \tag{8}$$

where the optimizable parameters $a$ and $c$ control the curvature and minimum of the quadratic while $\varepsilon_{10} = 10\,‰$ is a normalization constant. Sedimentary $\varepsilon_{\mathrm{Nd}}$ is taken from a modified version of the interpolated global map of sedimentary $\varepsilon_{\mathrm{Nd}}$ of Robinson et al. (2021). (Our modification caps the central and North Pacific $\varepsilon_{\mathrm{Nd}}$ values to a minimum of $-5\,‰$ because it appears that Robinson et al. (2021) artificially disconnected seafloor areas at the $180°$ meridian, resulting in $\varepsilon_{\mathrm{Nd}}$ values that we





flagged as too negative.) This quadratic parameterization is motivated by the fact that extreme sedimentary $\varepsilon_{\mathrm{Nd}}$ values are often

associated with rather fresh, and thus reactive, detrital material. Maybe coincidentally, extremely high $\varepsilon_{\mathrm{Nd}}$ values are generally associated with relatively young volcanic Nd that is more reactive and readily soluble (Lacan and Jeandel, 2005; Pearce et al., 2013; Wilson et al., 2013; Blaser et al., 2016, 2020). The quadratic function $\alpha(\varepsilon_{\mathrm{Nd}})$ is shown in Fig. 4a and the resulting scaling factor for the global map is show in Fig. 4e.

Finally, to account for large glaciers that may produce fine-grained glacial flour from previously unexposed bedrock that

likely contains reactive Nd (von Blanckenburg and Nägler, 2001), we additionally scale Nd release from the sedimentary source along the coast of Greenland by a factor $\alpha_{\mathrm{GRL}}$. (We did not account for potentially enhanced Nd release in Antarctic.) Combined with the reactivity $\alpha$, the resulting sedimentary source is given by

$$\boldsymbol{s}_{\mathrm{sed}} = \boldsymbol{\alpha}_{\mathrm{GRL}}\,\alpha(\boldsymbol{\varepsilon}_{\mathrm{sed}})\,\frac{\phi_{\mathrm{sed}}(\boldsymbol{z}_{\mathrm{bot}})}{\Delta\boldsymbol{z}}\,\boldsymbol{f}_{\mathrm{topo}}, \tag{9}$$

where $\boldsymbol{\alpha}_{\mathrm{GRL}}$ is the vector equal to $\alpha_{\mathrm{GRL}}$ for grid cells against the coast of Greenland and equal to 1 otherwise, $\boldsymbol{\varepsilon}_{\mathrm{sed}}$ is the column

vector of the $\varepsilon_{\mathrm{Nd}}$ map of Robinson et al. (2021) repeated at all depths throughout the water column, $\boldsymbol{z}_{\mathrm{bot}}$ is the vector of the depths of the bottom of each model grid cell, $\Delta\boldsymbol{z}$ is the vector of the height of each model grid cell, and $\boldsymbol{f}_{\mathrm{topo}}$ is a mask equal to 1 for grid cells on the seafloor and equal to 0 otherwise. (Functions and operations in Eq. (9) are applied element wise.) The horizontally integrated sedimentary source is shown Fig. 4d.

For the RNd sedimentary flux, we use the interpolated seafloor map of $\varepsilon_{\mathrm{Nd}}$ values from Robinson et al. (2021) (modifed

with a $-5\,‰$ minimum in the Pacific north of $40°\mathrm{S}$). For grid cells with heterogeneous $\varepsilon_{\mathrm{Nd}}$ in sediments, our quadratic implementation of the reactivity $\alpha$ as a function of $\varepsilon_{\mathrm{Nd}}$ implies a statistical "shift" of the mean released $\varepsilon_{\mathrm{Nd}}$ towards extreme values because extremely light or heavy Nd is released more efficiently. Assuming that $\boldsymbol{\varepsilon}_{\mathrm{sed}}$ represents the observed in situ mean of a normally distributed $\varepsilon_{\mathrm{Nd}}$ sediment composition in each grid cell and a uniform global standard deviation $\sigma_{\varepsilon}$ *within* each grid cell, the $\varepsilon_{\mathrm{Nd}}$ that is effectively released at any location, denoted $\varepsilon_{\mathrm{sed}}^{\mathrm{eff}}$, is given by

$$\varepsilon_{\mathrm{sed}}^{\mathrm{eff}} = \frac{a\,(\boldsymbol{\varepsilon}_{\mathrm{sed}} - 2\,(c - \boldsymbol{\varepsilon}_{\mathrm{sed}}))\,\sigma_{\varepsilon}^2 + (a\,(c - \boldsymbol{\varepsilon}_{\mathrm{sed}})^2 + \varepsilon_{10}^2)\,\boldsymbol{\varepsilon}_{\mathrm{sed}}}{a\,\sigma_{\varepsilon}^2 + a\,(c - \boldsymbol{\varepsilon}_{\mathrm{sed}})^2 + \varepsilon_{10}^2}. \tag{10}$$

The difference between $\varepsilon_{\mathrm{sed}}^{\mathrm{eff}}$ and in situ $\boldsymbol{\varepsilon}_{\mathrm{sed}}$ is shown in Fig. 4b to shift $\varepsilon_{\mathrm{Nd}}$ values by up to about $\pm0.03\,‰$. In other words, for in situ $\varepsilon_{\mathrm{Nd}}$ values lower than the minimum of $\alpha$ (dashed gray line in Fig. 4a, b), the released $\varepsilon_{\mathrm{Nd}}$ value is pushed toward even lower values, and for in situ $\varepsilon_{\mathrm{Nd}}$ values greater than the minimum of $\alpha$, released $\varepsilon_{\mathrm{Nd}}$ is pushed toward even larger values. (The derivation of Eq. (10) is given in Appendix C). Applying $\boldsymbol{R}_{\mathrm{sed}}^{\mathrm{eff}} = R_{\mathrm{CHUR}}\,(\boldsymbol{\varepsilon}_{\mathrm{sed}}^{\mathrm{eff}} + 1)$ gives the sedimentary source of RNd

as $\boldsymbol{R}_{\mathrm{sed}}^{\mathrm{eff}}\boldsymbol{s}_{\mathrm{sed}}$.

### 2.2.4 Rivers

For riverine sources, we use the Global River Flow and Continental Discharge Dataset (Dai, 2017), originally described by Dai and Trenberth (2002) and later updated by Dai et al. (2009) and Dai (2016). This dataset provides an estimate of the volumetric flow rate of the 200 largest rivers on Earth. We assume that all rivers share the same Nd concentration $c_{\mathrm{river}}$, which is the





parameter that controls the global riverine source magnitude (see Table 2). Hence, the vector of the riverine source is given by

$$\boldsymbol{s}_{\text{river}} = c_{\text{river}} \frac{\boldsymbol{Q}_{\text{river}}}{\boldsymbol{v}} \tag{11}$$

where $\boldsymbol{Q}_{\text{river}}$ is the vector of the volumetric flow rates of the rivers from the dataset of Dai (2017) gridded (cumulatively) onto the OCIM v2.0 grid and where $\boldsymbol{v}$ is the vector of the volumes of the model grid cells. Note that in order to prevent numerical noise from the large gradients caused by the discrete nature of their distribution, we additionally artificially spatially smooth
out the riverine sources, shown in Fig. 2d.

Riverine $\varepsilon_{\text{Nd}}$ values are taken from the global map of interpolated sedimentary $\varepsilon_{\text{Nd}}$ by Robinson et al. (2021), which we also use for the sedimentary source, so that the RNd riverine source is given by $\boldsymbol{R}_{\text{sed}}^{\text{eff}} \boldsymbol{s}_{\text{river}}$. We note that the sedimentary $\varepsilon_{\text{Nd}}$ map of Robinson et al. (2021) overlays the nearest continental $\varepsilon_{\text{Nd}}$ signal where sediment thickness is more than $1\,\text{km}$, such that the $\varepsilon_{\text{Nd}}$ of the GNOM v1.0 riverine sources are mostly from continental measurements that lie within or close to the river drainage
basins. Riverine $\varepsilon_{\text{Nd}}$ values are shown in Fig. 2j.

### 2.2.5  Groundwater

Neodymium also enters the oceans via coastal groundwater (Johannesson and Burdige, 2007). We use the coastal submarine and terrestrial groundwater discharge dataset of Luijendijk et al. (2019), described by Luijendijk et al. (2020), which provides the location and volumetric flow rate of 40,082 coastal watersheds. Similarly to the riverine sources, we assume that [Nd] is
constant across riversheds, such that the single parameter $c_{\text{gw}}$ controls the global magnitude of the groundwater Nd source (see Table 2). The groundwater Nd source is given by

$$\boldsymbol{s}_{\text{gw}} = c_{\text{gw}} \frac{\boldsymbol{Q}_{\text{gw}}}{\boldsymbol{v}}, \tag{12}$$

where $\boldsymbol{Q}_{\text{gw}}$ is the groundwater volumetric flow rates from the dataset of Luijendijk et al. (2020), gridded cumulatively onto the GNOM grid. The pattern of $\boldsymbol{s}_{\text{gw}}$ is shown in Fig. 2e.
Following Jeandel et al. (2007), we assume that the $\varepsilon_{\text{Nd}}$ of Nd released through groundwater is determined by the local lithogenic isotopic composition. However, instead of the dataset of Jeandel et al. (2007), we use the more recent Robinson et al. (2021) data, exactly like for the riverine source. These $\varepsilon_{\text{Nd}}$ values, which are located near the coast, are likely adequately representing the local lithogenic composition because Robinson et al. (2021) assign continental values where sediment thickness is greater than $1\,\text{km}$. The groundwater $\varepsilon_{\text{Nd}}$ values are shown on Fig. 2k.

### 2.2.6  Hydrothermal vents

A minor fraction of the marine neodymium budget presumably comes from hydrothermal vents, which deliver likely young Nd (high $\varepsilon_{\text{Nd}}$) along the mid-ocean ridges (Piepgras and Wasserburg, 1985; Stichel et al., 2018). Here, we assume that the release of hydrothermal Nd is proportional to that of helium. For consistency, we use the mantle helium source field that was used in the data assimilation of the OCIM v2.0 (DeVries and Holzer, 2019). The global magnitude and $\varepsilon_{\text{Nd}}$ of the hydrothermal Nd





source are set by parameters $\sigma_{\mathrm{hydro}}$ and $\varepsilon_{\mathrm{hydro}}$, respectively (see Table 2), with

$$\boldsymbol{s}_{\mathrm{hydro}} = \sigma_{\mathrm{hydro}} \frac{\boldsymbol{s}_{\mathrm{He}}}{\boldsymbol{v}^{\mathsf{T}} \boldsymbol{s}_{\mathrm{He}}}, \tag{13}$$

where $\boldsymbol{s}_{\mathrm{He}}$ is the vector of the $^{3}\mathrm{He}$ mantle source and $\boldsymbol{v}^{\mathsf{T}} \boldsymbol{s}_{\mathrm{He}}$ is its global magnitude, i.e., its volume integral, used here for normalization. (One can easily check that $\boldsymbol{v}^{\mathsf{T}} \boldsymbol{s}_{\mathrm{hydro}} = \sigma_{\mathrm{hydro}}$.) The hydrothermal source of RNd is simply given by $\boldsymbol{R}_{\mathrm{hydro}} \boldsymbol{s}_{\mathrm{hydro}} = R_{\mathrm{CHUR}} \left(1 + \varepsilon_{\mathrm{hydro}}\right) \boldsymbol{s}_{\mathrm{hyrdo}}$. Fig. 2f and l show the spatial distribution of the hydrothermal Nd source and its $\varepsilon_{\mathrm{Nd}}$, respectively.

## 2.3 Reversible scavenging

Neodymium is removed from the system through scavenging onto particles. We follow, e.g., Bacon and Anderson (1982) and Siddall et al. (2008) and assume that dissolved and scavenged Nd are exchanged via a first-order kinetic reaction,

$$\mathrm{Nd} + X \rightleftharpoons X\mathrm{Nd}, \tag{R1}$$

where $X$ is a given particle type. The rate of change of $[\mathrm{Nd}]$ in reaction (R1) can be written as

$$-\frac{\partial [\mathrm{Nd}]}{\partial t} = \frac{\partial [X\mathrm{Nd}]}{\partial t} = k_X^+ [\mathrm{Nd}] [X] - k_X^- [X\mathrm{Nd}] \tag{14}$$

with equilibrium constant

$$K_X = \frac{k_X^+}{k_X^-} = \frac{[X\mathrm{Nd}]}{[\mathrm{Nd}][X]}. \tag{15}$$

We further assume that each scavenging particle type $X$ has a constant settling velocity $w_X$ that dominates the transport rates of the ocean circulation. For each particle type $X$, we construct its flux divergence operator, denoted $\mathbf{T}_X$, such that $\mathbf{T}_X \boldsymbol{x}$, is the discrete equivalent of $\nabla \cdot (\boldsymbol{w}_X [X])$ (where $\boldsymbol{x}$ is the particulate concentration vector and $\boldsymbol{w}_X$ is the downward three-dimensional settling velocity vector).

However, we avoid the explicit simulation of particle $X$ and scavenged particulate $X\mathrm{Nd}$ by assuming that reaction (R1) operates on even shorter timescales than that of vertical particulate transport and ocean transport, such that dissolved and scavenged Nd are in equilibrium and Eq. (15) uniquely determines $[X\mathrm{Nd}]$ from $[\mathrm{Nd}]$ and a prescribed particle concentration $[X]$. In practice, we thus model scavenged Nd implicitly by transporting only a fraction of Nd downwards with gravitational settling velocity $w_X$ (The fraction is determined by Eq. (15)). This is done by adding the steady-state tracer equation for $X\mathrm{Nd}$ as it is scavenged by each particle type into Eq. (2), such that the vertical transport of $X\mathrm{Nd}$ is added to $\mathbf{T}_{\mathrm{scav}}$ and the scavenging rates from Eq. (14) cancel out. We further assume that a fraction $f_X$ of the scavenged Nd that reaches the seafloor is removed from the system, providing a net sink for our model. (Note that this is the same implicit approach as in the AWESOME OCIM (John et al., 2020).)

We consider four different particle types for scavenging Nd. (i) Scavenging by dust particles is modelled using the dust deposition fields of Kok et al. (2021b), assuming a vertically constant concentration as dust particles settle with velocity $w_{\mathrm{dust}} = 1\,\mathrm{km\,yr^{-1}}$ through the water column. (ii) Scavenging by particulate organic carbon (POC) is modelled using the three-dimensional POC concentration field from the work of Weber et al. (2018) and following the AWESOME OCIM implementation of John et al. (2020), with a settling velocity $w_{\mathrm{POC}} = 40\,\mathrm{m\,d^{-1}}$. (iii) Scavenging by biogenic silica (bSi), or opal, is





modelled using a simple, nutrient-restoring offline model of Si-cycling described in Appendix A. (iv) A particle-independent scavenging is included to prevent accumulation of Nd where the concentration fields of dust, POC, and opal are unrealistically low. This mechanism effectively behaves like spontaneous precipitation, and, as such, will be referred to as "precipitation" throughout this study (subscript "prec"). In practice, precipitation is implemented by using a spatially uniform fictitious parti-

cle concentration of $1\,\mathrm{mol\,m^{-3}}$ that settles with velocity $w_{\mathrm{prec}} = 0.7\,\mathrm{km\,yr^{-1}}$. We note that this additional particle-independent scavenging sink could compensate for additional types of particles not currently implemented in the model. Overall, the scavenging transport operator is thus defined by summing the flux divergence for all particle types and using Eq. (15), i.e.,

$$\mathbf{T}_{\mathrm{scav}} = K_{\mathrm{dust}}\,\mathbf{T}_{\mathrm{dust}}\,\mathbf{D}_{\mathrm{dust}} + K_{\mathrm{POC}}\,\mathbf{T}_{\mathrm{POC}}\,\mathbf{D}_{\mathrm{POC}}$$

$$+ K_{\mathrm{bSi}}\,\mathbf{T}_{\mathrm{bSi}}\,\mathbf{D}_{\mathrm{bSi}} + K_{\mathrm{prec}}\,\mathbf{T}_{\mathrm{prec}}, \quad (16)$$

where $\mathbf{D}_X$ is a diagonal matrix with diagonal $\boldsymbol{x}$, the vector of the concentrations of particle type $X$. Hence, for each scavenging particle type $X$, the corresponding scavenging rate and downward transport is controlled by the concentration $[X]$, the equilibrium constant $K_X$, the settling velocity $w_X$, and the burial fraction $f_X$.

## 2.4 Optimization

The output of our model is governed by a set of $43$ free parameters that control the magnitude and $\varepsilon_{\mathrm{Nd}}$ of each source as well as the reversible scavenging and burial rates. The computational speed afforded by the model implementation allows us to jointly optimize almost all of these parameters by minimizing the mismatch of modelled and observed $[\mathrm{Nd}]$ and $\varepsilon_{\mathrm{Nd}}$ values. This is done in practice by minimizing an objective function $\hat{f}(\boldsymbol{p})$ that quantifies the mismatch between model and observations, for a given set of parameters $\boldsymbol{p}$.

### 2.4.1 Objective function


The mismatch with each observation is quantified by the square of the difference between the observed value and the modelled value from the closest grid cell. Because we use observations of $[\mathrm{Nd}]$ and $\varepsilon_{\mathrm{Nd}}$, the mismatch function, denoted $f$, depends on the 3D fields of the two modelled tracers ($\boldsymbol{\chi}_{\mathrm{Nd}}^{\mathrm{mod}}$ and $\boldsymbol{\chi}_{\mathrm{RNd}}^{\mathrm{mod}}$). We also include an additional cost for parameter values themselves. The mismatch function is defined by


$$f(\boldsymbol{\chi}_{\mathrm{Nd}}^{\mathrm{mod}}, \boldsymbol{\chi}_{\mathrm{RNd}}^{\mathrm{mod}}, \boldsymbol{p}) = \omega_{\mathrm{Nd}} \frac{1}{2} \frac{\sum_{\boldsymbol{r} \in \mathscr{O}_{\mathrm{Nd}}} \left( \boldsymbol{\chi}_{\mathrm{Nd}}^{\mathrm{mod}}[\boldsymbol{r}] - \boldsymbol{\chi}_{\mathrm{Nd}}^{\mathrm{obs}}[\boldsymbol{r}] \right)^2}{\sum_{\boldsymbol{r} \in \mathscr{O}_{\mathrm{Nd}}} \left( \boldsymbol{\chi}_{\mathrm{Nd}}^{\mathrm{obs}}[\boldsymbol{r}] \right)^2}$$

$$+ \omega_{\varepsilon_{\mathrm{Nd}}} \frac{1}{2} \frac{\sum_{\boldsymbol{r} \in \mathscr{O}_{\varepsilon_{\mathrm{Nd}}}} \left( \boldsymbol{\varepsilon}_{\mathrm{Nd}}^{\mathrm{mod}}[\boldsymbol{r}] - \boldsymbol{\varepsilon}_{\mathrm{Nd}}^{\mathrm{obs}}[\boldsymbol{r}] \right)^2}{\sum_{\boldsymbol{r} \in \mathscr{O}_{\varepsilon_{\mathrm{Nd}}}} \left( \boldsymbol{\varepsilon}_{\mathrm{Nd}}^{\mathrm{obs}}[\boldsymbol{r}] \right)^2} - \omega_{\boldsymbol{p}} \sum_i \log(P(\boldsymbol{p}_i)), \quad (17)$$

where the first term represents the normalized mismatch between modelled and observed $[\mathrm{Nd}]$, the second term represents the normalized mismatch between modelled and observed $\varepsilon_{\mathrm{Nd}}$, and the last term represents the inverse of the likelihood of the

model parameters. We detail each term below.

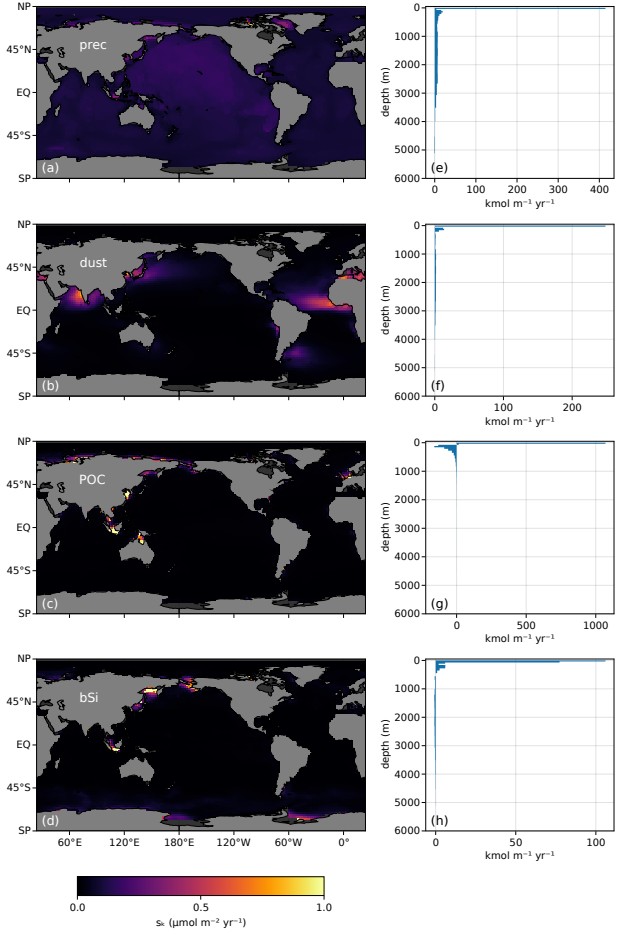

**Figure 5.** (a)–(d) Vertically integrated and (e)–(f) horizontally integrated scavenging by "precipitation", dust, POC, and opal particles, where positive values represent Nd removal and negative values represent a source of "pumped" Nd.

In Eq. (17), $\boldsymbol{\chi}_{\text{Nd}}^{\text{obs}}$ is a vector of all the [Nd] observations and $\boldsymbol{\chi}_{\text{Nd}}^{\text{obs}}[\boldsymbol{r}]$ denotes the observed [Nd] at location $\boldsymbol{r}$, which spans all the locations of observations, $\mathcal{O}_{\text{Nd}}$. (One can think of $\boldsymbol{r}$ as indexing the vector of observations $\boldsymbol{\chi}_{\text{Nd}}^{\text{obs}}$.) We compare each observation with the model output from the closest model-grid cell, denoted $\boldsymbol{\chi}_{\text{Nd}}^{\text{mod}}[\boldsymbol{r}]$ for simplicity. (Technically, the location of the observed and modelled value being compared may not match, in which case we use the closest "wet" model grid

cell using a nearest-neighbor algorithm.) We use the same approach for $\varepsilon_{\text{Nd}}$, by comparing the modelled vector $\boldsymbol{\varepsilon}_{\text{Nd}}^{\text{mod}}$ to the observed vector $\boldsymbol{\varepsilon}_{\text{Nd}}^{\text{obs}}$ at the locations $\boldsymbol{r} \in \mathcal{O}_{\varepsilon_{\text{Nd}}}$ of each $\varepsilon_{\text{Nd}}$ observation. The $\omega_{\text{Nd}}$ and $\omega_{\varepsilon_{\text{Nd}}}$ values, fixed at 1 in this study, control the relative contributions of the mismatches in Nd concentrations and $\varepsilon_{\text{Nd}}$ values. Given these, an error of $1\,\%_{00}$ in $\varepsilon_{\text{Nd}}$ weighs the same as an error of about $4.5\,\text{pM}$.

The third term of Eq. (17) adds a direct penalty constraint on the parameters to prevent them from reaching unrealistic values.

For each parameter $\boldsymbol{p}_i$ we prescribe a realistic domain $D_i$ and an initial guess (see Table 2) that we use to determine a reasonable


prior probability distribution $d_i$ and to randomize the initial parameter values. Specifically, each parameter with a semi-infinite range (from 0 to $\infty$) is given a log-normal prior of which the logarithm has mean equal to the logarithm of the initial guess and has variance equal to 1. Each parameter with a finite range is given a logit-normal prior that is scaled and shifted such that its support matches the range $D_i$ exactly and such that the initial guess equals the median of $d_i$. For example, the $\phi_0$ parameter

for the sedimentary flux at the surface is given the $(0, \infty)$ range and an initial guess of $20 \, \mathrm{pmol \, cm^{-2} \, yr^{-1}}$. Taken as a random variable, the prior distribution of $\phi_0$ is the log-normal distribution such that $\log(\phi_0/(\mathrm{pmol \, cm^{-2} \, yr^{-1}})) \sim \mathcal{N}(\log(20), 1)$. ($\mathcal{N}(\mu, \sigma^2)$ denotes the normal distribution with mean $\mu$ and variance $\sigma^2$.)

For the performance and robustness of the optimization, we additionally perform a variable transform $\lambda_i$ on each parameter using a bijection from the parameter domain $D_i$ to $(-\infty, \infty)$. This variable transform prevents parameters from

reaching beyond their prescribed ranges. We also carefully chose the bijection such that the prior distribution is normally distributed in the transformed parameter space, and thus incurs an inverse log-likelihood that is quadratic, a property that benefits the performance of the optimization. In the case of the parameter $\phi_0$, which is transformed via the bijection $\lambda_i$ : $\phi_0 \mapsto \log(\phi_0/(\mathrm{pmol \, cm^{-2} \, yr^{-1}}))$, the corresponding transformed random variable is normally distributed by construction. For bounded parameters, such as $\beta_{\mathrm{volc}}$, a shifted and scaled logit transform is applied, which also yields a transformed random

variable that is normally distributed by construction.

The $\omega_{\boldsymbol{p}}$ value, fixed at $10^{-4}$ in this study, controls the relative size of the penalty for the parameters compared to the cost of Nd and $\varepsilon_{\mathrm{Nd}}$. It is chosen such that the [Nd] and $\varepsilon_{\mathrm{Nd}}$ mismatch costs are generally about 2 to 3 orders of magnitude larger than the parameter penalty (although there is no bound on the parameter penalty for extreme parameter values). The primary role of this added parameter cost and the associated variable transform is to improve the convergence rate of the optimization and

help prevent it from "getting stuck" in valleys of parameter space (see, e.g., Wright et al., 1999).

The objective function depends on $\boldsymbol{p}$ only and is defined by

$$\hat{f}(\boldsymbol{p}) \equiv f\big(\boldsymbol{\chi}_{\mathrm{Nd}}^{\mathrm{mod}}(\boldsymbol{p}), \varepsilon_{\mathrm{Nd}}^{\mathrm{mod}}(\boldsymbol{p}), \boldsymbol{p}\big), \tag{18}$$

where we have explicitly marked $\boldsymbol{\chi}_{\mathrm{Nd}}^{\mathrm{mod}}(\boldsymbol{p})$ and $\varepsilon_{\mathrm{Nd}}^{\mathrm{mod}}(\boldsymbol{p})$ as functions of the parameters $\boldsymbol{p}$. That is, for any choice of parameters $\boldsymbol{p}$, before evaluating the model mismatch as quantified by the objective function, we must first compute the vectors $\boldsymbol{\chi}_{\mathrm{Nd}}^{\mathrm{mod}}$ and

$\boldsymbol{\chi}_{\mathrm{RNd}}^{\mathrm{mod}}$ by solving for the steady-state solution to Eq. (2). The gradient, $\nabla \hat{f}$, and Hessian, $\nabla^2 \hat{f}$, of the objective function are computed using a combination of autodifferentiation and adjoint techniques available from within the AIBECS.jl package (Pasquier, 2020a; Pasquier et al., under review) or specifically developed in parallel for computational efficiency (F1Method.jl, Pasquier, 2020b).

### 2.4.2   Neodymium and $\varepsilon_{\mathbf{Nd}}$ data

The [Nd] and $\varepsilon_{\mathrm{Nd}}$ observations used in this study consist of 3 datasets: (i) the pre-GEOTRACES compilation of Nd and $\varepsilon_{\mathrm{Nd}}$ data by van de Flierdt et al. (2016), (ii) the GEOTRACES Intermediate Data Product 2017 (IDP17, Schlitzer et al., 2018) (including specifically Nd-linked publications: Stichel et al., 2012a, b, 2015; Garcia-Solsona et al., 2014; Basak et al., 2015; Fröllje et al., 2016; Lambelet et al., 2016, 2018; Behrens et al., 2018a, b), and (iii) our post-IDP17 compilation of data from



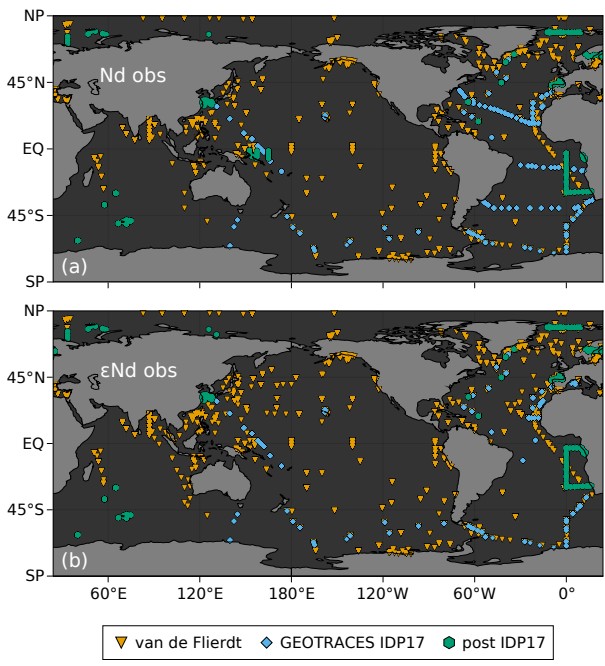

**Figure 6.** Locations of (a) Nd and (b) $\varepsilon_{\mathrm{Nd}}$ observations used in this study, labelled per the data source.

the Indian Ocean (Amakawa et al., 2019), the Barents Sea (Laukert et al., 2018; Laukert et al., 2019), the northern Iceland
Basin (Morrison et al., 2019), the Northwestern Pacific (Che and Zhang, 2018), the Kerguelen Plateau (Grenier et al., 2018),
the southeastern Atlantic Ocean (GA08, Rahlf et al., 2020; Rahlf et al., 2019; Rahlf et al., 2021; Rahlf et al., 2020), the Bay of
Biscay (Dausmann et al., 2020; Dausmann et al., 2019), the Western North Atlantic (Stichel et al., 2020; Stichel et al., 2020),
the Arctic (Laukert et al., 2017; Laukert et al., 2017a, d), and the Bermuda Atlantic Time-series Study (BATS; Laukert et al.,
2017; Laukert et al., 2017b, c). The spatial distribution of these observations as used in this study are shown in Fig. 6.

**2.4.3   Minimization algorithm**

We use the Newton Trust Region algorithm from the Optim.jl package (Mogensen and Riseth, 2018) to minimize the objective
function $\hat{f}(\boldsymbol{p})$. This requires, at every iteration, the objective function, its gradient, and its Hessian, which are evaluated using
the F1Method.jl and AIBECS.jl packages (Pasquier, 2020a; Pasquier et al., under review; Pasquier, 2020b).

Thanks to the computationally efficient gradient optimization algorithm that leverages gradient and Hessian information,
the entire optimization run takes a few hours on a modern laptop. In our experience, for comparison, using the more standard
finite-differences approach or an optimization algorithm that does not have access to derivatives would likely take multiple
months.

Because the Newton Trust Region algorithm performs local optimization rather than *global* optimization, we leverage the
computational speed by running multiple optimization runs starting from randomized initial parameter values sampled from





the parameter distributions $d_i$. Although not all optimization runs end up in the same state, due to the likely many local minima, we find that most of them converge towards the solution with the smallest objective-function value, which we denote as our "best" estimate, and out of which all the figures in this manuscript are created from.

## 3  Results

### 3.1  Parameter values

Our "best" estimate of the state of the Nd cycle is given by the set of parameters that minimizes the objective function defined in Eqs. (17)–(18). We emphasize that our estimate is determined by a local minimum of a specific parameterization, such that "best" here is somewhat subjective. In all likelihood, there exist other models and other parameter choices which produce a similar fit to global observations, though we expect all such models to capture the same key features of global Nd biogeochemical cycling. Initial guesses and final parameter values, along with unit, prescribed range, and a brief description, are given in

Table 2. Parameter final values, as well as their prior distributions, are shown in Fig. B1.

In Table 2, parameters without a range indicate that they were not optimized and held fixed at given previous model or literature values. For example, in the case of scavenging by each particle type $X$, we only optimized $K_X$ and $f_X$ (not $w_X$). As described in Section 2.3 above, the settling velocities for POC and bSi are not optimized and are instead fixed to match the values of their respective parent offline models. While there are no parent models for dust and precipitation, we do not optimize

the corresponding settling velocities for these particle types either, because $K_X$ and $w_X$ perfectly compensate each other. In fact, only their product, $K_X\,w_X$, which sets strength of the "scavenging pump" through the operator matrix $\mathbf{T}_{\mathrm{scav}}$, appears in the tracer equations (see Eq. (16) or, e.g., John et al., 2020).

### 3.2  Fit to observations

The general fit to observations is illustrated in Fig. 7, which shows the percentiles of the cumulative joint probability distribution

of the modelled and observed Nd concentrations and $\varepsilon_{\mathrm{Nd}}$ values. Despite the slightly visible spread, most of the modelled–observed [Nd] and $\varepsilon_{\mathrm{Nd}}$ values lie close to the 1:1 line, indicating a good match.

While statistics such as Fig. 7 provide important information at a quick glance, they do not retain any geographical information, so that a more detailed investigation is required to fully assess the model's skill. Indeed, the deviations shown by [Nd] and $\varepsilon_{\mathrm{Nd}}$ clusters slightly off the 1:1 line (Fig. 7) likely reflect groups of geographically proximate data points that may be

symptomatic of systematic biases, which must be analyzed in further detail.

We explore the regional variations of the model's skill with depth in Figure 8, which shows the basin-averaged profiles of modelled and observed [Nd] and $\varepsilon_{\mathrm{Nd}}$ for the Atlantic, Pacific, Indian, and Southern Oceans. Simulated [Nd] fits the "nutrient-like" profiles of basin-mean observations and captures the bulk of inter-basin variance fairly well despite systematic biases of about $-3\,\mathrm{pM}$ in the mid-depth Atlantic, $+3\,\mathrm{pM}$ in the deep Atlantic, up to $+6\,\mathrm{pM}$ in the mid-depth Indian Ocean, and $-6\,\mathrm{pM}$

in the $4,250\,\mathrm{m}$-deep Southern Ocean, (Fig. 8a–d). Similarly, for $\varepsilon_{\mathrm{Nd}}$ values, we find an overestimate of about $+1\,\%_{00}$ below




**Table 2.** List of parameters. Realistic parameter ranges were arbitrarily prescribed based on the literature and the expertise of the authors. Final values have been rounded to 3 significant digits. Parameters without a range are not optimized (final value equals initial guess).

| Symbol | Optimized value | Initial guess | Range | Unit | Description |
|---|---|---|---|---|---|
| $\alpha_a$ | 6.79 | 1 | $(0,20)$ | | Curvature of Nd release enhancement parabola |
| $\alpha_c$ | $-12.7$ | $-10$ | $(-20,0)$ | ‰ | Center of Nd release enhancement parabola |
| $\alpha_{GRL}$ | 1.57 | 2 | $(0,\infty)$ | | Geenland Nd release enhancement |
| $\sigma_\varepsilon$ | 0.379 | 0.5 | $(0,5)$ | ‰ | Per-pixel variance (std) of $\varepsilon_{Nd}$ |
| $c_{river}$ | 376 | 100 | $(0,\infty)$ | pM | River effective [Nd] |
| $c_{gw}$ | 109 | 100 | $(0,\infty)$ | pM | Surface groundwater effective [Nd] |
| $\sigma_{hydro}$ | 0.792 | 1 | $(0,\infty)$ | $\mathrm{Mmol\,yr^{-1}}$ | Hydrothermal source magnitude |
| $\varepsilon_{hydro}$ | 10.9 | 10 | $(-10,15)$ | ‰ | Hydrothermal source $\varepsilon_{Nd}$ |
| $\phi_0$ | 83.7 | 20 | $(0,\infty)$ | $\mathrm{pmol\,cm^{-2}\,yr^{-1}}$ | Sedimentary flux at surface |
| $\phi_\infty$ | 1.11 | 10 | $(0,\infty)$ | $\mathrm{pmol\,cm^{-2}\,yr^{-1}}$ | Sedimentary flux at infinite depth |
| $z_0$ | 170 | 200 | $(0,\infty)$ | m | Sedimentary flux depth attenuation |
| $\varepsilon_{EAsia}$ | $-7.6$ | $-8$ | $(-12,-2)$ | ‰ | EAsia dust $\varepsilon_{Nd}$ |
| $\varepsilon_{NEAf}$ | $-13.7$ | $-12$ | $(-15,-9)$ | ‰ | NEAf dust $\varepsilon_{Nd}$ |
| $\varepsilon_{NWAf}$ | $-12.3$ | $-12$ | $(-15,-9)$ | ‰ | NWAf dust $\varepsilon_{Nd}$ |
| $\varepsilon_{NAm}$ | $-4.25$ | $-8$ | $(-12,-4)$ | ‰ | NAm dust $\varepsilon_{Nd}$ |
| $\varepsilon_{SAf}$ | $-21.6$ | $-10$ | $(-25,-6)$ | ‰ | SAf dust $\varepsilon_{Nd}$ |
| $\varepsilon_{SAm}$ | $-3.15$ | $-3$ | $(-7,0)$ | ‰ | SAm dust $\varepsilon_{Nd}$ |
| $\varepsilon_{MECA}$ | 0.119 | $-2$ | $(-5,3)$ | ‰ | MECA dust $\varepsilon_{Nd}$ |
| $\varepsilon_{Aus}$ | $-4.03$ | $-4$ | $(-7,-1)$ | ‰ | Aus dust $\varepsilon_{Nd}$ |
| $\varepsilon_{Sahel}$ | $-11.9$ | $-12$ | $(-15,-9)$ | ‰ | Sahel dust $\varepsilon_{Nd}$ |
| $\beta_{EAsia}$ | 23 | 5 | $(0,100)$ | % | EAsia dust Nd solubility |
| $\beta_{NEAf}$ | 23.3 | 5 | $(0,100)$ | % | NEAf dust Nd solubility |
| $\beta_{NWAf}$ | 3.17 | 5 | $(0,100)$ | % | NWAf dust Nd solubility |
| $\beta_{NAm}$ | 82.8 | 5 | $(0,100)$ | % | NAm dust Nd solubility |
| $\beta_{SAf}$ | 38.5 | 5 | $(0,100)$ | % | SAf dust Nd solubility |
| $\beta_{SAm}$ | 2.52 | 5 | $(0,100)$ | % | SAm dust Nd solubility |
| $\beta_{MECA}$ | 14.7 | 5 | $(0,100)$ | % | MECA dust Nd solubility |
| $\beta_{Aus}$ | 11.6 | 5 | $(0,100)$ | % | Aus dust Nd solubility |
| $\beta_{Sahel}$ | 2.95 | 5 | $(0,100)$ | % | Sahel dust Nd solubility |
| $\varepsilon_{volc}$ | 13.1 | 10 | $(0,15)$ | ‰ | Volcanic ash $\varepsilon_{Nd}$ |
| $\beta_{volc}$ | 76 | 10 | $(0,100)$ | % | Volcanic ash Nd solubility |
| $K_{prec}$ | 0.00576 | 0.01 | $(0,\infty)$ | | Precipitation reaction constant |
| $f_{prec}$ | 0.124 | 0.4 | $(0,1)$ | | Fraction of non-buried precipitated Nd |
| $w_{prec}$ | 0.7 | 0.7 | | $\mathrm{km\,yr^{-1}}$ | Settling velocity of precipitated Nd |
| $K_{POC}$ | $7.83 \times 10^{13}$ | $3 \times 10^{13}$ | $(0,\infty)$ | | POC-scavenging reaction constant |
| $f_{POC}$ | 0.312 | 0.78 | $(0,1)$ | | Fraction of non-buried POC-scavenged Nd |
| $w_{POC}$ | 40 | 40 | | $\mathrm{m\,d^{-1}}$ | Settling velocity of POC-scavenged Nd |
| $K_{bSi}$ | $3.4 \times 10^{12}$ | $3 \times 10^{13}$ | $(0,\infty)$ | | bSi-scavenging reaction constant |
| $f_{bSi}$ | 0.784 | 0.5 | $(0,1)$ | | Fraction of non-buried bSi-scavenged Nd |
| $w_{bSi}$ | 714 | 714 | | $\mathrm{m\,d^{-1}}$ | Settling velocity of bSi-scavenged Nd |
| $K_{dust}$ | $2 \times 10^{15}$ | $2 \times 10^{15}$ | $(0,\infty)$ | | Dust-scavenging reaction constant |
| $f_{dust}$ | 0.0861 | 0.073 | $(0,1)$ | | Fraction of non-buried dust-scavenged Nd |
| $w_{dust}$ | 1 | 1 | | $\mathrm{km\,yr^{-1}}$ | Settling velocity of dust-scavenged Nd |





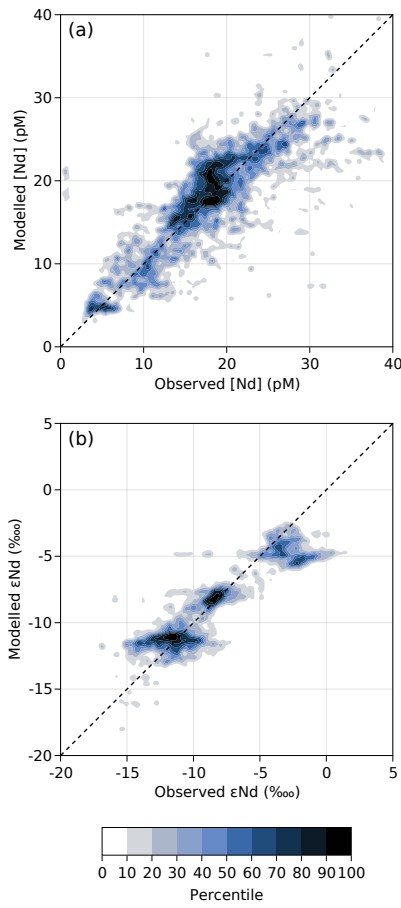

**Figure 7.** Quantiles of the cumulative joint probability density functions of modelled and observed (a) Nd concentrations and (b) $\varepsilon_{\mathrm{Nd}}$ values. Darker colors indicate high density of data, such that $n\,\%$ of the modelled and observed data lie outside of the $n$-th percentile contour. The closer the darker contours are to the 1:1 black dashed line the better the fit.

700 m in the Atlantic, and an underestimate of up to $-2\,\%_{00}$ in the Pacific, particularly near the surface, while the modelled and observed basin-averaged Southern-Ocean profiles are a tight fit (Fig. 8e–h).

We further assess the model skill by looking at GEOTRACES transects individually. Out of the 3483 observations of [Nd] that we use to constrain our model, 1575 ($\sim$45 %) come from the IDP17 Schlitzer et al. (2018), and were collected along the
GA02, GA03, GA10, GA11, GAc01, GIPY04, GIPY05, GIPY06, GPc02, GPpr04, and GPpr05 cruises (Fig. 9b–l). Similarly, out of the 2988 $\varepsilon_{\mathrm{Nd}}$ observations, 790 ($\sim$26 %) come from IDP17 cruises GA02, GA03, GA11, GIPY04, GIPY05, GIPY06, GPc02, GPpr04, and GPpr05 (Fig. 10b–j).

Figure 9 reveals in detail how well GNOM output matches observational [Nd] data. The model captures the broad inter-basin and intra-basin variations with high fidelity despite some slight mismatches. Specifically, Fig. 9c reveals overestimates



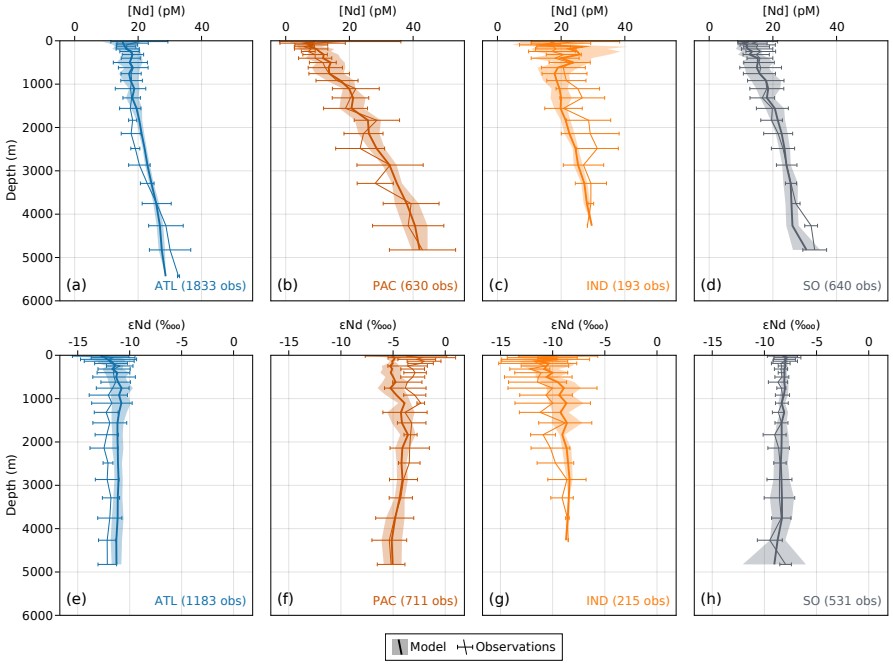

**Figure 8.** Basin-averaged profiles of (a–d) Nd concentrations and (e–f) $\varepsilon_{Nd}$ values versus depth. The basins (Atlantic, Pacific, Indian, and Southern Ocean) with the number of observations for each tracer are reported in the bottom right corner of each panel. The mean and standard deviation of observations are calculated at each vertical grid level of the OCIM v2.0 grid and represented by the thin line and error bars. The mean and standard deviation of the model are represented by the thick line and lighter-colored ribbon.

of mid-depth and deep [Nd] in the west Pacific (cruise GPpr04) with an overestimated gradient with depth, potentially due to too large a sedimentary source or too strong scavenging. Mid-depth overestimates of [Nd] also appear in the Atlantic (GA03, GAc01, GA10; Fig. 9h, j, k). However, the deepest [Nd] are underestimated in association with a generally underestimated vertical gradient, particularly along GA10. Hence, the [Nd] mismatches in the Pacific are suggestive of either too weak a deep sedimentary source or too efficient scavenging and burial in the deep. These systematically opposed mismatches between the

Atlantic and Pacific are likely due to the optimization procedure, which balances out all the mismatches simultaneously. Future improvements of the GNOM that resolve these discrepancies could include different parameterizations of the sedimentary source and scavenging or the addition of a currently absent third mechanism, such as a nepheloid source or sink.

Figure 10 shows modelled and observed $\varepsilon_{Nd}$ values along IDP17 cruise transects. While the observed inter-basin variability is adequately represented, the GNOM does not perfectly capture the finer spatial details of observed $\varepsilon_{Nd}$, suggesting that there

is still room for model improvement. The model agrees well with observations along Southern Ocean transects (Fig. 10e, i, j; GPc02, GIPY04, GIPY05). However, in the North Atlantic (e.g., Fig. 10f, GA02), the GNOM does not entirely capture a strong negative $\varepsilon_{Nd}$ plume along the North Atlantic Deep Water (NADW). Conversely, in the West Pacific (Fig. 10c; GPpr04), our model misses strongly positive surface $\varepsilon_{Nd}$ observations and instead displays a deep plume of positive $\varepsilon_{Nd}$ that is absent



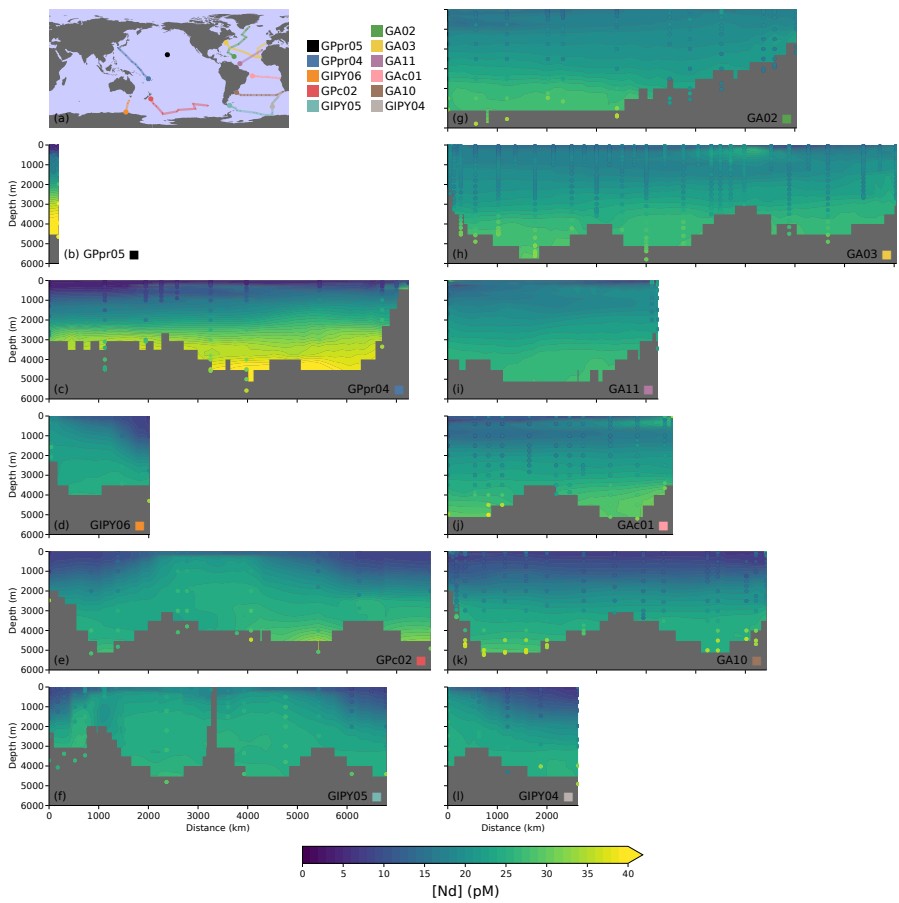

**Figure 9.** (a) GEOTRACES cruise tracks with [Nd] observations. The legend layout matches the layout of the other subplots of the figure. (b)–(l) Modelled and observed [Nd] along GEOTRACES transects. Modelled values are shown as filled contours while observed values are overlaid as a scatter plot.

from the observational data. This is potentially due to missing mechanisms, sources or sinks, the correct implementation of
which would likely benefit from more observational $\varepsilon_{Nd}$ data in the Pacific and Indian basins.

Figure 11 shows the model and observed [Nd] and $\varepsilon_{Nd}$ for all observations averaged over different depth ranges. Contrary to Figs. 9 and 10, this includes all observations used to constrain the GNOM v1.0 (i.e., not just IDP17). As expected, the model broadly matches the observational data well, with some systematic mismatches in different locations. Figure 11d–e reveal an underestimate of deep [Nd] in the Northern Indian Ocean in the Bay of Bengal, which is likely attributable to too strong
scavenging or too weak sedimentary fluxes into the deeper layers of the model. Fig. 11c shows elevated Nd concentrations in the deepest parts of the Baffin Bay, potentially due to too large sources, lack of data, or even circulation issues related to the resolution of the model grid in that region. Notably, Fig. 11c–e reveals discrepancies among observations, with a few [Nd] values near the GA02 transect that stand out compared to neighboring observations. Figures 11f–i show a substantial

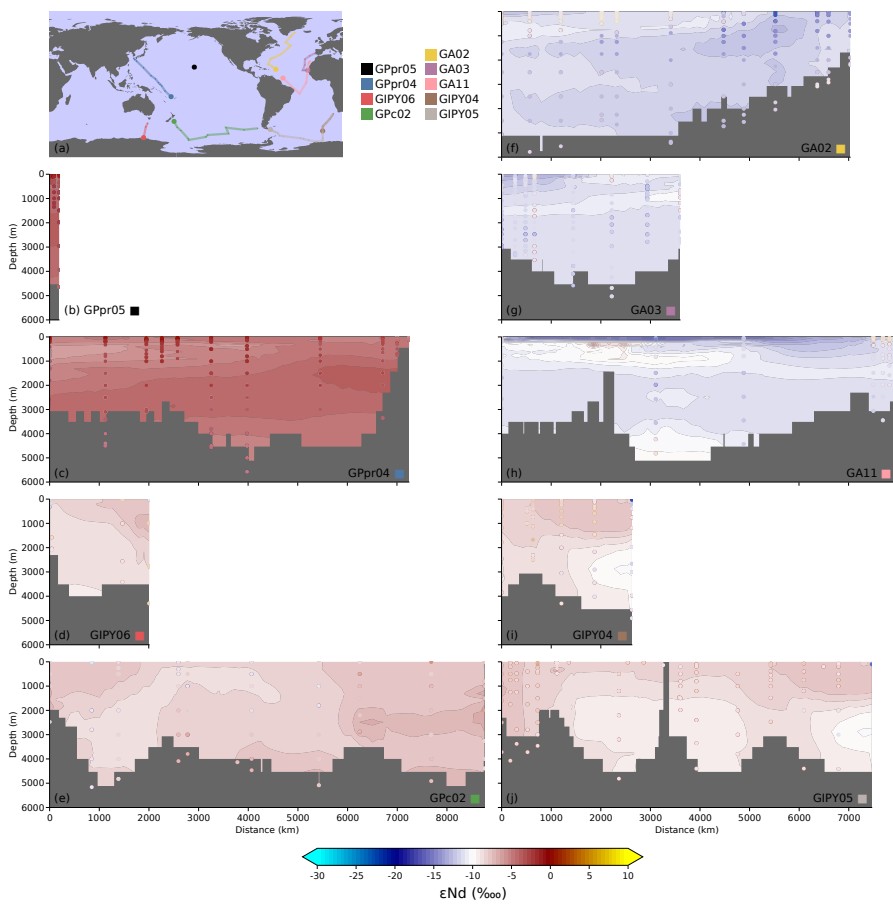

**Figure 10.** (a) GEOTRACES cruise tracks with $\varepsilon_{Nd}$ observations. The legend layout matches the layout of the other subplots of the figure. (b)–(j) Modelled and observed $\varepsilon_{Nd}$ along GEOTRACES transects. Modelled values are shown as filled contours while observed values are overlaid as a scatter plot.

underestimate of equatorial Pacific $\varepsilon_{Nd}$ values above $1500\,\mathrm{m}$ depth, from east Indonesia to the west coast of Ecuador and Peru.
Strongly negative observed $\varepsilon_{Nd}$ values in the North West Atlantic and near the west coast of Africa from the Congo to Namibia are not well captured by the model. Conversely, rather positive $\varepsilon_{Nd}$ values at the surface going north along the western coast of North Africa also seem to evade the capability of GNOM to represent observations. The lack of resolution or some important missing mechanisms are likely the cause of these larger mismatches. In future versions of GNOM, we intend to improve the model by targeting these regions of particularly pronounced misfits.

**3.3 Diagnostics**

One of the biggest advances in the GNOM v1.0, compared to earlier models of the marine Nd cycle, is due to the steady-state matrix formulation of the model which allows to compute advanced and detailed diagnostics that can directly address





**Figure 11.** Model (heatmaps) and observations (markers) for (a–f) Nd concentrations and (g–k) $\varepsilon_{Nd}$ values. Model values are averaged over the indicated depth range. Individual observed values are overlaid on top of the modelled heatmap and on top of each other, so that perfect model–observation matches and deeper observations can be hard to see or hidden.



**Table 3.** Source magnitudes, their Nd contributions, and corresponding bulk residence time. Values reported with two significant digits.

| Symbol | Source type | Global magnitude | | Nd contribution | | Residence time |
|--------|-------------|:---:|:---:|:---:|:---:|:---:|
| | | $\mathrm{Mmol\,yr^{-1}}$ | % | pM | % | yr |
| $\sigma_{\mathrm{dust}}$ | Mineral dust | 24 | 36 | 9.8 | 44 | 540 |
| $\sigma_{\mathrm{volc}}$ | Volcanic ash | 0.23 | 0.35 | 0.12 | 0.52 | 650 |
| $\sigma_{\mathrm{sed}}$ | Sedimentary flux | 32 | 48 | 8.8 | 39 | 370 |
| $\sigma_{\mathrm{river}}$ | Riverine discharge | 9.4 | 14 | 3.2 | 14 | 450 |
| $\sigma_{\mathrm{gw}}$ | Groundwater discharge | 0.024 | 0.037 | 0.01 | 0.045 | 540 |
| $\sigma_{\mathrm{hydro}}$ | Hydrothermal vents | 0.79 | 1.2 | 0.33 | 1.5 | 550 |
| $\sigma_{\mathrm{tot}}$ | Total | 66 | 100 | 22 | 100 | 440 |

fundamental questions about the distribution of tracers and better understand their cycle. In the following sections we showcase a few such diagnostics.

### 3.3.1 Source magnitudes

The optimized parameters determine the magnitude of the sources, which are collected in Table 3. In our best estimate, about 66 Mmol of Nd (or about 9,500 metric tons) are injected into the global ocean every year. This falls slightly above the 38–57 $\mathrm{Mmol\,yr^{-1}}$ range of previous GCM models (Table 1). The aeolian dust and sedimentary sources are the dominant ones contributing 24 $\mathrm{Mmol\,yr^{-1}}$ and 32 $\mathrm{Mmol\,yr^{-1}}$ (about 35 % and 50 % of to total source), respectively. While this falls within the 0.69–60 $\mathrm{Mmol\,yr^{-1}}$ range for global aeolian source magnitudes of previous modelling studies, our aeolian sources are an order of magnitude larger than previous GCM-based modelling studies (0.69–3.5 $\mathrm{Mmol\,yr^{-1}}$). This is likely due to our optimization procedure, during which Nd solubility is allowed to be adjusted through the whole 0–100 % range, compared to previous GCM-based studies that typically use a fixed 2 % solubility (Arsouze et al., 2009; Gu et al., 2019; Pöppelmeier et al., 2020). This is despite the initial guesses for $\beta_r$ values for dust set at 5 %, which penalizes large solubilities more than low solubilities (see Fig. B1). We note that worse general fit to the observations have been achieved with some of our optimization runs ending in a distinct local minima with significantly smaller dust solubilities (not shown). At 32 $\mathrm{Mmol\,yr^{-1}}$, the GNOM sedimentary source falls right within the 0–78 $\mathrm{Mmol\,yr^{-1}}$ range of previous models (28–55 $\mathrm{Mmol\,yr^{-1}}$ for GCM-based studies). The third largest source is riverine, with about 10 $\mathrm{Mmol\,yr^{-1}}$, also in accord with the published 1.8–12.4 $\mathrm{Mmol\,yr^{-1}}$ range in previous models. Hence, apart from a relatively elevated dust source, the GNOM optimization objectively supports previous estimates of the magnitude of these sources.

### 3.3.2 Partition according to source type

We partition Nd concentration according to source type simply by removing all the other sources. (The superposition principle applies directly to our model because it is linear in Nd; see, e.g., Holzer et al., 2016, who partitioned dissolved iron concentrations, first requiring the construction of a linear equivalent model.). With $\mathrm{Nd}_k$ denoting neodymium that was injected by source





type $k$, its corresponding column vector is thus computed by solving

$$\mathbf{H}\boldsymbol{\chi}_{\mathrm{Nd}_k}^{\mathrm{mod}} = \boldsymbol{s}_k, \tag{19}$$

where $\mathbf{H} = \mathbf{T}_{\mathrm{circ}} + \mathbf{T}_{\mathrm{scav}}$. Taking the global volume-weighted mean of each $\boldsymbol{\chi}_{\mathrm{Nd}_k}^{\mathrm{mod}}$ gives the contribution of each source type to the total Nd inventory and are collected in Table 3.

As is the case for most global biogeochemical cycles, the relative source magnitudes and their relative contribution to the standing stock do not necessarily match. For instance, mineral dust, volcanic ash, and hydrothermal vents contribute more to the mean Nd concentration than their relative source magnitudes. These variations can be directly linked to the bulk residence time of Nd molecules, which vary with location of injection and consequently with source type.

### 3.3.3 Bulk residence times

The bulk residence time of $\mathrm{Nd}_k$ is given by taking the ratio of its inventory to its source magnitude. Total Nd (i.e., from all sources) has a bulk residence time of roughly $440\,\mathrm{yr}$, which is within the $350$ to $2900\,\mathrm{yr}$ range of previous Nd-cycling models. (The bulk residence time for GNOM falls slightly below the $560$ to $750\,\mathrm{yr}$ range of GCM-based models; see Table 1).

Sedimentary-sourced Nd, which is injected just above the seafloor where it is likely to be buried quickly, has the shortest residence time at $370\,\mathrm{yr}$. In comparison, volcanic-ash Nd, most of which is deposited onto the surface of the Pacific, remains on average $650\,\mathrm{yr}$ in the system (i.e., about $75\,\%$ longer than sedimentary-sourced Nd). Mineral-dust deposited Nd as well as riverine and surface groundwater Nd all show a residence time of about $550\,\mathrm{yr}$.

### 3.3.4 Nd and $\varepsilon_{\mathrm{Nd}}$ conservativeness

Sediment-core records of $\varepsilon_{\mathrm{Nd}}$ are of considerable importance for paleoceanography because they serve as a fingerprint of past ocean circulation and have been used, in particular, to infer changes in the Atlantic meridional overturning circulation (AMOC) (e.g., Palmer and Elderfield, 1985; Rutberg et al., 2000; Piotrowski et al., 2004, 2005; van de Flierdt et al., 2006; Piotrowski et al., 2008; Pena et al., 2013; Pena and Goldstein, 2014; Huang et al., 2014; Kim et al., 2021; Pöppelmeier et al., 2021; Hines et al., in revision). Observations of modern ocean $\varepsilon_{\mathrm{Nd}}$ have thus been extensively used to estimate Atlantic water mass fractions, typically inferred from North–South end-members and assuming quasi-conservative transport and mixing of $\varepsilon_{\mathrm{Nd}}$ as an ocean circulation tracer (Hartman, 2015; Wu, 2019; Wu et al., submitted).

Our GNOM model – or, more precisely, the steady-state matrix schema with which it is built – allows for *exact* computations of these water-masses and the contributions of various sources and regions to the modern-ocean Nd and $\varepsilon_{\mathrm{Nd}}$ distributions. We emphasize that by "exact", here, we do not mean exact for the real ocean, but instead for our given choice of ocean-circulation model (in this case the OCIM v2.0, DeVries and Holzer, 2019), which is arguably the best ocean-circulation model available for such climatological estimates (see, e.g. John et al., 2020). Here, we merely showcase these diagnostics but we plan to further explore the underlying scientific questions in future work.



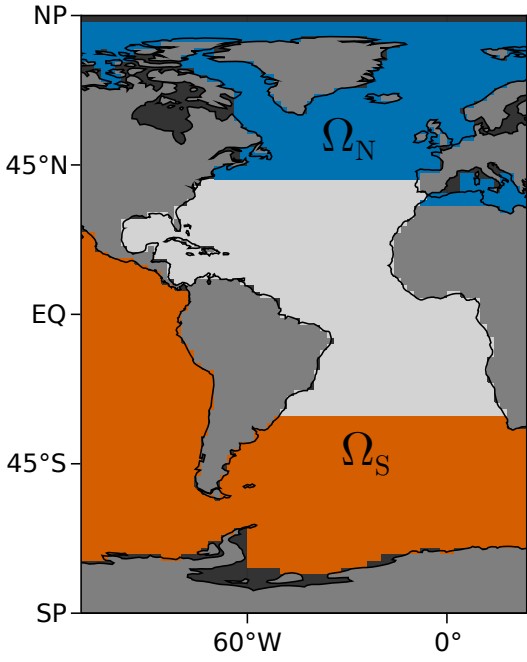

**Figure 12.** Northern Atlantic ($\Omega_N$; blue) and southern Atlantic ($\Omega_S$; orange) regions used for the $\varepsilon_{Nd}$-conservativeness and the water-tagged Nd diagnostics within the central Atlantic region (from $30°$S to $40°$N; light gray).

We chose two simple regions that cover the entire ocean except for the central Atlantic between $30°$S and $40°$N. We denote these regions by $\Omega_N$ and $\Omega_S$ such that $\Omega_N$ borders the northern Atlantic and $\Omega_S$ the southern Atlantic. The regions are shown in Fig. 12.

     Firstly, we track Nd concentrations from each of these regions. Neodymium concentrations are *not* conservative, in part due to reversible scavenging and in part due to external sources that inject new Nd along transport pathways. For example, we can

track Nd that came from region $\Omega_N$ by "tagging" Nd that comes into contact with $\Omega_N$ and removing that tag when Nd enters $\Omega_S$. Mathematically, we can perform this tagging/untagging by simulating a fictitious tracer, denoted $Nd_{N\text{-tag}}$, for which we enforce a concentration equal to simulated [Nd] in region $\Omega_N$, a concentration equal to zero in $\Omega_S$, and allowing reversible scavenging and burial to remove $Nd_{N\text{-tag}}$ in between. In practice, we compute the corresponding column vector $\chi_{Nd_{N\text{-tag}}}$ by solving

$$\left(\mathbf{H} + \mathbf{M}_N + \mathbf{M}_S\right)\chi_{Nd_{N\text{-tag}}} = \mathbf{M}_N\,\chi_{Nd}^{mod} \tag{20}$$

where the $\mathbf{M}_i$ are diagonal matrices of which the diagonals are $1\,\mathrm{s}^{-1}$ for indices (i.e., coordinates) within region $\Omega_i$ and $0\,\mathrm{s}^{-1}$ otherwise. Because of the very short timescale of $1\,\mathrm{s}$ employed, Eq. (20) effectively enforces that $[Nd_{N\text{-tag}}] = 0\,\mathrm{pM}$ in region $\Omega_S$ and that $[Nd_{N\text{-tag}}] = [Nd]$ in region $\Omega_N$ (where [Nd] denotes the simulated concentration from the GNOM). We track $Nd_{S\text{-tag}}$ in the same way by replacing $\mathbf{M}_N$ with $\mathbf{M}_S$ in the right-hand side of Eq. (20). The neodymium concentration that came from neither $\Omega_N$ or $\Omega_S$ is then simply given by $[Nd] - [Nd_{N\text{-tag}}] - [Nd_{S\text{-tag}}]$.





Figure 13 shows the Atlantic zonal averages of $[\mathrm{Nd_{N\text{-}tag}}]$, $[\mathrm{Nd_{S\text{-}tag}}]$, and "non-tagged" $[\mathrm{Nd}]$ (i.e., Nd that was injected in the central Atlantic between $30°\mathrm{S}$ and $40°\mathrm{N}$). In the $\Omega_\mathrm{N}$ region (north of $40°\mathrm{N}$), $100\,\%$ of Nd is tagged as $\mathrm{Nd_{N\text{-}tag}}$ (Fig. 13a). Similarly, $100\,\%$ of Nd is $\mathrm{Nd_{S\text{-}tag}}$ in $\Omega_\mathrm{S}$ (Fig. 13b). Figure 13c shows the remaining non-tagged Nd and Fig. 13d combines panels (a–c) by showing only the dominant fraction. A clear signal of the influence of surface sources is visible down to about $1500\,\mathrm{m}$, confining the dominance of the N- and S-tagged fractions of Nd to deep waters and to high latitudes, close to their

respective tagging regions. To the best of our knowledge, such partitions of neodymium have never been performed in previous modelling studies. In the future, we intend to further explore the extent of the influence of high latitudes on the distribution of Nd.

We now track $\varepsilon_{\mathrm{Nd}}$ as if it were a conservative tracer, from the $\Omega_\mathrm{N}$ and $\Omega_\mathrm{S}$ regions. Technically, this is done by tracking water itself as it leaves either region. Water-mass fractions estimated with this approach can be used to directly "propagate" modelled

$\varepsilon_{\mathrm{Nd}}$ values from the boundaries of the North and South Atlantic regions to provide an exact end-member mixing estimate that serves as a reference.

Water-mass fractions have been estimated using a Green-function boundary propagator in similar model contexts (e.g., Holzer and Hall, 2008; Primeau, 2005; Holzer and Primeau, 2008). They can be used to calculate the conservative transport of any tracer. As an illustrative example, here, we propagate modelled $\varepsilon_{\mathrm{Nd}}$ simultaneously from both the $\Omega_\mathrm{N}$ and $\Omega_\mathrm{S}$ regions into

the central Atlantic. This theoretical conservative value is denoted by $\varepsilon_{\mathrm{Nd}}^{\Omega}$ (with $\Omega$ as a superscript to denote that its value is entirely determined by the $\varepsilon_{\mathrm{Nd}}$ inside the $\Omega_\mathrm{N}$ and $\Omega_\mathrm{S}$ regions) and the corresponding column vector $\boldsymbol{\varepsilon}_{\mathrm{Nd}}^{\Omega}$ is computed by solving

$$(\mathbf{T}_\mathrm{circ} + \mathbf{M}_\mathrm{N} + \mathbf{M}_\mathrm{S})\,\boldsymbol{\varepsilon}_{\mathrm{Nd}}^{\Omega} = (\mathbf{M}_\mathrm{N} + \mathbf{M}_\mathrm{S})\,\boldsymbol{\varepsilon}_{\mathrm{Nd}}^{\mathrm{mod}}. \tag{21}$$

Like Eq. (20), Eq. (21) effectively enforces that $\boldsymbol{\varepsilon}_{\mathrm{Nd}}^{\Omega}$ matches modelled $\varepsilon_{\mathrm{Nd}}$ inside $\Omega_\mathrm{N}$ and $\Omega_\mathrm{S}$ but is conservatively propagated

by the advective–diffusive transport operator $\mathbf{T}_\mathrm{circ}$ outside of $\Omega$. (Note the different operators on both the left-hand side and right-hand sides between Eqs. (20) and (21).) Atlantic zonal averages of modelled $\varepsilon_{\mathrm{Nd}}$, conservatively-propagated $\varepsilon_{\mathrm{Nd}}$, and their difference ($\Delta(\varepsilon_{\mathrm{Nd}})$) are shown in Fig. 14. Figure 14c, in particular, shows where $\varepsilon_{\mathrm{Nd}}$ behaves conservatively (white) and where it does not (green or pink), in our model. In accord with Fig. 13, $\varepsilon_{\mathrm{Nd}}$ is the least conservative close to the surface, where most of the Nd was never in contact with $\Omega_\mathrm{S}$ or $\Omega_\mathrm{N}$ and was instead injected in the mid-latitude Atlantic. This surface overestimate

is most pronounced away from $\Omega_\mathrm{N}$ and $\Omega_\mathrm{S}$ with $\Delta(\varepsilon_{\mathrm{Nd}})$ values of up to $+10\,\permil$. Going from 200 to $1000\,\mathrm{m}$ depth, we find $\Delta(\varepsilon_{\mathrm{Nd}})$ values decreasing from $+5$ to $+1\,\permil$. Conservative $\varepsilon_{\mathrm{Nd}}$ and true $\varepsilon_{\mathrm{Nd}}$ remain within $1\,\permil$ of each other below $1000\,\mathrm{m}$ depth, despite a slight $\Delta(\varepsilon_{\mathrm{Nd}})$ overestimate near the seafloor (likely due to the effect of local sedimentary flux) and a slight underestimate around $1,500\,\mathrm{m}$ (potentially due to reversible scavenging). We intend to investigate the distinct conservativeness of Nd and $\varepsilon_{\mathrm{Nd}}$ (the "neodymium paradox") further in future work.

## 4 Conclusions

The most prominent caveat of GNOM v1.0 is the steady-state assumption, which we apply to both the circulation and the Nd cycle. Hence, by construction, daily, seasonal, decadal, or multi-decadal fluctuations that deviate from the climatological mean



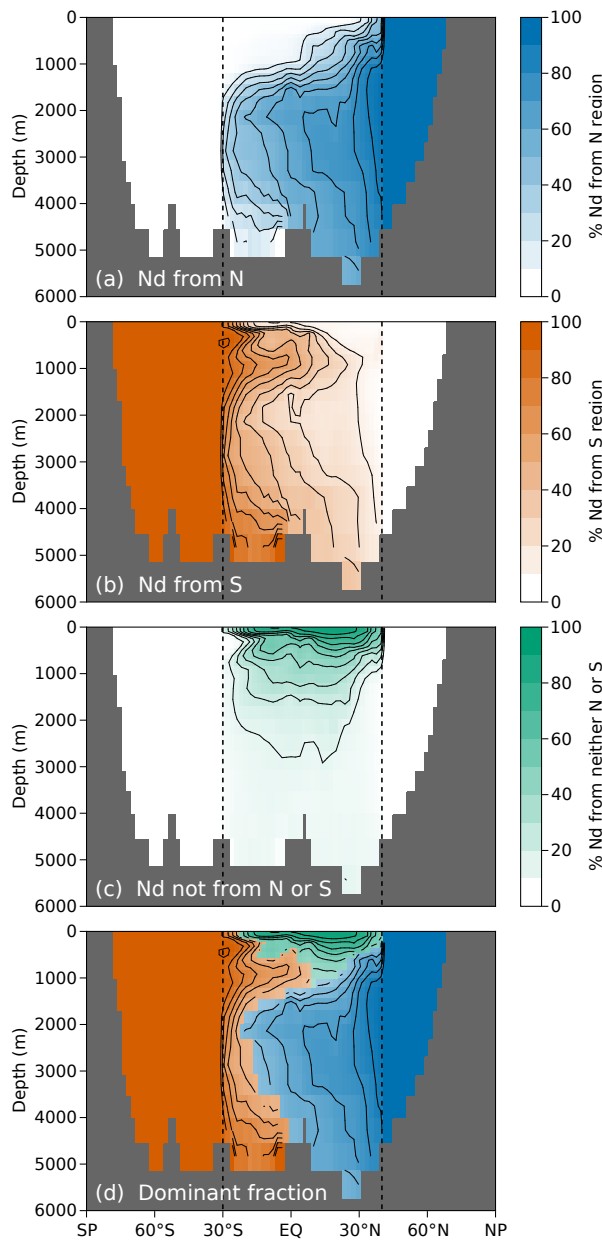

**Figure 13.** (a) Atlantic zonal average of the fraction of Nd tagged in the $\Omega_N$ region (north of 40°N; blue). (b) Same for Nd tagged in the $\Omega_S$ region (south of 30°S; orange). (c) Same for Nd tagged injected within the mid-latitude Atlantic (between 30°S and 40°N; green). (d) Atlantic zonal average showing only the dominant fraction of Nd coming from either $\Omega_N$ or $\Omega_S$ or from within 30°S–40°N. Contour lines are shown for each 10 % increment. Tagging regions are shown in Fig. 12.





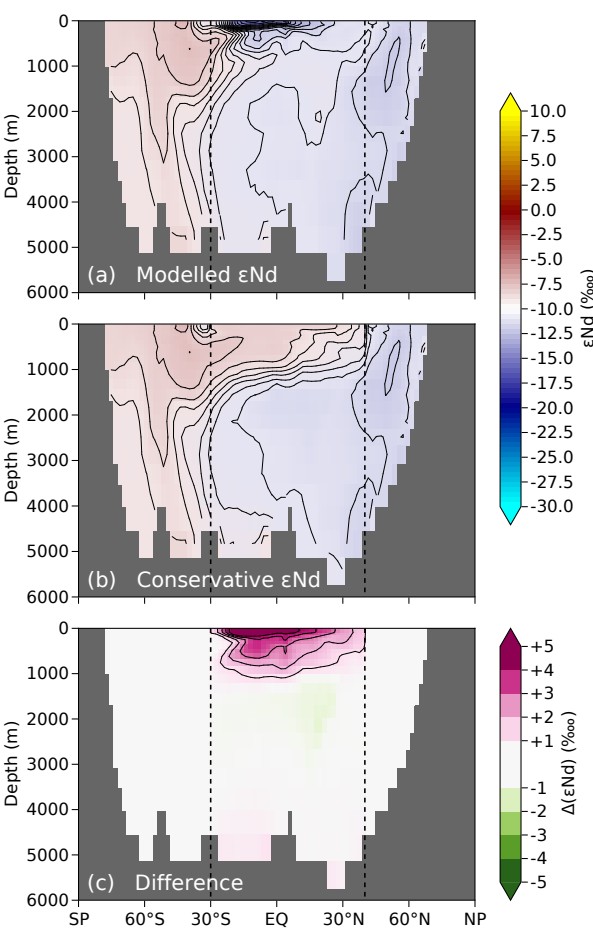

**Figure 14.** (a) Atlantic zonal average of modelled $\varepsilon_{Nd}$. (b) Same for conservatively transported $\varepsilon_{Nd}$ from regions $\Omega_S$ and $\Omega_N$. (c) Difference between (a) and (b). Contour lines are shown for each $1\,‰$ increment. (Contour lines for $\Delta(\varepsilon_{Nd}) > +5\,‰$ not shown to avoid clutter). Regions are shown in Fig. 12.



cannot be captured by our model. However, we trust that the circulation model used (the OCIM v2.0; DeVries and Holzer, 2019), which is data-assimilated with ventilation tracers, captures the predominant features and pathways of the modern ocean

circulation and provides the most realistic steady-state transport to date (e.g., DeVries and Holzer, 2019; John et al., 2020).

We note that compared to previous modelling studies, the GNOM does not represent scavenging by calcium carbonate ($CaCO_3$) because there is no publicly available particulate $CaCO_3$ field to the best of our knowledge. Modelling scavenging is a challenging task that the GNOM model does not pretend to achieve with high accuracy. However, we deem the current implementation satisfactory considering the quality of the overall model–observation fit. Future versions of the GNOM could

include $CaCO_3$-particle scavenging or a generally improved scavenging parameterization.

Our model reveals some locations of particular interest for improving our understanding of the Nd cycle and $\varepsilon_{Nd}$ patterns. For instance, there are only two GEOTRACES transects in the Pacific Ocean which cover the western Pacific and the Southern Ocean, the zonal transects in the Atlantic contain only a few $\varepsilon_{Nd}$ measurements compared to [Nd] (e.g., GA03, GA10, GA11, GAc01), and there are no published transects in the Indian Ocean, which may contribute a non negligible fraction of Nd to

both the Atlantic and Pacific. In the future, we hope that more data will be made available and improve the capabilities of data-constrained model estimates of $\varepsilon_{Nd}$ and the Nd cycle.

While our model endeavors to use formulations and parameter constraints which have reasonable biogeochemical interpretations, there is always room for improvement. For example, our sedimentary source parameterization, which we plan to investigate further in future work, assumes an exponential profile for the sedimentary source flux. While this parameterization

is flexible enough to reproduce most of the qualitative features of the sedimentary fluxes in previous models, one might argue that a more mechanistic model of the sedimentary budget would result in a more realistic overall Nd-cycling model.

Despite the theoretical advantages they confer to the convergence rate of our optimization procedure, our specific choice of prior distributions for the parameters (Fig. B1) remains arbitrary, though the ranges chosen are of course informed by prior work. Different ranges and initial conditions would yield different optimal solutions and therefore different parameter

values. Our choice of a local Newton-Trust region optimizer also exposes our strategy to the risk of getting stuck in local minima. To counter that risk, we have opted for the traditional ad-hoc counter mesaure of running the optimization from multiple, randomized initial guesses. While this offers the advantage of spanning a large number of cases, it offers no guarantees as to the globality of the optimal set of parameters. It is thus entirely likely that a "better" set of parameter values would reduce the objective-function value and improve the model skill, though again we note that the most important features of Nd

biogeochemical cycling are likely to converge to similar values regardless of initial conditions.

Our specific choice of objective function gives a measure of the skill of the model for reproducing [Nd] and $\varepsilon_{Nd}$ observations. Roughly speaking, despite our arbitrary choices for the weights $\omega_{Nd}$, $\omega_{\varepsilon_{Nd}}$, and $\omega_{\boldsymbol{p}}$ involved in Eq. (17), the value of the objective function can also be interpreted as the negative log-likelihood of observing the measured $\varepsilon_{Nd}$ and [Nd] given the model and its parameters. In the future, we plan to determine the values of hyperparameters such as $\omega_{Nd}$, $\omega_{\varepsilon_{Nd}}$, and $\omega_{\boldsymbol{p}}$ through more formal

Bayesian approach.

Qualitatively, GNOM compares well to previous models that are embedded in ocean general circulation models and simulate two explicit Nd isotopes (Arsouze et al., 2009; Rempfer et al., 2011; Gu et al., 2019; Pöppelmeier et al., 2020). Given that



these models were built with a different objective, less available data, and without a systematic optimization of all parameters, we expect GNOM v1.0 to perform better against the objective function used in this study. However, we emphasize that these previous models are more suited to specific experiments, including simulations of transient changes in circulation on millennial timescales. In other words, these previous models are not all restricted by the caveats of a coarse steady-state circulation. We also note that the underlying circulation used in GNOM can be swapped with any other circulation available through the AIBECS.jl framework (see Pasquier et al., under review), although we reiterate that the OCIM v2.0 likely offers the best available representation of the current ocean circulation.

The main advantages of the GNOM are skill and computational efficiency. The GNOM v1.0 owes its low computational cost and quick simulation time to the steady-state OCIM v2.0 circulation in which it is embedded and the linear representation that allows us to solve the system of tracer equations in a single matrix inversion. The model's skill comes from the optimization procedure and likely benefits from the quality of the OCIM v2.0 circulation. The GNOM is also versatile in many respects owing to its simplicity. Parameter values can be tuned, entire mechanisms can be turned off or added with a few changes of simple lines of code. This versatility is compounded by computational speed, which makes GNOM ideally-suited for quick experimentation and further optimization. The GNOM model is also easily diagnosed, owing to the powerful tools of linear algebra. Novel diagnostics offer new insights by revealing features often hidden in standard model output. Finally, as it is available in a self-contained package (except for the GEOTRACES dataset, which is not programmatically accessible), the GNOM v1.0 offers unprecedented reproducibility, which is sorely lacking in advanced research (Peng, 2011; Irving, 2016). Thanks to the advantages listed above, the GNOM is well positioned for answering long-lasting questions and exploring new ideas about the Nd cycle and the $\varepsilon_{\mathrm{Nd}}$ distributions.

*Code and data availability.* The GNOM model code is open source and publicly available for free. An archive of the GNOM v1.0.0 code used in this study is hosted permanently on Zenodo at https://doi.org/10.5281/zenodo.5651296 (Pasquier et al., 2021). New developments are also available directly on GitHub at https://github.com/MTEL-USC/GNOM. The code is written Julia, which is itself free and open-source (Bezanson et al., 2017). The GNOM v1 model was designed using the open-source AIBECS framework, available as a Julia package (Pasquier, 2020a; Pasquier et al., under review) at https://github.com/JuliaOcean/AIBECS.jl. All the dependencies are free and open source and are version controlled through the GNOM project manifest file. The Julia version that was used for this study is v1.6.2. Except for the GEOTRACES IDP17 data, which must be downloaded manually (see www.geotraces.org), the entirety of the data used in this study can be programmatically downloaded by the GNOM code (also explained in the GNOM documentation). That is, the OCIM v2.0 circulations by DeVries and Holzer (2019) (with original files available at https://tdevries.eri.ucsb.edu/models-and-data-products/), the two-dimensional dust deposition fields partitioned according to region of origin from Kok et al. (2021a) and Kok et al. (2021b) (original files available from Adebiyi et al., 2020), the aerosol-type partitioned dust source that includes volcanic ash by Chien et al. (2016) (original files available from http://www.geo.cornell.edu/eas/PeoplePlaces/Faculty/mahowald/dust/Chienetal2016/), the riverine discharge dataset from Dai and Trenberth (2002) (original files available from Dai, 2017), the groundwater discharge from coastal sheds dataset compilation from Luijendijk et al. (2020) (original files available from Luijendijk et al., 2019), the hydrothermal $^{3}$He mantle source from the OCIM v2.0 product of DeVries and Holzer (2019) (original files available at https://tdevries.eri.ucsb.edu/models-and-data-products/),



and the particulate organic carbon three-dimensional fields from Weber et al. (2018), are all available through the AIBECS.jl interface (Pasquier, 2020a; Pasquier et al., under review). The three-dimensional field for biogenic opal particles described in Appendix A is entirely generated by a parallel inverse model of the Si cycle embedded in the GNOM code. The World Ocean Atlas silicate data (Garcia et al., 2019) used to constrain the parallel Si-cycle can be downloaded programatically by the WorldOceanAtlasTools.jl package and is available at https://data.nodc.noaa.gov/woa/. The pre-GEOTRACES IDP17 "historical" dataset for Nd and $\varepsilon_{\mathrm{Nd}}$ from van de Flierdt et al. (2016) is available at https://figshare.com/articles/dataset/Global_Database_from_Neodymium_in_the_oceans_a_global_database_a_regional_comparison_and_implications_for_palaeoceanographic_research/3980064 and the post-IDP17 data compiled for this study is available at https://figshare.com/articles/dataset/Post-IDP17_Nd_data/15058329, both of which are also downloadable programmatically by the GNOM code. Except for the model schematic (Fig. 1) that was created with TikZ (Tantau, 2013), all the figures in this manuscript were created using the Makie.jl package (Danisch et al., 2021; Danisch and Krumbiegel, 2021). (All the plotting scripts are available in the GNOM repository at https://github.com/MTEL-USC/GNOM/ and in the GNOM v1.0.0 Zenodo archive at https://doi.org/10.5281/zenodo.5651296.)

## Appendix A: Particulate Si model

To represent scavenging by opal (particulate silica), we designed and optimize a simple Si-cycling model in parallel to our Nd-cycling model. Our Si-cycling model is a simple nutrient restoring-model embedded in the same OCIM v2.0 circulation. We emphasize that the goal here is only to generate a reasonable 3D field for particulate biogenic silica concentrations.

The Si-cycling model considered here explicitly tracks two tracers, DSi and PSi. We thus denote the modelled column vectors for DSi and PSi concentrations by $\boldsymbol{\chi}_{\mathrm{DSi}}^{\mathrm{mod}}$ and $\boldsymbol{\chi}_{\mathrm{PSi}}^{\mathrm{mod}}$. Biological uptake of silicate in the euphotic zone is essentially modelled after the simple nutrient-restoring scheme of the OCIMP-2 protocol (Najjar et al., 2007) with some slight modification. Specifically, the uptake rate $\boldsymbol{J}_{\mathrm{up}}(\boldsymbol{\chi}_{\mathrm{DSi}}^{\mathrm{mod}})$ vector is defined by

$$\boldsymbol{J}_{\mathrm{up}}\left(\boldsymbol{\chi}_{\mathrm{DSi}}^{\mathrm{mod}}\right) = \left(\boldsymbol{z} < z_0\right)\frac{\left(\boldsymbol{\chi}_{\mathrm{DSi}}^{\mathrm{mod}} - \alpha_{\mathrm{up}}\boldsymbol{\chi}_{\mathrm{DSi}}^{\mathrm{obs}}\right)^{+}}{\tau_{\mathrm{up}}} \tag{A1}$$

where $(\boldsymbol{z} < z_0)$ ensures that uptake only happens in the euphotic zone, and $\left(\boldsymbol{\chi}_{\mathrm{DSi}}^{\mathrm{mod}} - \alpha_{\mathrm{up}}\boldsymbol{\chi}_{\mathrm{DSi}}^{\mathrm{obs}}\right)^{+}/\tau_{\mathrm{up}}$ is the modified nutrient-restoring scheme. Technically, $(\boldsymbol{z} < z_0)$ is an abuse of notation and here is meant to be equal to 1 in the euphotic zone and 0 otherwise ($\boldsymbol{z}$ is the column vector depths and $z_0 = 80\,\mathrm{m}$ is the depth of the base of the euphotic zone). The column vector $\boldsymbol{\chi}_{\mathrm{DSi}}^{\mathrm{obs}}$ represents World Ocean Atlas silicate observations (WOA18 Garcia et al., 2019) regridded to the OCIM v2.0 grid. Here, $(x)^{+}$ is a shortcut notation for $x(x > 0)$, such that $(\boldsymbol{\chi}_{\mathrm{DSi}}^{\mathrm{mod}} - \alpha_{\mathrm{up}}\boldsymbol{\chi}_{\mathrm{DSi}}^{\mathrm{obs}})^{+}/\tau_{\mathrm{up}}$ only "activates" when $\boldsymbol{\chi}_{\mathrm{DSi}}^{\mathrm{mod}} > \alpha_{\mathrm{up}}\boldsymbol{\chi}_{\mathrm{DSi}}^{\mathrm{obs}}$. The difference with the standard OCMIP-2 protocol lies in the addition of the $\alpha_{\mathrm{up}}$ modifier, which is a scalar that scales the field of observed silicate, allowing the optimized model to better fit observations. With $\alpha_{\mathrm{up}}$ close to 1, this parameterization essentially allows for the model to take up silicate in the euphotic zone when the simulated DSi concentration exceeds the observations.

All the silicate that is taken up in this model is converted to sinking particulate PSi, which gravitationally settles with optimizable velocity parameter $w_{\mathrm{Si}}$. Particulate biogenic silica is assumed to remineralize and redissolve into DSi in the water column with optimzable timescale $\tau_{\mathrm{rem}}$. The rate of remineralization, $\boldsymbol{J}_{\mathrm{rem}}(\boldsymbol{\chi}_{\mathrm{PSi}}^{\mathrm{mod}})$, is thus simply defined by

$$\boldsymbol{J}_{\mathrm{rem}}(\boldsymbol{\chi}_{\mathrm{PSi}}^{\mathrm{mod}}) = \boldsymbol{\chi}_{\mathrm{PSi}}^{\mathrm{mod}}/\tau_{\mathrm{rem}}, \tag{A2}$$



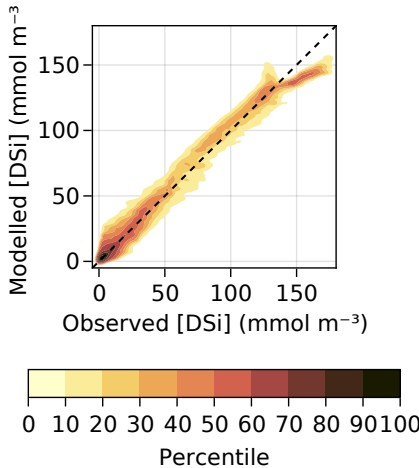

**Figure A1.** Quantiles of the cumulative joint probability density functions of modelled and observed dissolved Si concentrations for the parallel Si-cycling model. Darker colors indicate high density of data, such that $n\,\%$ of the modelled and observed data lie outside of the $n$-th percentile contour. The closer the darker contours are to the 1:1 black dashed line the better the fit.

which essentially closes the Si cycle.

Hence, the steady-state tracer equation for dissolved silicate (DSi) is

$$\mathbf{T}_{\mathrm{circ}}\,\boldsymbol{\chi}_{\mathrm{DSi}}^{\mathrm{mod}} = \boldsymbol{J}_{\mathrm{up}}(\boldsymbol{\chi}_{\mathrm{DSi}}^{\mathrm{mod}}) - \boldsymbol{J}_{\mathrm{rem}}(\boldsymbol{\chi}_{\mathrm{PSi}}^{\mathrm{mod}}) + \boldsymbol{J}_{\mathrm{geo}}(\boldsymbol{\chi}_{\mathrm{DSi}}^{\mathrm{mod}}) \tag{A3}$$

where $\boldsymbol{J}_{\mathrm{geo}}(\boldsymbol{\chi}_{\mathrm{DSi}}^{\mathrm{mod}})$ is an added term used to constrain the global inventory of Si in the system, which is set on geological timescales. In practice, we use

$$\boldsymbol{J}_{\mathrm{geo}}(\boldsymbol{\chi}_{\mathrm{DSi}}^{\mathrm{mod}}) = ([\mathrm{DSi}]_{\mathrm{geo}} - \boldsymbol{\chi}_{\mathrm{DSi}}^{\mathrm{mod}})/\tau_{\mathrm{geo}} \tag{A4}$$

where $[\mathrm{DSi}]_{\mathrm{geo}}$ is the optimizable global mean DSi concentration to which DSi concentrations are restored to with timescale $\tau_{\mathrm{geo}} = 1\,\mathrm{Myr}$. Conversely, the connected steady-state tracer equation for particulate biogenic silica (PSi) is

$$(\mathbf{T}_{\mathrm{circ}} + \mathbf{T}_{\mathrm{grav}})\,\boldsymbol{\chi}_{\mathrm{PSi}}^{\mathrm{mod}} = \boldsymbol{J}_{\mathrm{rem}}(\boldsymbol{\chi}_{\mathrm{PSi}}^{\mathrm{mod}}) - \boldsymbol{J}_{\mathrm{up}}(\boldsymbol{\chi}_{\mathrm{DSi}}^{\mathrm{mod}}), \tag{A5}$$

where $\mathbf{T}_{\mathrm{grav}}$ is the vertical (downward) transport operator representing the flux divergence of particles. Particulate Si reaching the seafloor is *not* buried and instead remains in the deepest grid cell until eventual remineralization and redissolution into DSi.

A similar optimization procedure as for the Nd-cycle is applied to optimize the five parameters collected in Table A1. The joint probability distribution of the mismatch between modelled and observed silicate is shown in Fig. A1. Overall, this simple model achieves a excellent fit to observations with a root mean square error of $XXX$.

## Appendix B: Parameters

Figure B1 shows the prior distributions and optimized values of the parameters listed in Table 2.



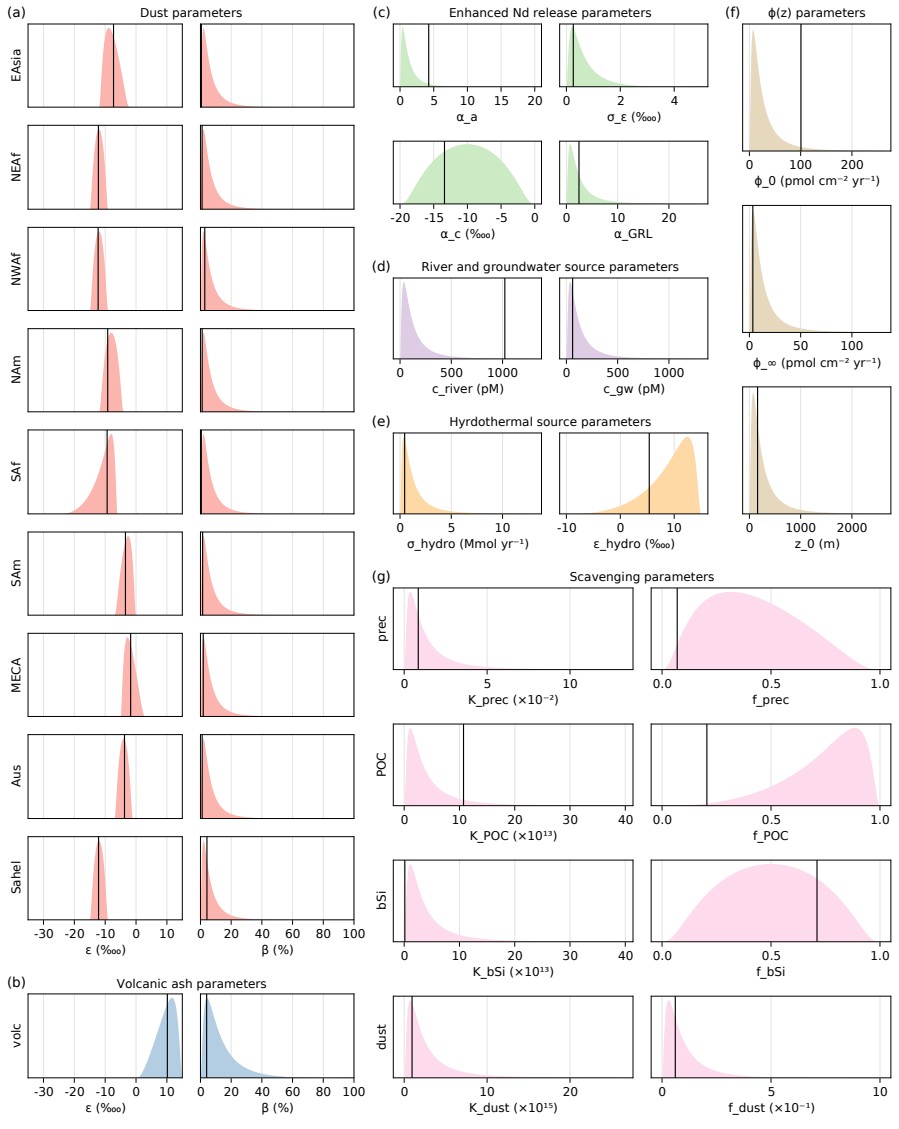

**Figure B1.** Parameter prior distributions (color-filled densities) and optimized values (vertical black lines). (a) Dust $\varepsilon_{\mathrm{Nd}}$ values ($\varepsilon_r$) and solubility ($\beta_r$) for each region of origin $r$. (b) Volcanic ash $\varepsilon_{\mathrm{Nd}}$ value ($\varepsilon_{\mathrm{volc}}$) and solubility ($\beta_{\mathrm{volc}}$). (c) Enhanced Nd release parameters, $\alpha$ curve parameters (curvature $\alpha_{\mathrm{a}}$ and center $\alpha_{\mathrm{c}}$), Greenland enhancement ($\alpha_{\mathrm{GRL}}$), and global standard deviation of per-grid-cell in situ sedimentary $\varepsilon_{\mathrm{Nd}}$ ($\sigma_\varepsilon$). (d) Rivers and groundwater Nd concentrations ($c_{\mathrm{river}}$ and $c_{\mathrm{gw}}$). (e) Global Hydrothermal source magnitude ($\sigma_{\mathrm{hydro}}$) and $\varepsilon_{\mathrm{Nd}}$ ($\varepsilon_{\mathrm{hydro}}$). (f) Sedimentary flux parameters ($\phi_0$, $\phi_\infty$, and $z_0$) (g) Reversible scavenging reaction equilibrium constants, $K_X$, and fractions returned to water when reaching seafloor, $f_X$, for each particle type $X$. See also Table 2.





**Table A1.** Optimized Si-cycling model parameters.

| Symbol | Value | Initial guess | Range | Unit | Description |
|---|---|---|---|---|---|
| $w$ | 652 | 200 | $(0, \infty)$ | $\mathrm{m\,d^{-1}}$ | Settling velocity of particulate Si |
| $\tau_{\mathrm{up}}$ | 231 | 30 | $(0, \infty)$ | d | Silicate restoring timescale |
| $\alpha_{\mathrm{up}}$ | 0.775 | 1 | $(0, \infty)$ | | Silicate scaling for restoring |
| $\tau_{\mathrm{rem}}$ | 3.12 | 1 | $(0, \infty)$ | d | bSi remineralization timescale |
| $[\mathrm{DSi}]_{\mathrm{geo}}$ | 88.1 | 80 | $(0, \infty)$ | $\mathrm{mmol\,m^{-3}}$ | Silicate geological restoring target |

**Appendix C: Shift in effectively released $\varepsilon_{\mathrm{Nd}}$**

This appendix describes the effect that enhanced Nd release with extreme $\varepsilon_{\mathrm{Nd}}$ values has on *effectively* released $\varepsilon_{\mathrm{Nd}}$. Let $X$ be a random variable with normal distribution $\mathcal{N}(\mu_\varepsilon, \sigma_\varepsilon)$ denote the observable in situ $\varepsilon_{\mathrm{Nd}}$ value, and let $Y$ denote the random variable of the effectively released $\varepsilon_{\mathrm{Nd}}$ value. The mean expected value for $Y$ is given by

$$\mathrm{E}[Y] = \frac{\mathrm{E}[X\,\alpha(X)]}{\mathrm{E}[\alpha(X)]} \tag{C1}$$

where $\alpha$ is defined in Eq. (8).

From the moments of a normal distribution, one can show that

$$\mathrm{E}[\alpha(X)] = \frac{a\sigma_\varepsilon^2 + a(c - \mu_\varepsilon)^2 + \varepsilon_{10}^2}{\varepsilon_{10}^2} \tag{C2}$$

and

$$\mathrm{E}[X\,\alpha(X)] = \frac{a(\mu_\varepsilon - 2(c - \mu_\varepsilon))\sigma_\varepsilon^2 + (a(c - \mu_\varepsilon)^2 + \varepsilon_{10}^2)\mu_\varepsilon}{\varepsilon_{10}^2} \tag{C3}$$

so that

$$\mathrm{E}[Y] = \frac{a(\mu_\varepsilon - 2(c - \mu_\varepsilon))\sigma_\varepsilon^2 + (a(c - \mu_\varepsilon)^2 + \varepsilon_{10}^2)\mu_\varepsilon}{a\sigma_\varepsilon^2 + a(c - \mu_\varepsilon)^2 + \varepsilon_{10}^2}. \tag{C4}$$

*Author contributions.* SKVH, BP, SLG, and SGJ designed the study. HL wrote preliminary MATLAB code. BP wrote the Julia model code, performed simulations, and analyzed data with input from all authors. SKVH and BP wrote the original manuscript draft, and all authors contributed to revision and editing of the manuscript prior to submission.

*Competing interests.* The authors declare no competing interests.

*Acknowledgements.* This work was supported by the Simons Foundation (Award 426570SP to SGJ), the National Science Foundation (OCE-1736896 to SGJ and OCE-1831415 to SLG, SKVH), the Investment in Science Fund at WHOI and the John E. and Anne W. Sawyer Endowed



Fund in Support of Scientific Staff (SKVH), and the Storke Endowment of the Department of Earth and Environmental Sciences, Columbia University (SLG).





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
