# Peer review of "GNOM v1.0: An optimized steady-state model of the modern marine neodymium cycle"

_Geoscientific Model Development, 2021_

## Referee Comment (RC4)

Pasquier et al has presented an interesting inverse model of Nd based on OCIM, and performed Nd cycle parameter optimization against observations of seawater Nd and eNds. Such models have been developed for many other biogeochemical tracers in the GEOTRACES era, and it's great to see that it is now applied to Nd. I think the model has a potential for wide use given the computational efficiency compared with GCMs or ESMs, and obvious advantage over box models.

From the modeling perspective, I found the OCIM modeling framework easy to understand as it has matured through its history of modeling biogeochemical tracers. The optimization procedure here still has more to be desired, though the inherent difficulty is understood. Dr. Pasquier should be applauded to have taken considerable effort to develop various open source Julia packages devoted to ocean biogeochemistry under the transport matrix framework of AIBECS.jl, of which GNOM.jl is a special case for Nd.

However, from the Nd cycle perspective, as an observationalist and a geochemist, I disagree with the treatment of the Nd sources in the model. Here are my main criticisms.

1. There is a disregard of the huge literature on observations and measurements of globe Nd sources and the relevant parameters, including dust/ash solubility, riverine/groundwater Nd concentrations and eNd, benthic flux and scavenging. Such that the "optimized" parameters are not evaluated against measurements, leading to values that are unrealistic: it's hard to accept the model "optimized" value being 9 orders of magnitude higher than measurements (in the case of scavenging Kxs), or exact opposite to the trend in observations (in the case of benthic flux).

2. Despite the "optimized" parameters being orders of magnitude different from observations and previous models, this study arrived at global integrated sources similar to previous studies, suggesting that the Nd source parameters can NOT be uniquely determined ONLY by optimizing the seawater Nd and isotopes data. Either that the parameterizations are incorrect, or that a proper inversion should include measurements of the Nd sources in the cost function. In this respect, proper evaluation of the parameter covariance is required before optimization.

3. The parameterizations for Nd sources often make unsubstantiated assumptions on secondary issues that we have no constraint on, while adopt overly simplistic, even incorrect, representations of the first-order variables that we do have fairly good constraints on. For example, in the riverine parametrization, the authors use the detail reconstruction of global river discharge, which is only a secondary parameter in the Nd flux, yet force the river Nd concentration, the first-order parameter, to be constant despite observations that showing it varying by 2~3 orders of magnitude. Another example is that when treating the sediment flux, a complicated and arbitrary parameterization was created for the differential release of Nd from certain lithology (of secondary important, even smaller than analytical uncertainty), while ignoring that the baseline sediment flux eNd (the first-order variable) map of Robinson has large uncertainties.

Overall, without proper considerations of the actual measurements of the Nd source parameters, I disagree that the black box optimization approach can give any meaningful insight into the Nd cycle compare to previous studies.

Following are my comments on the parameterizations of Nd sources, and optimization approach and some other issues.

Dust

- 40 ppm Nd in dust is higher than that in PAAS (34 ppm) and UCC (27 ppm) (Rudnick and Gao, 2014). Why is this number chosen?
- There are observations of dust eNd and solubility (Goldstein et al., 1984; Robinson et al., 2021), which are disregarded. Asian dust on average have an eNd of ~ -10 (Chen and Li, 2011), the "optimized" value of -7.6 seems too high. Australia is mainly made of old cratonic rocks, how could it supply a dust source as high as -4.0? Similarly, it's hard to image that the North American source could be as radiogenic as -4.3. It's difficult to accept the optimized values based on our knowledge of local geology. The parameter range used for optimization seems too narrow to me for many regions.
- The "optimized" (~20%, even up to 80%!) solubility is orders of magnitude higher than observations (~1%) (Greaves et al., 1994). Not surprise that this study gives a dust contribution to the global Nd budget orders of magnitude higher than previous models.
- L480. This is a misunderstanding of the dust flux in (Tachikawa et al., 2003). The "known" dust flux used by Tachikawa is only 2.8 Mmol/yr, like the GCM models. In fact, Tachikawa is the main source of dust flux in the GCM models cited, that's why they are similar. The extremely high 60 Mmol/yr number cited (I think this is a miss citation. It should be 42 Mmol/yr in Tachikawa, I don't find this 60 Mmol/yr number) is only "fictional", that Tachikawa suggested that if it is true then we can solve the "missing source" problem. But there's no evidence that it is true. Thus, it is safe to say the optimized dust flux in this study is one order of magnitude higher than all previous estimates.

Ash

- L180. Explosive eruptions mainly produce felsic/rhyolitic ash that can travel long distance. Mafic eruptions are generally effusive and the resulting ash doesn't travel far (Scudder et al., 2016).
- L181. I don't find any mention of volcanic ash in Chien. So where are the flux data from?
- L190. Ash eNd is varied. See the global volcanic arc eNd data of (Kelemen et al., 2014). It seems the authors have chosen an initial value of +10, which already is the upper limit of volcanic materials. The "optimized" ash eNd of +13 is clearly unrealistic.
- The "optimized" ash solubility of 76% is extreme. Using the ash leaching experiment of (Du et al., 2016) we can get a ash Nd solubility of ~1%. The argument that ash is more soluble than dust is somewhat an urban legend without clear evidence. In situ observations also show no effect of a recent Icelandic ash fall on surface seawater eNd (Lambelet et al., 2016). Regardless, 76% seems unrealistic.

Sediment

- The sediment source parametrization is completely unsubstantiated and published sediment flux data have been disregarded. I see no rationale to support the assumption that benthic flux at the deep ocean is zero, which is clearly contradicted by observations (Abbott et al., 2015; Du et al., 2020) that instead show the highest benthic flux is from the deep ocean. The recent study from the abyssal plain of the Pacific (even at ~5000 m) still show a benthic flux of 3.5 pmol/cm$^2$/yr (Haley et al., 2021), similar to what's observed on the Oregon shelf. Based on the existing data (Du et al., 2020), benthic flux either doesn't change with depth, or increases with depth, opposite to the

parameterization here. It has also been show that benthic Nd flux doesn't correlate with POC flux or bottom water O2 (Du et al., 2018).

- L210. If linear increasing is allowed, then z0 should be allowed to be negative, rather than forced to be positive.
- L219. This is another urban legend with unclear evidence.
- L226. Evidence? Check (Blaser et al., 2019) which estimated the benthic flux from freshly eroded Heinrich Layers. It doesn't seem there is reason to believe it to be much higher than elsewhere.
- L240. This parameterization seems unnecessary if the resulting difference is only 0.03 epsilon, given typical analytical uncertainty of ~0.4 epsilon, and the reproducibility of sediment digestion/leach is ~1 epsilon or more. The uncertainty of the gridded product of Robinson is probably more than 5 epsilons because of the poor spatial coverage of the raw data and the inconsistency in the types of raw data (leach residual, bulk sediment, size fractions etc.). Thus, the biggest uncertainty to capture is that of the Robinson dataset, which should be allowed to vary 5~10 epsilon in the model.

River

- I disagree with the parameterization of riverine input. In my option the necessary source data should come from: (Bayon et al., 2015; Goldstein and Jacobsen, 1987, 1988a, b; Goldstein et al., 1984). I would rather use the measured riverine Nd source, however limited, than using clearly incorrect parameterization, however globally detailed. This is an example of ignoring the first order issue, i.e., the variations in riverine Nd concentrations, while overly concerned with the secondary issue, i.e., the river discharge.
- L249. Riverine Nd concentrations vary by 2 orders of magnitude (Goldstein and Jacobsen, 1987).
- L255. How is the smoothing done? What is the spatial scale? The effect of riverine input doesn't extend beyond the estuary. What is in fig2d seems too extreme to be true.
- Where is the estuary removal term in the parametrization?
- L256. I disagree with the use Robinson data for this purpose. Remember the margin eNd of Robinson is an interpolated product based on rough geological map with very limited outcrop rock eNd, which has nothing to do with rivers. The relationship between riverine eNd and the eNd of the rocks from the drainage basin is a complicated matter (Bayon et al., 2015). There are measurements of riverine dissolved and particulate eNd from (Bayon et al., 2015; Goldstein and Jacobsen, 1987, 1988a, b; Goldstein et al., 1984) that should be used.

Groundwater

- Same problems as the riverine parameterization: groundwater Nd concertation is not constant, and it varies by 7 orders of magnitude! I see no reason to use the Robinson eNd dataset either. We already have the measured Nd flux and eNd from (Johannesson and Burdige, 2007), why would you use parameterizations that's unsubstantiated.
- What is the removal parameterization of the groundwater flux upon entering the ocean?

Hydrothermal

- Hydrothermal system is a net SINK of Nd (Stichel et al., 2018; Basak et al., 2021). The release of hydrothermal Nd makes local seawater eNd only slightly more radiogenic (within analytical uncertainty), while the seawater Nd concentration decreases significantly because of scavenging by

hydrothermal particles. Therefore, both a source and a sink should be implemented for the hydrothermal system, with the sink being much larger than the source.

Scavenging

- L316. I disagree. For some reason the most important Nd scavengers, Fe-Mn oxides (Schijf et al., 2015; Sholkovitz et al., 1994), are ignored like previous models. And I don't see how scavenging by Fe-Mn oxides can be treated as "precipitation" as it will depend on oxide concentrations that are redox sensitive.
- The "optimized" Kxs are unrealistic. Previous models generally use values around 10^6 (Siddall et al., 2008; Arsouze et al., 2009). Lab measurements also give results on similar orders of magnitude (Schijf et al., 2015). Yet here the optimized values are 10^13~10^15!
- I think that such high optimized Kxs are the results of overly high dust flux in the model (see my comments up), so extremely intense scavenging is needed to bring surface Nd concentration down.

Optimization

- There are so many parameters to be optimized. What is the covariance structure of them? For example, the dust solubility will probably covary with the scavenging Kxs. Therefore, can we get a unique solution of the parameter values? These questions need to be studied before performing optimization. That so many of the optimized parameters are unrealistic give me little confidence in the optimization.
- L335. It is nice to include penalty terms for the parameters, but I think the choice of parameter ranges are not very reasonable give that the optimized values are often unrealistic.
- The optimization should be penalized against not just measured seawater Nd/eNd, but also measurements of Nd sources.
- A volumetric weighing should be included in the cost function, so the optimized results are not biased toward the surface ocean because of high data density. The Nd community is generally more concerned about the deep ocean than the surface, in contrast to other communities, for several reasons: 1 Surface eNd is heterogenous and local, thus contains little global information. 2. Surface input generally varies a lot in the past, so surface eNd isn't useful for watermass tracing. 3. We can only reconstruct deep ocean eNd in the past, as there is no archive of surface water eNd.
- Additionally, basinal weighing could be useful given that the data is so concentrated in the Atlantic.
- How does the choice of the prior affect the results? For many parameters we know very little but the orders of magnitude. Isn't a uniform prior more reasonable than lognormal/logit priors when we essentially have no information on the actual distributions?
- L400. This needs to be demonstrated. And are the ranges of the randomized initial values large enough that we can really believe the results are global?

  Minor points

- The unit of eNd is epsilon, not per 10 thousand. This is a radiogenic isotopic ratio, not a stable isotopic ratio. The expression of epsilon looks like delta for stable isotopes, but the reporting convention is different. I have never seen the per 10 thousand unit for eNd in literature.

- L129. Not because there's no stable isotopic fractionation, but because by convention when reporting radiogenic Nd isotopes stable fractionation effect, be it natural or instrumental, is removed by normalizing to constant 146Nd/144Nd ratio. That is, even if you model isotopic fractionation, you have to remove it when you convert to eNd units.
- L408. Needs demonstration.
- Fig9/10. It's very difficult to see the "fit". Instead show the model-date misfit rather than just overlay the data on top of the model.
- L484. If the optimized solubility is not supported by observations, then there is no reason to believe it.
- L487. I disagree with this philosophy of "consistent with previous model" but ignoring actual observations of the fluxes (Du et al., 2020; Goldstein and Jacobsen, 1987; Tachikawa et al., 2003; Johannesson and Burdige, 2007).
- L522. Seem exaggeration. What about the MIT-ECCO, which has also been used as TMM?
- L545. Perhaps not exactly the same calculation, but similar concept of partitioning N/S sources already exist (Gu et al., 2020)
- L571. The most prominent caveat is the disregard of observations of Nd sources.
- L543. Disagree, parameter choices often unrealistic.

Abbott, A. N., Haley, B. A., and McManus, J.: Bottoms up: Sedimentary control of the deep North Pacific Ocean's εNd signature, Geology, 43, 1035–1035, https://doi.org/10.1130/G37114.1, 2015.

Arsouze, T., Dutay, J.-C., Lacan, F., and Jeandel, C.: Reconstructing the Nd oceanic cycle using a coupled dynamical – biogeochemical model, Biogeosciences, 6, 2829–2846, https://doi.org/10.5194/bg-6-2829-2009, 2009.

Basak, C., Wu, Y., Haley, B. A., Muratli, J., Pena, L. D., Bolge, L., Fitzsimmons, J., Sherrell, R. M., and Goldstein, S. L.: Role of suspended particulate matter in governing dissolved Nd in the Southern East Pacific Rise hydrothermal plume, Goldschmidt2021 • Virtual • 4 - 9 July, 2021.

Bayon, G., Toucanne, S., Skonieczny, C., André, L., Bermell, S., Cheron, S., Dennielou, B., Etoubleau, J., Freslon, N., Gauchery, T., Germain, Y., Jorry, S. J., Ménot, G., Monin, L., Ponzevera, E., Rouget, M.-L., Tachikawa, K., and Barrat, J. A.: Rare earth elements and neodymium isotopes in world river sediments revisited, Geochim. Cosmochim. Acta, 170, 17–38, https://doi.org/10.1016/j.gca.2015.08.001, 2015.

Blaser, P., Pöppelmeier, F., Schulz, H., Gutjahr, M., Frank, M., Lippold, J., Heinrich, H., Link, J. M., Hoffmann, J., Szidat, S., and Frank, N.: The resilience and sensitivity of Northeast Atlantic deep water εNd to overprinting by detrital fluxes over the past 30,000 years, Geochim. Cosmochim. Acta, 245, 79–97, https://doi.org/10.1016/j.gca.2018.10.018, 2019.

Chen, J. and Li, G.: Geochemical studies on the source region of Asian dust, Sci. China Earth Sci., 54, 1279, https://doi.org/10.1007/s11430-011-4269-z, 2011.

Du, J., Haley, B. A., and Mix, A. C.: Neodymium isotopes in authigenic phases, bottom waters and detrital sediments in the Gulf of Alaska and their implications for paleo-circulation reconstruction, Geochim. Cosmochim. Acta, 193, 14–35, https://doi.org/10.1016/j.gca.2016.08.005, 2016.

Du, J., Haley, B. A., Mix, A. C., Walczak, M. H., and Praetorius, S. K.: Flushing of the deep Pacific Ocean and the deglacial rise of atmospheric CO 2 concentrations, Nat. Geosci., 11, 749–755, https://doi.org/10.1038/s41561-018-0205-6, 2018.

Du, J., Haley, B. A., and Mix, A. C.: Evolution of the Global Overturning Circulation since the Last Glacial Maximum based on marine authigenic neodymium isotopes, Quat. Sci. Rev., 241, 106396, https://doi.org/10.1016/j.quascirev.2020.106396, 2020.

Goldstein, S. J. and Jacobsen, S. B.: The Nd and Sr isotopic systematics of river-water dissolved material: Implications for the sources of Nd and Sr in seawater, Chem. Geol. Isot. Geosci. Sect., 66, 245–272, https://doi.org/10.1016/0168-9622(87)90045-5, 1987.

Goldstein, S. J. and Jacobsen, S. B.: Nd and Sr isotopic systematics of river water suspended material: implications for crustal evolution, Earth Planet. Sci. Lett., 87, 249–265, https://doi.org/10.1016/0012-821X(88)90013-1, 1988a.

Goldstein, S. J. and Jacobsen, S. B.: Rare earth elements in river waters, Earth Planet. Sci. Lett., 89, 35–47, https://doi.org/10.1016/0012-821X(88)90031-3, 1988b.

Goldstein, S. L., Onions, R. K., and Hamilton, P. J.: A Sm-Nd isotopic study of atmospheric dusts and particulates from major river systems, Earth Planet. Sci. Lett., 70, 221–236, https://doi.org/10.1016/0012-821X(84)90007-4, 1984.

Greaves, M. J., Statham, P. J., and Elderfield, H.: Rare earth element mobilization from marine atmospheric dust into seawater, Mar. Chem., 46, 255–260, https://doi.org/10.1016/0304-4203(94)90081-7, 1994.

Gu, S., Liu, Z., Oppo, D. W., Lynch-Stieglitz, J., Jahn, A., Zhang, J., and Wu, L.: Assessing the potential capability of reconstructing glacial Atlantic water masses and AMOC using multiple proxies in CESM, Earth Planet. Sci. Lett., 541, 116294, https://doi.org/10.1016/j.epsl.2020.116294, 2020.

Haley, B. A., McManus, J., Du, J. (JD), and Vance, D.: Rare earth elements in the pore waters of abyssal sediments, Goldschmidt2021 • Virtual • 4 - 9 July, 2021.

Johannesson, K. H. and Burdige, D. J.: Balancing the global oceanic neodymium budget: Evaluating the role of groundwater, Earth Planet. Sci. Lett., 253, 129–142, https://doi.org/10.1016/j.epsl.2006.10.021, 2007.

Kelemen, P. B., Hanghøj, K., and Greene, A. R.: One View of the Geochemistry of Subduction-Related Magmatic Arcs, with an Emphasis on Primitive Andesite and Lower Crust, in: Treatise on Geochemistry (Second Edition), edited by: Holland, H. D. and Turekian, K. K., Elsevier, Oxford, 749–806, 2014.

Lambelet, M., van de Flierdt, T., Crocket, K., Rehkämper, M., Kreissig, K., Coles, B., Rijkenberg, M. J. A., Gerringa, L. J. A., de Baar, H. J. W., and Steinfeldt, R.: Neodymium isotopic composition and concentration in the western North Atlantic Ocean: Results from the GEOTRACES GA02 section, Geochim. Cosmochim. Acta, 177, 1–29, https://doi.org/10.1016/j.gca.2015.12.019, 2016.

Robinson, S., Ivanovic, R., van de Flierdt, T., Blanchet, C. L., Tachikawa, K., Martin, E. E., Cook, C. P., Williams, T., Gregoire, L., Plancherel, Y., Jeandel, C., and Arsouze, T.: Global continental and marine

detrital εNd: An updated compilation for use in understanding marine Nd cycling, Chem. Geol., 567, 120119, https://doi.org/10.1016/j.chemgeo.2021.120119, 2021.

Rudnick, R. L. and Gao, S.: Composition of the Continental Crust, in: Treatise on Geochemistry (Second Edition), edited by: Holland, H. D. and Turekian, K. K., Elsevier, Oxford, 1–51, 2014.

Schijf, J., Christenson, E. A., and Byrne, R. H.: YREE scavenging in seawater: A new look at an old model, Mar. Chem., 177, 460–471, https://doi.org/10.1016/j.marchem.2015.06.010, 2015.

Scudder, R. P., Murray, R. W., Schindlbeck, J. C., Kutterolf, S., Hauff, F., Underwood, M. B., Gwizd, S., Lauzon, R., and McKinley, C. C.: Geochemical approaches to the quantification of dispersed volcanic ash in marine sediment, Prog. Earth Planet. Sci., 3, https://doi.org/10.1186/s40645-015-0077-y, 2016.

Sholkovitz, E. R., Landing, W. M., and Lewis, B. L.: Ocean particle chemistry: The fractionation of rare earth elements between suspended particles and seawater, Geochim. Cosmochim. Acta, 58, 1567–1579, https://doi.org/10.1016/0016-7037(94)90559-2, 1994.

Siddall, M., Khatiwala, S., van de Flierdt, T., Jones, K., Goldstein, S. L., Hemming, S., and Anderson, R. F.: Towards explaining the Nd paradox using reversible scavenging in an ocean general circulation model, Earth Planet. Sci. Lett., 274, 448–461, https://doi.org/10.1016/j.epsl.2008.07.044, 2008.

Stichel, T., Pahnke, K., Duggan, B., Goldstein, S. L., Hartman, A. E., Paffrath, R., and Scher, H. D.: TAG Plume: Revisiting the Hydrothermal Neodymium Contribution to Seawater, Front. Mar. Sci., 5, 96, https://doi.org/10.3389/fmars.2018.00096, 2018.

Tachikawa, K., Athias, V., and Jeandel, C.: Neodymium budget in the modern ocean and paleo-oceanographic implications, J. Geophys. Res. Oceans, 108, 3254, https://doi.org/10.1029/1999JC000285, 2003.

---

## Author Comment (AC1)

**Reply to RC1: 'Comment on gmd-2021-338' by Referee #1**

Below are Referee #1 comments reproduced in grey italic and our point-by-point replies in black. Unless specified otherwise, line numbers refer to the revised manuscript.

*Pasquier and co-authors propose the first inverse model of the global marine biogeochemical cycle of Nd and its isotopes (GNOM). In their approach, the GNOM is embedded in a data-constrained steady-state circulation allowing them to estimate the biogeochemical parameters controlling the Nd/eNd cycle (sources, transformation and sink) via systematic objective optimization. This is a very interesting approach, efficient and full of promises. The manuscript is clearly written although sometimes not explicit enough despite its length. The illustrations and figures are correct. I consider that this work deserves publication in GMD, not without some improvements. I hope my comments below will be of use.*

*My general comment is that most of the working hypotheses are not discussed and argued enough. In addition, references are too often lacking. Some examples below.*

We thank Referee #1 for their comments and respond to each below.

*Aeolian sources*

*Line 164 : How is the solubility parameter estimated ? The optimized value of 82.8% solubility for N-Am is incredibly high compared to what is published so far. This surprising value could be better discussed.*

As explained in the original manuscript, each solubility parameter is "free" to be optimized to any value within (0–100)%. The optimization of these solubilities aims to minimize the mismatch with observed seawater [Nd] and $\varepsilon_{Nd}$ but also includes a penalty for deviating too much from their original value (set at 5 % for dust) within the (0–100) % range. That penalty smoothly goes from 0 to infinity at the bounds of the range, proportionally to the negative-log-likelihood of the prior distributions shown in Fig. B1. We note that the difference in solubility values between our study and previous modelling studies was already pointed out in our manuscript in Section 3.3.1.

In response, we have added more discussion to Section 3.3.1 (L519–531):
"While this falls within the 0.69–60 Mmol yr⁻¹ range for global aeolian source magnitudes of previous modelling studies, our aeolian sources are an order of magnitude larger than previous GCM-based modelling studies (0.69–3.5 Mmol yr−1) (Table 1). This is likely due to our optimization procedure, during which Nd solubility is allowed to be adjusted within the whole 0–100% range, compared to previous GCM-based studies that typically use a fixed 2% solubility (Arsouze et al., 2009; Gu et al., 2019; Pöppelmeier et al., 2020). This is despite the initial guesses for $\beta_r$ values for dust set at 5%, which penalizes large solubilities more than low solubilities (see the increased probability densities for low solubilities in Fig. B1a and B1b). We note that generally worse fits to [Nd] and $\varepsilon_{Nd}$ observations have been achieved with some of our optimization runs ending in distinct local minima with significantly smaller dust solubilities. (We

do not show these worse mismatches but we show the corresponding initial and final parameter values in Fig. B1.) We emphasize that it is not the goal of this model description to establish estimates of the GNOM parameters and that we welcome future GNOM users to apply narrower ranges for those parameters for which they have better constraints (for example, restricting dust Nd solubilities to values below 10%)."

*The same manner, how is eNd estimated (what are the references leading to the values attributed to each area?)?*

Dust $\varepsilon_{Nd}$ parameters are jointly estimated in the same manner as the solubility parameters during the same optimization of the objective function. Initial values and ranges have been chosen based on literature and the expertise of the authors, as was already noted in the caption of Table 2 in which initial values and ranges of each parameter are collected.

In response, we have added the following in Section 3.1 (L445–447):
"Initial guesses and optimizable ranges for each parameter were determined from the literature and the expertise of authors. Initial guesses and final parameter values, along with unit, prescribed range, and a brief description, are given in Table 2. Parameter prior distributions, their randomized initial values, and final optimized values are shown in Fig. B1."

*Line 183 : the authors assume a constant Nd value for dust and volcanic ash inputs, but the reference allowing this hypothesis is missing.*

We appreciate that Nd content in dust and volcanic ash could vary. However, to our knowledge, there is no published model or global map of Nd content in ocean-deposited atmospheric dust or volcanic ash. The simplest model for dust content is then to assume it is constant. Because the Nd entering the ocean is proportional to the Nd content multiplied by the solubilities, we chose 40 µg g$^{-1}$, which is a value closer to the maximum dust Nd occurrence as reported, e.g., in Goldstein et al. (1984).

In response, we have added (L163–164):
"(which is within the 11.93 to 45.76 ppm range of atmospheric dust observations in Goldstein et al., 1984)".

**Sediments**

*The paragraph presenting the modelling of the Nd release from the sediments is not clear and could be re-written.*

We have tried to clarify this section, see specific responses below.

*Line 210, how the normalization constant was chosen? why 10 per mil?*

The $\varepsilon_{10}$ = 10 ‰ normalization constant was chosen such that, for $a$ = 1, an $\varepsilon_{Nd}$ deviation of ±10 ‰ from $c$ incurs a doubling of effectively released Nd. Note that its specific value does not truly

matter because it is implicitly absorbed by the free parameter *a* that controls the curvature of $\alpha(\varepsilon_{Nd})$. Its purpose is only to non-dimensionalize the base number being squared, $((\varepsilon_{Nd} - c) / \varepsilon_{10})$.

In response, we now say (L227–229):
"(...) the optimizable parameters *a* and *c* control the curvature, respectively, while $\varepsilon_{10} = 10$ ‰ is a normalization constant. (The specific value of ε10 is unimportant because it gets absorbed by the optimizable parameter a during the optimization).."

*It is written « Extreme sedimentary eNd values are often associated with rather fresh, and thus reactive, detrital material » (argument used again line 237). Although this is verified for fresh basaltic material (often highly radiogenic and soluble), this is not verified for the very non-radiogenic materials which are of granite or metamorphic origin, and thus not known to be soluble. Again, references allowing such hypothesis are lacking. Thus, the scaling (and the quadratic function) proposed in figure 4 should be better explained.*

We agree with Referee #1 that the text describing the motivation behind this parameterization was lacking in references. However, it is important to note that the Nd enhancement is not imposed. It is merely an option that the optimizer can turn "on" or "off" to best match the observations.

In response, we now say (L232–240):
"This quadratic parameterization is motivated by the fact that extreme sedimentary $\varepsilon_{Nd}$ values are often associated with rather fresh, and thus reactive, detrital material. We emphasize that this enhancement can be turned "on"' or "off" depending on the choice of parameters (*a*=0 turns it off). However, maybe coincidentally, extremely high $\varepsilon_{Nd}$ values are generally associated with relatively young volcanic Nd that is more reactive and readily soluble (Lacan and Jeandel, 2005; Pearce et al., 2013; Wilson et al., 2013; Blaser et al., 2016, 2020) and previous model studies have resorted to different enhanced Nd release parameterizations to achieve a similar effect (see, e.g., Poppelmeier et al., 2020). The same is not necessarily true for rocks with extremely low $\varepsilon_{Nd}$ values, however it so happens that much of the region around the Labrador Sea (Greenland and northern Canada) is currently, or was previously, glaciated, which has resulted in a large amount of fine-grained crystalline (and thus labile) detritus with extremely negative $\varepsilon_{Nd}$ (von Blanckenburg and Nägler, 2001)."

*Line 226 : The hypothesis of not considering enhanced release in Antarctic should be justified (see for example Paul Carter's work)*

We did not include such an enhancement because we did not think it was necessary to reproduce global $\varepsilon_{Nd}$ patterns owing to the fact that sedimentary release of Nd with extreme $\varepsilon_{Nd}$ values would be rapidly mixed and averaged by the circumpolar current, as evidenced by the relatively uniform values of observed Southern Ocean $\varepsilon_{Nd}$. Future versions of the GNOM could include such an Antarctic enhancement, should Antarctic sedimentary Nd source be shown to be important for the global Nd cycle.

In response, we now say (L244–246):
"(For simplicity, we did not account for potentially enhanced Nd release in Antarctic because we assume that extreme $\varepsilon_{Nd}$ released by sediments in the Antarctic would be relatively rapidly mixed along the circumpolar current.)"

*Riverine inputs*

*There is no consideration of the estuarine removal, estimated at ca 70%. The initial value is high (100 pM) and the optimized one even higher (376 pM) which is far above what is measured yet at the exit of the estuaries. This could be more discussed, since the real source of Nd to the ocean is at the estuarine mouths.*

We considered estuarine removal as being implicitly included in our *effective* riverine [Nd] concentration and although this was indicated in Table 2 of our original submission, we agree that we should have been more explicit. We emphasize that the goal of our manuscript is not to accurately estimate the parameter values of GNOM v1.0 but rather to describe a model with a built-in framework for efficient optimization. We believe that an inverse-model estimate of the effective riverine Nd concentrations would be an excellent follow-up work. This could consist, for example, of implementing separately [Nd] parameters for major rivers and tightening their prior ranges based on available observations.

In response, we now say (L268–273):
"As a simplification, and to reduce the total number of free parameters in the model, we assume that all rivers share the same Nd concentration $c_{river}$, which is the parameter that controls the global riverine source magnitude (see Table 2). (Future improvements of the GNOM could include optimizable [Nd] parameters for each individual major river, constrained by ranges based on observations.) Because the GNOM v1.0 does not resolve estuary removal processes, our $c_{river}$ is to be understood as an *effective* Nd concentration that implicitly accounts for Nd removal in estuaries and is thus the concentration that makes it into the ocean."

*Reversible scavenging*

*Line 289 : What are the particle types that will be considered ?*

The answer was a few lines below in the main text, lines 306–316: "We consider four different particle types for scavenging Nd. (i) Scavenging by dust particles (...) (ii) Scavenging by particulate organic carbon (POC) (...) (iii) Scavenging by biogenic silica (bSi), or opal (...) (iv) A particle-independent scavenging is included to prevent accumulation of Nd where the concentration fields of dust, POC, and opal are unrealistically low."

No changes to the manuscript in response to this comment.

*What will be the role (and justification) of the divergence operator?*

The flux-divergence operator is the linear differential operator that, applied to the concentration of dissolved [Nd], returns the flux divergence of scavenged Nd. Hence, its "role" is to compute

the rate at which scavenging adds and removes [Nd] in each model-grid box. Its "justification" is that using a linear operator allows us to represent the Nd cycle entirely linearly, both allowing for efficient solving of the system (~10 seconds on a modern laptop) and for advanced diagnostics such as showcased later in the manuscript.

We have added a sentence that explains this term in plain english (L324–325):
"We use $\mathbf{T}_X$ to compute the rate at which reversible scavenging adds or removes Nd in each grid box."

*Line 298 : which reference allows the hypothesis that reversible exchange is occurring faster than particulate and ocean transport?*

This simplifying assumption is common in models that include particle scavenging (e.g., van Hulten et al. 2018, https://doi.org/10.5194/gmd-11-3537-2018). Particle scavenging is extremely important in the ocean and there is clearly still much to learn. While the reversible exchange model is a simplification of what really happens in the ocean, it appears to adequately capture the major patterns observed.

In response, we have added (L329–330):
"(This assumption is common in models that include scavenging and simpler than resolving the adsorption/desorption rates dynamically (e.g., van Hulten et al., 2018).)"

*Line 308 : the vertical velocity of dust (1 km/y) is high when refering to Hayes' estimates using Th isotopes in the oligotrophic areas (ca 300 m/y) ; again, a reference is lacking here.*

While we agree that this value may be too high compared to previous estimates, it does not matter for our model. This is because, as already explained in the original manuscript lines 414–417, "$K_X$ and $w_X$ perfectly compensate each other. In fact, only their product, $K_X w_X$ (...) appears in the tracer equations (see Eq. (16) or, e.g., John et al., 2020)".

In response, we have added some plain-language example and now say (L452–455):
"(...) because $K_X$ and $w_X$ can perfectly compensate each other. For example, doubling $K_X$ while halving $w_X$ has no effect on Nd distributions and the objective function. Only their product, $K_X w_X$, which sets strength of the "scavenging pump" through the operator matrix $\mathbf{T}_{scav}$, appears in the tracer equations (see Eq. (16) or, e.g., John et al., 2020), such that these parameters cannot be easily optimized independently."

*Line 312 : where are the areas where the concentrations of dust, POC and opal are too low ? which surface of the modelled ocean this represents ? Is this additional arbitrary term important for the modelling?*

Preliminary runs without the additional term (not shown) gave significantly worse global fits. Being its first release, the GNOM v1.0 has a relatively simple scavenging parameterization. Scavenging is a complicated biochemical process involving many particles that remains largely elusive in the current state of ocean modelling.

In response to this comment and corresponding comments by the other referees, we have added (L345–350):
"We note that while this additional particle-independent scavenging sink could compensate for additional types of particles not currently implemented in the model, it is likely that more scavenging particle types are required for an accurate representation of the Nd cycle. These include hydrothermal particles (which should result in hydrothermal systems being a net sink; Stichel et al., 2018) and iron–manganese oxides (which are potentially the most important scavenging particles; Schijf et al., 2015; Sholkovitz et al., 1994)."

*Actually, I appreciated the initiative, and the effort made to simulate this biogeochemical cycle using an inverse method but I am left hungering for more discussion and critical debate regarding the results. I acknowledge that the manuscript is already long. An option could be to report the different modelling hypotheses (and most of the equations) in the suppl. material (which would allow to better describe them) and to propose more science in the main text. Some sensitivity studies should also be presented (as limiting the dust dissolution rates into published ranges, reducing c to values compatible with estuarine outputs, increasing the sediment release from the shelves…etc…*

This is out of the scope for a model-description paper. We fully acknowledge that some optimized parameter values are unrealistic, and hope to dig into the details of the scientific results in a separate future publication following some refinement.

In response, we have tried to clarify in the abstract that the intention of this paper is to describe a model and present initial results (L6–13):
"To make sense of the increasing collection of observational Nd and $\varepsilon_{Nd}$ data, we present and describe the global neodymium ocean model (GNOM) v1.0, the first inverse model of the global marine biogeochemical cycle of Nd. (...) allowing us to present preliminary estimates of biogeochemical parameters. (...) This model is open-source and freely accessible, is written in Julia, and its code is easily understandable and modifiable for further community developments, refinements, and experiments."

**Reply to RC4: 'Comment on gmd-2021-338' by Referee #2**

Below are Referee #2 comments reproduced in grey italic and our point-by-point replies in black. Line numbers in our response refer to the revised manuscript with changes accepted.

*Pasquier et al has presented an interesting inverse model of Nd based on OCIM, and performed Nd cycle parameter optimization against observations of seawater Nd and eNds. Such models have been developed for many other biogeochemical tracers in the GEOTRACES era, and it's great to see that it is now applied to Nd. I think the model has a potential for wide use given the computational efficiency compared with GCMs or ESMs, and obvious advantage over box models.*

*From the modeling perspective, I found the OCIM modeling framework easy to understand as it has matured through its history of modeling biogeochemical tracers. The optimization procedure here still has more to be desired, though the inherent difficulty is understood. Dr. Pasquier should be applauded to have taken considerable effort to develop various open source Julia packages devoted to ocean biogeochemistry under the transport matrix framework of AIBECS.jl, of which GNOM.jl is a special case for Nd.*

We thank Referee #2 for their extremely thorough evaluation of our paper and have replied to each comment below. We would like to emphasize at the start of this response, that the goal of this paper is to describe a new model we have developed. With this description we also provide preliminary estimates of parameter values, based on our objective optimization and the stated ranges. The goal is *not* to present final parameter values or to assert that our first guesses are necessarily correct. We hope that by publishing this model description paper, others can make use of the model and make improvements as they see fit. We also hope to improve upon the model in the future.

In response, we have clarified the goal of the paper in the abstract by adding/revising the following passages (L6–13):
"(...) we present and describe the global neodymium ocean model (GNOM) v1.0, the first inverse model of the global marine biogeochemical cycle of Nd. The GNOM (...) affords spectacular computational efficiency, which we leverage to perform systematic objective optimization, allowing us to present preliminary estimates of biogeochemical parameters. (...) This model is open-source and freely accessible, is written in Julia, and its code is easily understandable and modifiable for further community developments, refinements, and experiments."

*However, from the Nd cycle perspective, as an observationalist and a geochemist, I disagree with the treatment of the Nd sources in the model. Here are my main criticisms.*

*1. There is a disregard of the huge literature on observations and measurements of globe Nd sources and the relevant parameters, including dust/ash solubility, riverine/groundwater Nd*

*concentrations and eNd, benthic flux and scavenging. Such that the "optimized" parameters are not evaluated against measurements, leading to values that are unrealistic: it's hard to accept the model "optimized" value being 9 orders of magnitude higher than measurements (in the case of scavenging Kxs), or exact opposite to the trend in observations (in the case of benthic flux).*

Referee #2 brings up multiple points in this main criticism, each of which we respectfully disagree with.

It is incorrect that we disregarded the literature on observational datasets pertaining to the sources of the Nd cycle. Each parameter is given a prior distribution that penalizes the objective function for "less likely" parameter values. These priors were determined based on reasonable ranges and initial values (whose choice was based on the authors' expertise with ranges larger than our best estimates but included them) according to the automated procedure explained in detail in Section 2.4.1. We emphasize that these prior distributions are settings that can be modified in a few lines of code. We find that the observational datasets suggested by Referee #2 in their detailed comments all have inadequate global coverage and were thus deemed not suited as direct simple constraints in the objective function definition (Eq. (17)).

Some parameter values may be unrealistic. This is a feature of objective optimization that does not impose strong constraints on the parameters. We re-emphasize however that the priors of these parameters can be adjusted at will by GNOM users.

The claim that some parameter values are "9 orders of magnitude higher than measurements" is incorrect and based on a misunderstanding. If scavenging in GNOM v1.0 were truly 9 orders of magnitude too strong, with sources of the same order as other models, then our simulated [Nd] values would be 9 orders of magnitude too small, which they are not. Referee #2 is comparing our dimensionless $K_X$ values to $K_X$ values reported *without units* in the literature (even though they should have the dimension of an inverse concentration). Our $K_X$ values are dimensionless because we employed a normalization (which we failed to report) that non-dimensionalizes our $K_X$. We provide further detail on the scavenging rate constant below.

In response, we have removed the normalization in the GNOM code and now report $K_X$ values with units in Table 2.

Similarly, we respectfully disagree with the claim that the optimized parameter values are the "exact opposite to the trend in observations (in the case of benthic flux)". Although it is not the goal of this model-description paper to specifically address this question, we contest Referee #2's "trend in observations" claim, which is based on a small number (order 10) data points from a narrow set of locations. More information is provided in the dedicated comment on sedimentary fluxes below. We also feel it is important to mention that the initial optimization that we present in this paper indicates that benthic fluxes are *extremely* important, constituting 48% of the global Nd flux, the largest contribution in the model.

*2. Despite the "optimized" parameters being orders of magnitude different from observations and previous models, this study arrived at global integrated sources similar to previous studies, suggesting that the Nd source parameters can NOT be uniquely determined ONLY by optimizing the seawater Nd and isotopes data. Either that the parameterizations are incorrect, or that a proper inversion should include measurements of the Nd sources in the cost function. In this respect, proper evaluation of the parameter covariance is required before optimization.*

As explained in the previous response, the premise of this main criticism is incorrect. There is also some confusion about what the optimization procedure employed aims to do and what covariance is. While it is true that compensating parameters (e.g., sources and sinks) cannot be uniquely determined with only the water concentrations and isotope values, we re-emphasize that we impose weak prior constraints on these parameters to avoid this issue. Contrary to what seems to be Referee #2's understanding, our manuscript's goal is to present a model with an easily reproducible environment so that users can run their own experiments, not to report estimates of source parameter values. Every model is incorrect to some degree, and GNOM users are welcome to experiment with tighter or looser constraints on the parameters.

We disagree with Referee #2's suggestion that our parameterizations are incorrect or our inversion improper. Including additional constraints to the GNOM cost function is not a simple task when the observational data proposed has extremely limited global coverage, as is the case with the datasets suggested by Referee #2, as we expand upon below. We also think there is some confusion about what covariance means in this context. However, we reiterate that apart from the GEOTRACES data, the GNOM repository provides a self-contained environment to reproduce our results. GNOM users are free to include any additional constraint they deem important.

No changes to the manuscript in response to this general comment. (There are many changes in response to the detailed comments below.)

*3. The parameterizations for Nd sources often make unsubstantiated assumptions on secondary issues that we have no constraint on, while adopt overly simplistic, even incorrect, representations of the first- order variables that we do have fairly good constraints on. For example, in the riverine parametrization, the authors use the detail reconstruction of global river discharge, which is only a secondary parameter in the Nd flux, yet force the river Nd concentration, the first-order parameter, to be constant despite observations that showing it varying by 2~3 orders of magnitude. Another example is that when treating the sediment flux, a complicated and arbitrary parameterization was created for the differential release of Nd from certain lithology (of secondary important, even smaller than analytical uncertainty), while ignoring that the baseline sediment flux eNd (the first-order variable) map of Robinson has large uncertainties.*

We respectfully disagree with the generally disparaging comments about our parameterizations. Perhaps this arises from lingering confusion about what we have done.

The first example provided is illustrative. The GNOM has a fairly simple riverine parameterization, where riverine input of Nd is proportional to the product of *effective* [Nd] concentration (assumed globally constant, controlled by a single free parameter) and volumetric flow rate (prescribed by a global dataset). As shown in Eq. (12), It is straightforward that concentration and river discharge have symmetric roles in determining the influx of Nd. Thinking of riverine [Nd] as primary and river discharge as secondary is incorrect. If riverine [Nd] doubles, so does the Nd flux to the ocean. Same if the river discharge doubles. Additionally, our simple riverine Nd source reports an *effective* [Nd] (Table 2), that is *effective* in the sense that it implicitly includes estuary effects (such as strong local removal by scavenging), yielding the overall input of Nd into the open ocean. We note that without a good parameterization of estuary processes, observations of the real, upstream riverine [Nd], which are the ones reported by Referee #2 to vary by 3 orders of magnitude, are hardly useful for a global ocean model. Our simple riverine parameterization is thus based on reasonable assumptions and simplifications.

In response, we have added explanations on the globally constant riverine effective [Nd] assumption in the text. The detailed revisions are listed in the detailed comments below.

The second example also shows some confusion. We proposed a simple parameterization to increase sedimentary release for extreme $\varepsilon_{Nd}$ values (as has been done by hand in previous published models, e.g. Pöppelmeier et al. 2020, Paleoceanography and Paleoclimatology). The "map of Robinson" is the best current estimate of sedimentary $\varepsilon_{Nd}$ values available that covers the entire seafloor. Referee #2 fails to suggest a better implementation, a better dataset, how one would use the uncertainty of the $\varepsilon_{Nd}$, or how it might help.

No changes to the manuscript in response to this part of the comment.

*Overall, without proper considerations of the actual measurements of the Nd source parameters, I disagree that the black box optimization approach can give any meaningful insight into the Nd cycle compare to previous studies.*

We respectfully disagree with the premise that we did not properly consider measurements of the Nd source parameters. We reiterate that GNOM uses every source-parameter dataset that has reasonable global coverage, and parameter ranges are chosen with broadly realistic values in mind.

We also note that in every other existing Nd-cycling model, most parameters are chosen by hand, generally by copying the values of previous models. In that sense the GNOM is the first model that allows model parameters to be freely objectively adjusted, which is a clear improvement. Although the results of this optimization do not always provide realistic values, we reiterate that the goal of this paper is not to assert final parameter values. We hope to further refine model parameters in the future, and will present those scientific results separately from the description of the model in this manuscript.

We disagree with Referee #2's broad and bleak opinion, that our objective optimization of most of its ~40 parameters cannot give any new meaningful insight into the Nd cycle. We invite Referee #2 to use the GNOM themselves (which should take a few minutes given the GNOM reproducibility) and impose the parameter constraints they deem realistic or implement the parameterizations they believe are missing or incorrect (which might take a little longer).

No changes to the manuscript in response to this part of the comment.

*Following are my comments on the parameterizations of Nd sources, and optimization approach and some other issues.*

**Dust**

- *40 ppm Nd in dust is higher than that in PAAS (34 ppm) and UCC (27 ppm) (Rudnick and Gao, 2014). Why is this number chosen?*

It is correct that 40 ppm is higher than the selected references by Referee #2. The ranges of [Nd] in riverine sediments, aeolian dust, and other continental material from Goldstein et al. (1984) ranges from about 10 to 60 ppm, such that 40 ppm is not unreasonable in this context. Also, its exact value is not that important because only the soluble fraction of Nd, controlled by the solubility parameters, makes it into the ocean.

In response, we have added: "(which is within the 11.93 to 45.76 ppm range of atmospheric dust observations in Goldstein et al., 1984)".

- *There are observations of dust eNd and solubility (Goldstein et al., 1984; Robinson et al., 2021), which are disregarded. Asian dust on average have an eNd of ~ -10 (Chen and Li, 2011), the "optimized" value of -7.6 seems too high. Australia is mainly made of old cratonic rocks, how could it supply a dust source as high as -4.0? Similarly, it's hard to image that the North American source could be as radiogenic as -4.3. It's difficult to accept the optimized values based on our knowledge of local geology. The parameter range used for optimization seems too narrow to me for many regions.*

It is incorrect that we disregard the observations of dust $\varepsilon_{Nd}$ and solubility. Both references suggested by Referee #2 are already cited in our manuscript, including multiple references to the study by Goldstein et al. (1984) in the section describing the dust source. As indicated in Table 2, the ranges and initial values that determine the prior likelihood of each of these parameters were determined based on literature and our expertise, which includes that of Steven L. Goldstein, a co-author on this publication, and our intent was to allow the prior values to have large ranges that encompass the real values.

Concerning the specific values, we reiterate that it is not the goal of this manuscript to provide a definite estimate on these values. Additionally, it takes a single-line-of-code change to modify the acceptable ranges of these parameters if a GNOM user wanted to do so. Finally, regarding

the εNd value in Australia, the extensive work of De Deckker (2019) in Earth-Science Reviews suggests that not every region in Australia emits the same amount of dust. Lake Eyre Basin in Australia has εNd values of about −4 ‰, much higher than the older cratonic rocks from the continent (see Revel-Rolland et al. 2006 in EPSL). These regions seem to be major dust sources to the South Pacific according to recent work by Struve et al. (2020, Nature Communications).

We note that following Referee #2's suggestion that our ranges are "too narrow" by increasing the ranges of these parameters would only allow them to reach values further away from Referee #2's expected values.

No changes to the manuscript in response to this part of the comment.

- *The "optimized" (~20%, even up to 80%!) solubility is orders of magnitude higher than observations (~1%) (Greaves et al., 1994). Not surprise that this study gives a dust contribution to the global Nd budget orders of magnitude higher than previous models.*

The high solubility rates were already acknowledged in the manuscript (original submission lines 479–481), along with an explanation, and a statement regarding less skilled state estimates with significantly lower solubilities. We reiterate that it is not the goal of this manuscript to provide a definite estimate on these parameter values. We also reiterate that it takes a single-line-of-code change to modify the ranges of these parameters if a GNOM user wanted to do so.

No changes to the manuscript in response to this part of the comment.

- *L480. This is a misunderstanding of the dust flux in (Tachikawa et al., 2003). The "known" dust flux used by Tachikawa is only 2.8 Mmol/yr, like the GCM models. In fact, Tachikawa is the main source of dust flux in the GCM models cited, that's why they are similar. The extremely high 60 Mmol/yr number cited (I think this is a miss citation. It should be 42 Mmol/yr in Tachikawa, I don't find this 60 Mmol/yr number) is only "fictional", that Tachikawa suggested that if it is true then we can solve the "missing source" problem. But there's no evidence that it is true. Thus, it is safe to say the optimized dust flux in this study is one order of magnitude higher than all previous estimates.*

We acknowledge that we misreported Tachikawa et al. (2003)'s "exterior" flux as the dust flux. In their study, which uses a simple 10-box model, calculated exterior surface fluxes (aeolian, rivers) are lumped together. We note that the value of 60 Mmol/yr that Referee #2 could not find is reported in Table 3 as the global calculated exterior surface flux (given as $8.6 \times 10^9$ g/yr). We also note that although Tachikawa et al. reject the explanation that dust and river sources can solve their missing source problem, they explore elevated solubilities: "The proportion of soluble Nd in dust (α) is highly variable (2~50%) depending on estimation methods". Regardless, we agree with Referee #2 that the GNOM dust flux is one order of magnitude higher than previous estimates, as was already stated in our initial manuscript submission on the same line (480).

However, we reiterate that it is not the goal of this manuscript to provide a definite estimate on these parameter values. We also reiterate that it takes a single-line-of-code change to modify the ranges of these parameters if a GNOM user wanted to do so.

In response, we now report the 60 Mmol/yr as the "Total exterior surface flux calculated from their model, 90 % of which is "missing" compared to their estimate based on observations." in Table 1.

**Ash**

- *L180. Explosive eruptions mainly produce felsic/rhyolitic ash that can travel long distance. Mafic eruptions are generally effusive and the resulting ash doesn't travel far (Scudder et al., 2016).*

We thank Referee #2 for catching this. This sentence was intended to justify the enhanced reactivity of (glassy) volcanic ash, regardless of silica ($SiO_2$) content.

In response, we have removed the ending of the sentence that mentions mafic/felsic compositions (L189–191).

- *L181. I don't find any mention of volcanic ash in Chien. So where are the flux data from?*

We thank Referee #2 for catching this. Another reference was missing (Brahney et al., GBC, 2015; doi:10.1002/ 2015GB005137). The underlying atmospheric model (from which the flux data used by the GNOM is from) did include volcanic ash. The data are available from the link that was already in the code and data availability statement of our initial submission (http://www.geo.cornell.edu/eas/PeoplePlaces/Faculty/mahowald/dust/Chienetal2016/).

In response, we have added the Brahney et al. reference in the volcanic ash Section (L191–192) and in the data availability Section (L694). (Bibliographic entry L806–807.)

- *L190. Ash eNd is varied. See the global volcanic arc eNd data of (Kelemen et al., 2014). It seems the authors have chosen an initial value of +10, which already is the upper limit of volcanic materials. The "optimized" ash eNd of +13 is clearly unrealistic.*

We agree that our volcanic ash $\varepsilon_{Nd}$ value is higher than expected, as objectively optimized within the (0,15) range. The GNOM v1.0 parameter ranges are intentionally large to allow for more model freedom. GNOM users can easily restrict ranges as they see fit.

No change to the manuscript in response to this comment.

- *The "optimized" ash solubility of 76% is extreme. Using the ash leaching experiment of (Du et al., 2016) we can get a ash Nd solubility of ~1%. The argument that ash is more soluble than dust is somewhat an urban legend without clear evidence. In situ*

*observations also show no effect of a recent Icelandic ash fall on surface seawater eNd (Lambelet et al., 2016). Regardless, 76% seems unrealistic.*

Volcanic ash consists mainly of volcanic glass, which is extremely reactive in seawater, as shown by published studies (e.g. Gislason & Oelkers, GCA 2003). Following from these results, we certainly would expect volcanic-derived Nd to be added. We believe the community would agree that the Lambelet et al. (2016) results are not the final word on the impacts of volcanic ash on seawater. It is important to note that in the optimized GNOM v1.0, volcanic ash constitutes a mere 0.35 % of the total Nd source, which is due to the small quantity of volcanic ash deposited on the ocean surface compared to mineral dust.

In response, we have added a sentence acknowledging the high value of our optimized solubility for volcanic ash. (L525-527):
"The high optimized solubility of volcanic ash βvolc is also likely unrealistic, although the total contribution of volcanic ash is much smaller than mineral dust."

**Sediment**

- *The sediment source parametrization is completely unsubstantiated and published sediment flux data have been disregarded. I see no rationale to support the assumption that benthic flux at the deep ocean is zero, which is clearly contradicted by observations (Abbott et al., 2015; Du et al., 2020) that instead show the highest benthic flux is from the deep ocean. The recent study from the abyssal plain of the Pacific (even at ~5000 m) still show a benthic flux of 3.5 pmol/cm2/yr (Haley et al., 2021), similar to what's observed on the Oregon shelf. Based on the existing data (Du et al., 2020), benthic flux either doesn't change with depth, or increases with depth, opposite to the parameterization here. It has also been show that benthic Nd flux doesn't correlate with POC flux or bottom water O2 (Du et al., 2018).*

We respectfully disagree with this detailed comment. Our sediment source parameterization is more versatile than any previously published model and, contrary to Referee #2's claim, it does not prescribe a deep ocean source of zero. In fact, our sedimentary flux converges asymptotically with depth towards the value of the free parameter $\phi_\infty$, whose range is $(0,\infty)$. Fig. 4d shows a significant flux of Nd in the abyssal ocean (between (3000–5000) m) when globally integrated, due to the hypsometry of the seafloor. We note that it is entirely possible for our flux parameterization to match the profile estimated by Du et al. (2020) suggested by Referee #2. This was mentioned on lines 209–211 of the original submission. As shown in Fig. RR1, different parameter values can give significantly different profile shapes. However, it seems our objective optimization favors a rather different profile than the one Referee #2 wishes for.

[Figure]

**Fig RR1**. *The GNOM versatile sedimentary flux parameterization can replicate many flux profiles, including POM-like (left) or linearly-increasing-with depth profiles shapes (right). Values of the sedimentary Nd flux parameters ($\phi_0$, $\phi_\infty$, and $z_0$) are indicated with dashed lines and annotated in yellow circles.*

Referee #2 suggests that our results are in contradiction with the observational studies of Abbott et al. (2015) and Du et al. (2020), which consists of only 17 data points located on continental margins with extremely poor coverage on the global scale (see locations shown below in Fig. RR2). The lack of global coverage makes it impractical for use in objective optimization of a global model of Nd cycle such as the GNOM.

[Figure]

**Fig RR2**. *Locations of sediment flux observations used in Fig 1C of Du et al. (2020). Abbott et al. 2015 Geology (circles), Abbott 2019 (triangles), Elderfield and Sholkovitz 1987 (square), German and Elderfield 1989 (diamond), Haley and Klinkhammer 2003 (inverted triangles).*

Additionally, we find that the "bottom-up hypothesis" is not even strongly supported by the underlying local data, let alone established on the global scale, at least given currently available data. For instance, when we tried to reproduce Figure 1C from the work of Du et al. (2020) (see Fig. RR3) using the citations provided in that manuscript, we found a few discrepancies. Further, going back to their original publications, we found 2 data points that directly contradict their linearly increasing Nd flux with depth from the studies of Elderfield and Sholkovitz (1987) and German and Elderfield (1989). Elderfield and Sholkovitz (1987) report an Nd sedimentary flux of 50 pmol m$^{-2}$ yr$^{-1}$ at 15 m depth. German and Elderfield (1989) do not explicitly report a Nd flux, but given the Sm flux of 16 pmol cm$^{-2}$ yr$^{-1}$ at 219 m depth and a Nd:Sm ratio of about 4 (taking for example the Nd:Sm values of their Table 4), ignoring the slight difference in solubility between Nd and Sm, we estimate a Nd flux of about 50 pmol m$^{-2}$ yr$^{-1}$. Neither of these data points have made it into Figure 1C from Du et al. (2020) although they strongly disagree with their linear interpolation, as can be clearly seen on Fig. RR3 below.

[Figure]

**Fig RR3**. *Fig 1C from Du et al. (2020) with missing data points from Elderfield & Sholkovitz (1987) and German & Elderfield (1989) highlighted in red, along with Haley et al. (2021).*

We note that the Haley et al. (2021) reference suggested by Referee #2 is a conference abstract that has not passed through peer review, and is thus not properly citable and inadequate for our manuscript. Being a 2021 abstract reference also means it was unlikely to be caught in our literature review. Despite these caveats, if we add the deep (5000 m) Pacific value of 3.5 pmol cm$^{-2}$ yr$^{-1}$ suggested by Referee #2 in Fig. RR3 (deep red dot), it is also in direct contradiction with the linearly-increasing-with-depth claim backed by Du et al. (2020). We also note that even if we only use the data in Du et al. (2020), the correlation is not a strong one, and of the 6 points below 1000 meters that are plotted, three have low concentrations and 3 have high concentrations. Adding the Haley et al. (2021) point, the majority of points are low (4 vs 3).

Regarding correlations with POC flux or O$_2$, the text may have been confusing, but we want to emphasize that our parameterization **does not** prescribe a POC-like profile shape. The parameterization allows a range of depth-dependences, including no change with depth, approximately linear increases or decreases with depth, and nonlinear changes with depth (Fig. RR1).

In response, we have removed the mention of POM flux, as this was likely causing confusion. We have edited this section to read (L219–222):

"The rationale behind the parameterization of Eq. (7) is versatility. For $\varphi_\infty < \varphi_0$ and small $z_0$, the flux profile is larger near the surface and smaller in the deepest parts of the ocean, resembling, e.g., the magnitude of eddy kinetic energy, while for $\varphi_\infty > \varphi_0$ and large $z_0$, the sedimentary flux increases linearly with depth (as in, e.g., Du et al., 2020, Fig. 1C). The optimization only enforces weak direct constraints on these parameters, allowing for any such profile shape."

Finally, we reiterate that it is not the goal of our manuscript to make a definitive statement on the shape of the sedimentary flux profile. Although not reported in the manuscript, we tried to prescribe the linearly increasing flux of Fig 1C of Du et al. (2020), but the general fit to Nd and $\varepsilon_{Nd}$ was worse. Hence, instead of arbitrarily imposing a linearly-increasing flux profile of Du et al. (2020), we chose to allow our parameters the freedom to objectively determine an optimal shape for this profile, which happens to be in contradiction with Referee #2's expectation.

- *L210. If linear increasing is allowed, then z0 should be allowed to be negative, rather than forced to be positive.*

Referee #2 seems to be confused here. While our parameterization cannot allow a strictly linear profile shape — it is an exponential after all — there exist parameter values for which the sediment flux profile is almost linearly increasing with depth within the (0, 6000) m depth range of our model. Such an example is clearly illustrated in Fig. RR1's right panel. It is also incorrect that $z_0$ need to be negative for this to happen, as illustrated in the right panel of Fig. RR1 in which the quasi-linear flux profile increasing with depth uses $z_0$ = 5000m depth (which is a positive value).

See response to comment above and revised paragraph.

- *L219. This is another urban legend with unclear evidence.*

We respectfully disagree. There is no published, let alone established, global model of the sedimentary Nd flux. In our experience, the large interbasin differences in seawater $\varepsilon_{Nd}$ cannot be reproduced without larger inputs of extreme $\varepsilon_{Nd}$ values, which is the motivation for including a parameterization of enhanced Nd release as a function of $\varepsilon_{Nd}$. We note that our parameterization makes it more likely for the sedimentary source to play an important part in producing the highly radiogenic north Pacific seawater $\varepsilon_{Nd}$ values and the unradiogenic north Atlantic seawater $\varepsilon_{Nd}$ values. We also note that it is not the first time that a Nd-cycling model implements a sedimentary flux enhancement along the areas of extreme $\varepsilon_{Nd}$ values (see, e.g., Fig. 3d in Poppelmeier et al. 2020).

Referee #2 also fails to recognize that the GNOM model does not prescribe enhanced Nd release with extreme $\varepsilon_{Nd}$ values. Instead, the GNOM parameterization offers the freedom of turning this enhancement "on" or "off" depending on the parameter value $\alpha_a$. As expected, a

non-zero value of $\alpha_a$ (i.e., an enhanced Nd release turned "on") was obtained after our objective optimization in order to best fit the observational datasets of both [Nd] and $\varepsilon_{Nd}$ values.

In response, we have added a reference to the Poppelmeir et al. (2020) model in this section, which now reads (L232–240):
"This quadratic parameterization is motivated by the fact that extreme sedimentary $\varepsilon_{Nd}$ values are often associated with rather fresh, and thus reactive, detrital material. We emphasize that this enhancement can be turned "on" or "off" depending on the choice of parameters (*a* = 0 turns it off). However, maybe coincidentally, extremely high $\varepsilon_{Nd}$ values are generally associated with relatively young volcanic Nd that is more reactive and readily soluble (Lacan and Jeandel, 2005; Pearce et al., 2013; Wilson et al., 2013; Blaser et al., 2016, 2020) and previous model studies have resorted to different enhanced Nd release parameterizations to achieve a similar effect (see, e.g., Pöppelmeier et al., 2020). The same is not necessarily true for rocks with extremely low $\varepsilon_{Nd}$ values, however it so happens that much of the region around the Labrador Sea (Greenland and northern Canada) is currently, or was previously, glaciated, which has resulted in a large amount of fine-grained crystalline (and thus labile) detritus with extremely negative $\varepsilon_{Nd}$ (von Blanckenburg and Nägler, 2001)."

- *L226. Evidence? Check (Blaser et al., 2019) which estimated the benthic flux from freshly eroded Heinrich Layers. It doesn't seem there is reason to believe it to be much higher than elsewhere.*

Quoting from Blaser et al. (2019), p.91, Section 4.3.2:
> "We provided evidence against significant vertical diffusion of REEs within the investigated sediment columns, but Nd may still have been released directly from the detrital layers into the bottom water while they were exposed at the sediment-water interface. Furthermore, both the deposition of turbidite Tu1 and of the IRD layers were likely accompanied by large amounts of fine particles slowly settling through the water column. In the case of the turbidite such particles probably formed a benthic nepheloid layer (McCave, 1986), which likely led to high exchange rates of Nd between detritus and bottom water. In the case of the HLs, flocculent material powdered by grinding of the ice sheets, so called 'glacial flour', contributed the major part of the total detrital load of icebergs and eventually sank to the deep ocean (Hesse and Khodabakhsh, 2016). Such fine material may have led to overprinting of the distinct Nd isotope signatures of local water masses, as suggested by Roberts and Piotrowski (2015). Up to now, the effect of such large short to medium-term events on large scale seawater Nd isotope signatures has not been investigated systematically."

As with the other parts of the benthic flux parameterization, the parameter $\alpha_{GRL}$ is optimizable, and the model is free to set this value at 1, meaning that Greenland-adjacent sediments do not behave any differently than other sediments.

No changes to the manuscript in response to this comment.

● *L240. This parameterization seems unnecessary if the resulting difference is only 0.03 epsilon, given typical analytical uncertainty of ~0.4 epsilon, and the reproducibility of sediment digestion/leach is ~1 epsilon or more. The uncertainty of the gridded product of Robinson is probably more than 5 epsilons because of the poor spatial coverage of the raw data and the inconsistency in the types of raw data (leach residual, bulk sediment, size fractions etc.). Thus, the biggest uncertainty to capture is that of the Robinson dataset, which should be allowed to vary 5~10 epsilon in the model.*

We agree that the shift in $\varepsilon_{Nd}$ is negligibly small, especially considering the analytical uncertainties. It therefore doesn't make much of a difference to the final model, and any user of GNOM could easily take it out entirely.

We object to the characterization of the Robinson et al. (2021) dataset as having "poor spatial coverage", as it is the most extensive and complete global compilation of Nd isotope data to date. To say that the Robinson compilation has poor data coverage, with the number of discrete measurements comprising the full dataset $n = 5,107$ with $n = 1,479$ defining the seafloor sediment map (Fig. RR4), while saying that the direct benthic flux measurements ($n = 17$, coming almost exclusively from the northeast Pacific; Fig. RR2) are complete enough to extrapolate globally is unreasonable.

[Figure]

**Fig RR4**. *Fig 3B from Robinson et al. (2021) showing locations of 1,479 measurements to define seafloor sediment εNd values.*

No changes to the manuscript in response to this comment.

**River**

- *I disagree with the parameterization of riverine input. In my option the necessary source data should come from: (Bayon et al., 2015; Goldstein and Jacobsen, 1987, 1988a, b; Goldstein et al., 1984). I would rather use the measured riverine Nd source, however limited, than using clearly incorrect parameterization, however globally detailed. This is an example of ignoring the first order issue, i.e., the variations in riverine Nd concentrations, while overly concerned with the secondary issue, i.e., the river discharge.*

To empirically determine the Nd flux to the ocean, it is necessary to know three things: the Nd concentration in a river, the flux of water from that river to the ocean, and the fraction of Nd removed in estuaries. All three of these are multiplied to calculate the Nd flux, so it is incorrect to say that river flux is secondary to Nd concentration in the river (or Nd removed in the estuary). Since there are global compilations of river discharge, we can account for this variability between river systems. To deal with Nd removal in estuaries, we calculate an *effective* Nd concentration in rivers—i.e., equal to the product of the true Nd concentration and 1 minus the amount removed in estuaries. It is a simplification to optimize this effective concentration as a single number, but to incorporate measured Nd concentrations from rivers, we would need values from all the major global rivers, along with calculated removal rates within each estuary. Even the most recent compilation of Bayon et al. (2015) only has Nd data from 22 rivers, ~10% of the total number of rivers included in the model, although they arguably carry about 70% of the total water flux.

Finally, as we have said before, the intent of this paper is to describe a new and flexible model rather than to provide final answers. GNOM users could add variation in river Nd concentration if they wish to do so.

In response, we have clarified that the riverine [Nd] that we optimize is an *effective* concentration rather than a measurable concentration and now say (L268–273):
"As a simplification, and to reduce the total number of free parameters in the model, we assume that all rivers share the same Nd concentration $c_{river}$, which is the parameter that controls the global riverine source magnitude (see Table 2). (Future improvements of the GNOM could include optimizable [Nd] parameters for each individual major river, constrained by ranges based on observations.) Because the GNOM v1.0 does not resolve estuary removal processes, our $c_{river}$ is to be understood as an effective Nd concentration that implicitly accounts for Nd removal in estuaries and is thus the concentration that makes it into the ocean."

● *L249. Riverine Nd concentrations vary by 2 orders of magnitude (Goldstein and Jacobsen, 1987).*

See response above. We have edited this sentence so it is clear that the assumption of uniform effective Nd concentration in rivers is a deliberate simplification to reduce free parameters in the model.

● *L255. How is the smoothing done? What is the spatial scale? The effect of riverine input doesn't extend beyond the estuary. What is in fig2d seems too extreme to be true. Where is the estuary removal term in the parametrization?*

As we state in the original manuscript, smoothing prevents numerical noise in the model that occurs as a result of a point source of Nd entering into a single grid cell (and causing large spatial gradients). We solve this numerical issue by artificially spreading the source over the neighboring grid cells. The spatial scale is thus on the order of the resolution, i.e., 2°×2° (or one pixel "cross" in Fig. 2d, which is hardly extreme). As we detail above, we are optimizing an effective river concentration, which accounts for estuary removal.

In response, we have added that the smoothing is accomplished "by spreading it over neighboring grid boxes" (L273).

● *L256. I disagree with the use Robinson data for this purpose. Remember the margin eNd of Robinson is an interpolated product based on rough geological map with very limited outcrop rock eNd, which has nothing to do with rivers. The relationship between riverine eNd and the eNd of the rocks from the drainage basin is a complicated matter (Bayon et al., 2015). There are measurements of riverine dissolved and particulate eNd from (Bayon et al., 2015; Goldstein and Jacobsen, 1987, 1988a, b; Goldstein et al., 1984) that should be used.*

This is again a simplification, but similar to what has been used in other Nd models with the less up-to-date Jeandel et al. (2007) compilation. Using additional data to estimate riverine $\varepsilon_{Nd}$ values is something we hope to explore in future work, and indeed something that GNOM users could improve upon as well.

As we mentioned above, the issue with using real data is that we need to have measurements for all the rivers in our compilation ($n$ = 200). A future user could amend the model code to only include rivers where Nd concentration has been measured.

No changes to the manuscript in response to this comment.

**Groundwater**

● *Same problems as the riverine parameterization: groundwater Nd concertation is not constant, and it varies by 7 orders of magnitude! I see no reason to use the Robinson*

*eNd dataset either. We already have the measured Nd flux and eNd from (Johannesson and Burdige, 2007), why would you use parameterizations that's unsubstantiated.*

Again, the issue is having this data in a format that allows it to be mapped globally. The underlying data from Johannesson and Burdige (2007) are not readily available. Additionally, in this paper the authors use a model to extrapolate fluxes of REEs to the ocean, acknowledging that submarine groundwater discharge, along with its Nd concentrations and isotope values, are not well known globally. They are able to calculate fluxes (with associated concentrations and isotope values) into each basin, but to incorporate this into our model, we would need to know where to put that flux, a nontrivial task! In order to implement the Johannesson and Burdige data into the model, we would need a data product closer to what is provided in Robinson et al. (2021), with a map that can be incorporated into the model.

No changes to the manuscript in response to this comment.

- *What is the removal parameterization of the groundwater flux upon entering the ocean?*

Just like for rivers, the concentration in groundwater is the *effective* concentration that makes it into the ocean.

In response, we now say (L287–289):
"Similarly to the riverine sources, we assume that [Nd] is constant across river watersheds and implicitly accounts for local Nd removal processes. The single parameter $c_{gw}$ is thus the *effective* groundwater concentration that makes it into the ocean (...)"

**Hydrothermal**

- *Hydrothermal system is a net SINK of Nd (Stichel et al., 2018; Basak et al., 2021). The release of hydrothermal Nd makes local seawater eNd only slightly more radiogenic (within analytical uncertainty), while the seawater Nd concentration decreases significantly because of scavenging by hydrothermal particles. Therefore, both a source and a sink should be implemented for the hydrothermal system, with the sink being much larger than the source.*

To the best of our knowledge, there is no established global climatological 3D field for hydrothermal particles that we could use to simulate hydrothermal scavenging of Nd. While we agree with Referee #2 that hydrothermal systems should behave as a net sink of Nd, we believe such an implementation is an improvement for future versions of the GNOM. For instance, it is possible — although not trivial — to implement an off-line model of non-buoyant particles that are transported by the circulation while settling at a specified velocity and then use this model to quantify extra scavenging sites (like we did with the Si-cycle to get opal concentrations).

Importantly, in our optimized model, the hydrothermal source accounts for only about 1 % of the global Nd source, which is in accord with the leading view that hydrothermal vents only play a minor role in the Nd-cycle of the modern ocean.

We note that the Basak et al. (2021) reference suggested by Referee #2 is yet another non-peer-reviewed conference abstract that is not properly citable in our manuscript.

In response, we have added the following paragraph (L309–312):
"Arguably, the hydrothermal system as a whole acts as a net sink of Nd in the ocean (Stichel et al., 2018). As described in Section 2.3, the GNOM v1.0 does not include a parameterization of scavenging due to hydrothermal particles. Future versions of the GNOM should attempt to include such a removal process in order to properly balance the hydrothermal source and allow the $\varepsilon_{Nd}$ signature to be modified along hydrothermal vents without increasing the [Nd] concentration at the same time."

**Scavenging**

- *L316. I disagree. For some reason the most important Nd scavengers, Fe-Mn oxides (Schijf et al., 2015; Sholkovitz et al., 1994), are ignored like previous models. And I don't see how scavenging by Fe-Mn oxides can be treated as "precipitation" as it will depend on oxide concentrations that are redox sensitive.*

The reason that Fe-Mn oxides have been ignored in the past is likely that, to our knowledge, there is no publicly available 3D field of the modern climatological concentration of particulate Fe-Mn oxides that modelers could use. Line 316 refers to a statement that our implementation of a background scavenging rate (that we have called "precipitation" simply because it is implemented in the same way as precipitation would be) *could* compensate for missing scavenging particle types. This is correct. We were not suggesting that Fe-Mn oxides should be treated as "precipitation". This mechanism is simply an effort to account for particle types for which we do not have global 3D fields. In the future, as they become available, more particle types could be added.

In response, we now say (L346–350):
"We note that while this additional particle-independent scavenging sink could compensate for additional types of particles not currently implemented in the model, it is likely that more scavenging particle types are required for an accurate representation of the Nd cycle. These include hydrothermal particles (which should result in hydrothermal systems being a net sink; Stichel et al., 2018) and iron–manganese oxides (which are potentially the most important scavenging particles; Schijf et al., 2015, Sholkovitz et al., 1994)."

- *The "optimized" Kxs are unrealistic. Previous models generally use values around 10^6 (Siddall et al., 2008; Arsouze et al., 2009). Lab measurements also give results on similar orders of magnitude (Schijf et al., 2015). Yet here the optimized values are 10^13~10^15!*

As per our response to the main point referencing a 9-order-of-magnitude difference, each study suggested by Referee #2 in this detailed comment fails to report a unit for their $K_X$. Our $K_X$ values were purposefully reported without units because they were normalized such that they were dimensionless. This was mistakenly not reported and is corrected in the revised version of the manuscript.

In response, we have removed the normalization and now report $K_X$ values with their units in Table 2.

- *I think that such high optimized Kxs are the results of overly high dust flux in the model (see my comments up), so extremely intense scavenging is needed to bring surface Nd concentration down.*

This is incorrect both on the premise and on the conclusion. Our dust source of Nd, which is an order of magnitude larger than previous models, would be *entirely dominated* by a scavenging rate too large by 9 orders of magnitude. This would leave almost no Nd in the surface, which is not the case for the optimized GNOM v1.0. As we say in our response to the comment above, the apparent discrepancy between our $K_X$ values and those reported in previous model papers is a matter of unit conversion (and units not being carefully reported).

As mentioned above, we have removed the normalization and now report $K_X$ values with their units in Table 2.

**Optimization**

- *There are so many parameters to be optimized. What is the covariance structure of them? For example, the dust solubility will probably covary with the scavenging Kxs. Therefore, can we get a unique solution of the parameter values? These questions need to be studied before performing optimization. That so many of the optimized parameters are unrealistic give me little confidence in the optimization.*

Referee #2 seems confused about model parameter covariance in this context. The model parameters are assumed independent in the sense that their prior distributions are independent. There is no physical reason for parameters such as dust solubility and the scavenging rate constant to be dependent on each other. However, it is correct that some parameters can compensate for each other in determining the distribution of [Nd] and $\varepsilon_{Nd}$ (such as dust solubilities and scavenging constants, which tend to have opposing effects on [Nd]). In a statistical Bayesian framework, this could be interpreted as the consequence of non-zero off-diagonal entries in the covariance matrix of the posterior likelihood evaluated at the maximum likelihood estimate of parameters. This covariance matrix is directly related to the sensitivity of the objective function to the parameter values: The Hessian matrix of the objective function (the matrix of the second order partial derivatives, which is computed as part of the objective function minimization algorithm) generally has nonzero off-diagonal entries when

evaluated at the optimal estimate of parameters—meaning parameters have compensating effects on the cost function. As such, a Bayesian parameter inference model *could be* built on top of GNOM to determine the posterior likelihood of its parameters given the priors chosen. These posteriors would be covaried (i.e., with nonzero off-diagonal entries in the Hessian matrix). While such a statistical model is within the scope of future versions of GNOM, this non-trivial work is out of the scope of GNOM v1.0 and our submitted manuscript.

The uniqueness of optimal parameters was already addressed in our original submission (l. 400), where we briefly mentioned that the algorithm employed is that of local optimization as opposed to global optimization. We note that global optimization is — in our limited experience — a prohibitively harder optimization problem unsuited for our model and its ~40 parameters. Hence, it is true that there is no guarantee that we have a unique solution or that we found the singular global optimal parameter set (as already mentioned in the manuscript), but that is not the goal of the GNOM anyway. It is important to note that the GNOM v1.0 contrasts with previous marine Nd-cycle models embedded in GCMs in allowing optimization of all parameters in the first place. GCM model parameters are generally fixed and assumed to be known a priori, rather than allowed to vary within specified ranges, because of the computational requirements for running these models. We are thus confident that our approach is sound for estimating/tuning the Nd-cycling parameters. We reiterate that future users of the GNOM are free to refine the current parameter ranges in order to improve their estimations as they see fit.

No changes have been made in response to this comment, although we note that the relevant Fig. B1 has been edited in response to other comments to show initial and final optimized parameter values.

- *L335. It is nice to include penalty terms for the parameters, but I think the choice of parameter ranges are not very reasonable give that the optimized values are often unrealistic.*

As we have said, the point of our model-description paper is to introduce a new model, describe what we have done, and share it (including all the code and datasets required to run it) for the community to use as they see fit. We are **not** saying that we have found the correct values for all parameters (even though we have chosen what we believe are broad but believable ranges, and our model seems to fit observations quite well). If the Reviewer, or other model users, think the parameter ranges are unrealistic we hope they will change the values and re-run the model. This is very simple to do (a few lines of code change, barely any setup required) and takes very little computational resources because the GNOM is highly efficient.

We realized that our Fig. B1 mistakenly showed a different optimized parameter set than intended. We have corrected that in the revised manuscript and now show a dozen of initial and final parameter values shown in transparent black and the selected "best" estimate parameter values shown in blue.

- *The optimization should be penalized against not just measured seawater Nd/eNd, but also measurements of Nd sources.*

Given the lack of coverage/availability of direct Nd source observations, this is an unreasonable request.

Our objective function, Eq. (17) in the manuscript, has three terms: the mismatch between modeled and measured Nd concentrations, the mismatch between modeled and measured $\varepsilon_{Nd}$ values, and a parameter penalty term that penalizes less likely parameter values to keep them within specified ranges. Each of the three terms is weighted, with the Nd and $\varepsilon_{Nd}$ costs weighted more than the parameters. Importantly, having no direct constraint on the 2D source fields means that they are objectively determined. Adding a strong source-magnitude constraint means the resulting simulated source magnitude would be "baked in", which is not the goal here.

Were we able to find a reasonable dataset to use in our cost function, one would still need a concrete and reasonable formulation for that term and to assign it a reasonable weight. None of these tasks are trivial. While sources have been measured at a few locations, *all studies* that make direct measurements must extrapolate to determine global sources and no study provides a global map. Adding a cost term that uses a global extrapolation from a few datapoints is equivalent to implicitly assuming that the extrapolation is valid in the first place.

Thus, based on our experience, the current GNOM approach makes more sense. That is, we give the model extended freedom and let the optimization output serve as a target for real world measurements and future model improvements — in particular where the output challenges the expected outcomes. If Referee #2, or any other reader, thinks a modeled value or source at a particular location is unrealistic, they could easily test that hypothesis and then revise the model so it is more realistic. The GNOM framework is ideally suited for such hypothesis-test experiments.

No changes have been made in response to this comment.

- *A volumetric weighing should be included in the cost function, so the optimized results are not biased toward the surface ocean because of high data density. The Nd community is generally more concerned about the deep ocean than the surface, in contrast to other communities, for several reasons: 1 Surface eNd is heterogenous and local, thus contains little global information. 2. Surface input generally varies a lot in the past, so surface eNd isn't useful for watermass tracing. 3. We can only reconstruct deep ocean eNd in the past, as there is no archive of surface water eNd.*

We are well aware that the Nd community is primarily concerned with deep ocean processes. However, our goal with the GNOM model is to describe the *whole* marine Nd system, not the deep circulation only or paleoceanography. Much of the Nd enters the ocean at the surface, so it is impossible to build a reasonable whole-ocean model while disregarding surface data. That

said, our model matches deep ocean values well, and we hope that it will be a useful tool for tracing deep ocean circulation processes.

This request for volumetric weighing is another instance where the practical implementation is much more difficult than the concept itself. It is true that seawater measurements are unevenly distributed globally and any GNOM user is welcome to alter the cost function weights in any way they see fit. There are many ways to do so, although it is unclear which of them would improve the model. For example, one could selectively sub-sample the data, add a weight that increases with depth, remove quasi-duplicates, attempt to correct seasonality, or add statistical noise. Importantly, using raw data (as is done currently by the GNOM) is reasonable because heavily sampled locations likely average to a more accurate value (and are thus more reliable) than poorly sampled locations with isolated measurements, which are less reliable for lack of reproduced values.

In the case of sparse seawater data (as is the case for Nd and $\epsilon_{Nd}$), implementing a volume-weighted mismatch is a risky endeavour with many ptifalls. To our knowledge, there are only two somewhat valid approaches to assign volume weights to data. (i) Extrapolate onto a 3D grid that has its own box volumes or (ii) directly assign weights. (i) Extrapolation is "hard" when the data coverage is bad (compared to, e.g., the gridded products from the World Ocean Atlas). Extrapolation onto a global 3D gridded field can still be done but at the expense of significant processing and assumptions, e.g., using a neural network (as has been recently done for Cd (Roshan and DeVries, 2021; 10.1029/2021GB006952)) or using advanced extrapolation routines for sparse and irregularly spaced data such as the DIVAnd software (https://github.com/gher-ulg/DIVAnd.jl, the workhorse behind the Ocean Data View software (https://odv.awi.de/)). Extrapolation adds the issue that the uncertainty of the extrapolation is then built in the model and is not trivial to account for. One goal of the GEOTRACES program is to develop more complete tracer data fields that would allow for the construction of reliable gridded datasets (and would in turn allow for the volumetric weighting that Referee #2 requests) but, unfortunately, we are just not there yet. (ii) Directly assigning weights also requires some assumptions and uncertainties to account for. One could, for example, use an approximate 3D Voronoi mesh around the locations of observations, which is not straightforward. The risk is then that isolated measurements are heavily weighted although they might be seen as less reliable for lack of colocated duplicates. Thus, regardless of the approach chosen, assigning volume weights is neither straightforward or guaranteed to improve the model skill.

No changes have been made in response to this comment.

- *Additionally, basinal weighing could be useful given that the data is so concentrated in the Atlantic.*

As per our response above, this is just one out of an innumerable number of alternative cost function implementations that GNOM users are welcome to try. We note that while this could reduce sampling location bias, it could also potentially exacerbate biases or systematic errors

that could have remained hidden because of the low sampling numbers and lack of colocated duplicates in other basins.

No changes have been made in response to this comment.

- *How does the choice of the prior affect the results? For many parameters we know very little but the orders of magnitude. Isn't a uniform prior more reasonable than lognormal/logit priors when we essentially have no information on the actual distributions?*

As already explained in detail in the manuscript, the priors help the convergence of the optimization and prevent it from reaching extreme parameter values. We chose the 3 types of distributions that we deemed the most natural choices depending on the physical ranges of the parameters, which are either:
- infinite $(-\infty, +\infty)$: Normal distribution
- "semi-inifinite" $(0, +\infty)$: LogNormal distribution
- finite $(a, b)$: LogitNormal distribution

The added advantage of these choices is that in transformed-parameter space, the log-likelihood penalties become quadratics, which is beneficial to the convergence of the Newton optimization algorithm because it uses Hessian (second differential) information.

We note that there is no such thing as a uniform prior over a non-finite interval as requested by Referee #2. We also note that a uniform prior would result in zero added constraint, such that one might as well disregard the prior penalty altogether in this case. It is precisely the non-uniform shape of the prior that makes the prior-penalty implementation worthwhile.

Finally, we reiterate that the GNOM users are free to improve the model by including their own prior estimates of parameters (be it to include a stronger or weaker constraint).

No changes to the manuscript in response to this comment.

- *L400. This needs to be demonstrated. And are the ranges of the randomized initial values large enough that we can really believe the results are global?*

The ranges of the possible initial parameter values are exactly the ranges of the parameters as reported in Table 2 (which are broad enough that they are characterized as "unrealistic" by Referee #2 throughout this review). While not reported individually in our original submission, each randomized initial value is sampled from the prior distributions of which the probability density functions are illustrated in Fig. B1.

It is unclear what "this" on l.400 refers to, although we assume it refers to "due to the likely many local minima", which is correct: Like most optimization problems, it is likely that there are multiple local minima, any of which our optimization runs could end in. Referee #2 seems to suggest that we somehow claim that our optimized solution is global. This is the exact opposite

of what the referenced paragraph states (specifically: that we can only expect to find local minima).

We emphasize that our manuscript is not a study on the optimization trajectories of parameter sets or an attempt to define the regions of initial conditions that converge to a given optimal solution. Rather, it is a model-description paper that includes a sound optimization routine and is easily extensible and reproducible. We leave global minimum proofs to theoretical mathematicians.

In response, we have edited the paragraph and added a display of the initial and final optimized parameter values for a dozen of optimization runs in Fig. B1.

**Minor points**

- *The unit of eNd is epsilon, not per 10 thousand. This is a radiogenic isotopic ratio, not a stable isotopic ratio. The expression of epsilon looks like delta for stable isotopes, but the reporting convention is different. I have never seen the per 10 thousand unit for eNd in literature.*

While this notation point is unimportant for our manuscript, we respectfully disagree with Referee #2—the fact that Nd isotopes are radiogenic is irrelevant. The important part is that "epsilon units" are just the arbitrarily preferred literature alias for "parts-per-ten-thousand" for isotope ratios that vary only in the 4th or 5th decimal place. The relative deviation of the isotopic ratio of a sample from a reference isotopic ratio is dimensionless. It can be quantified without units (because it is by definition dimensionless) although for $^{143}Nd/^{144}Nd$ the range of natural values on Earth are more easily expressed in ‰ (parts per ten thousand). Eq. (1) relates these quantities in a coherent manner and is universally true, regardless of the units used. While we acknowledge that there are legitimate inconsistencies between how epsilon values (and delta values) are reported, compared to percents (Coplen, 2011; 10.1002/rcm.5129), it is not our goal to change how the field reports Nd isotope ratios. We have therefore deliberately chosen to stick with the established use of the ε symbol but to use the correct equations and units (i.e. show Eq. (1) without the extraneous 10,000 factor and report $\varepsilon_{Nd}$ values in parts-per-ten-thousands using the ‰ symbol).

No change to the manuscript in response to this comment.

- *L129. Not because there's no stable isotopic fractionation, but because by convention when reporting radiogenic Nd isotopes stable fractionation effect, be it natural or instrumental, is removed by normalizing to constant 146Nd/144Nd ratio. That is, even if you model isotopic fractionation, you have to remove it when you convert to eNd units.*

Apologies, our intention was to say that any stable isotope fractionation upon scavenging is insignificant compared to the effects of radioactive decay on the $^{143}Nd/^{144}Nd$ ratio. Yes, of

course, mass fractionation during mass spectrometry is corrected by normalizing to the $^{146}Nd/^{144}Nd$ ratio, as Referee #2 says.

We have edited this sentence so it is accurate (L134–136):
"We omit stable isotope fractionation during scavenging because its effect is negligible compared to the effect of radioactive decay from $^{147}Sm$. Thus, in Eq. (2), only the external sources $s_k$ differ in their isotopic composition."

- *L408. Needs demonstration.*

Line 408 of the initial submission states: "In all likelihood, there exist other models and other parameter choices which produce a similar fit to global observations…". The intention of this sentence is to acknowledge that we do not believe we have created a model that is perfectly correct, and we don't think we have completely solved the marine Nd isotope cycle with the model we present here. This referee comment seems to miss the intention of the statement.

No change to the manuscript has been made in response to this comment.

- *Fig9/10. It's very difficult to see the "fit". Instead show the model-date misfit rather than just overlay the data on top of the model.*

This is exactly the point. The data and the model match well enough that it is difficult to distinguish between the colors. We note that because the observational data is sparse, it is not possible to produce a heatmap or a filled contour plot of the model–observation mismatch without performing an interpolation/extrapolation of the observations, which would obfuscate the comparison. For completeness, below, Figs. RR5 and RR6 reproduce our manuscript's Fig. 9 and 10 but with the scatter of observations replaced by model minus observations data. However, we believe that these partially repetitive figures are better left out of the manuscript.

No change to the manuscript in response to this comment.

[Figure]

**Fig RR5**. *Original manuscript submission Fig. 9 with observations replaced by model–observation mismatch.*

[Figure]

**Fig RR6**. *Original manuscript submission Fig.10 with observations replaced by model–observation mismatch.*

- *L484. If the optimized solubility is not supported by observations, then there is no reason to believe it.*

This is the same point made earlier about solubilities. There are relatively few direct observations of dust solubility, especially compared to the global distribution of aeolian input. Previous studies have typically chosen a 2% dissolution rate based on the 1994 study of Greaves et al. that used a single aerosol sample (likely of Saharan origin) and 4 experimental treatments. Referee #2 or any other GNOM user could easily change the dissolution parameter ranges as they see fit.

No change to the manuscript has been made in response to this comment.

- *L487. I disagree with this philosophy of "consistent with previous model" but ignoring actual observations of the fluxes (Du et al., 2020; Goldstein and Jacobsen, 1987; Tachikawa et al., 2003; Johannesson and Burdige, 2007).*

See detailed responses above. In nearly every case, the observations are limited spatially making them difficult to directly implement in the model. This is especially true for direct observations of benthic Nd fluxes (e.g., Fig. RR2 and RR3), but also true for riverine and groundwater fluxes as we describe above. We add this comment to note that our values in large part agree with estimations from previous models, which gives us confidence that our model is reasonable. This is a model description paper, so we are not saying that these values are perfectly correct and there is clearly more work to do. We hope to continue to improve the model in the future, and we hope others will as well.

No change to the manuscript has been made in response to this comment.

- *L522. Seem exaggeration. What about the MIT-ECCO, which has also been used as TMM?*

We have clarified this sentence to focus on steady-state models. Concerning the MIT-ECCO product, we refer to Devries et al. (2014): "*More recently, the Estimating the Circulation and Climate of the Ocean (ECCO) consortium assimilated global temperature and salinity data into a three-dimensional OGCM, which was then used to simulate the uptake and transport of anthropogenic CO2 by the ocean [Graven et al., 2012; Khatiwala et al., 2013]. A major shortcoming of the ECCO model is that ventilation tracers such as Δ14C and chlorofluorocarbons (CFCs) were not assimilated. It has been shown that even models that reproduce temperature and salinity distributions well may still have deficiencies in ventilation that can best be identified by simulating Δ14C and CFCs [England and Maier-Reimer, 2001]*".

We have clarified this sentence to focus on steady-state models in response to this comment.

- *L545. Perhaps not exactly the same calculation, but similar concept of partitioning N/S sources already exist (Gu et al., 2020)*

We thank Referee #2 for pointing us to this part of the work of Gu et al. (2020) that should have been referenced there. We note that our approach offers a number of advantages over that of Gu et al. (2020). First, the GNOM and these diagnostics take a few seconds only to run. Second, the OCIM2 circulation offers a more accurate representation of deep ocean circulation owing to its data-assimilation procedure (see, e.g., DeVries and Holzer, 2019). Third, our approach can be used to replace their 2-end-member mixing estimate of water masses by any *n*-end-member mixing. That is, using the GNOM one can conservatively propagate prescribed $\varepsilon_{Nd}$ from any number of grid boxes. Fourth, one can similarly propagate [Nd] rather than $\varepsilon_{Nd}$, and additionally control if said propagation is conservative or includes scavenging/or external

sources. This is important in face of the caveat from the Gu et al. (2020) approach in their water-mass calculations as inferred from $\varepsilon_{Nd}$ values without [Nd] weighting. Finally, the GNOM offers the ability for delving into further detail with ease, including quantifying timescales.

In response, we have added (L600–604):
"Gu et al. (2020) performed a North–South end-member partitioning using the CESM model and its POP2 circulation. They quantified water-mass fractions using dye injections at the surface and compared them with water-mass reconstructions from deep $\varepsilon_{Nd}$ values. Here we present more detailed partitions that have never been estimated in previous modelling studies to our knowledge."

- *L571. The most prominent caveat is the disregard of observations of Nd sources.*

As we have explained in great detail throughout this response, we did *not* disregard observations of Nd sources. We do, however, allow broad ranges of parameter values to give the model freedom in its optimization—these are easily changed by whoever wants to run the model as they see fit.

No change to the manuscript has been made in response to this comment.

- *L543. Disagree, parameter choices often unrealistic.*

We are not entirely sure what Referee #2 disagrees with here, but we think they object to the phrase "A clear signal of the influence of surface sources is visible down to about 1500 m…", and are arguing that our aeolian source is too large to trust this implied surface source influence. We have defended our parameter choices at length throughout this response, and if Referee #2 or any other user of the model wishes to change the parameter ranges to values that they feel are more realistic, and run the same diagnostics, this is very easy to do.

No change to the manuscript has been made in response to this comment.

**Reply to RC5: 'Comment on gmd-2021-338' by Referee #3**

Below are Referee #3 comments reproduced in grey italic and our point-by-point replies in black.

*Pasquier and colleagues describe a new inverse model of oceanic neodymium (Nd) and its isotopes (eNd) that attempts to resolve their geochemistry based on optimization of 43 free parameters and a prescribed ocean circulation (OCIM v2.0). The result of the (optimized?) model are compared to published data the authors have collated with laudable success. This ms. is written as a model description so there are only a few conclusions drawn, and these are somewhat guarded (which seems reasonable given the initial nature of the work).*

*The paper is well written, and I was only lost in the mathematics in a few instances. An honest modeler (not a geochemist such as I) should review this work for the mechanics of the more complicated mathematics (see notes below).*

We thank Referee #3 for their positive comments and have replied to each comment below.

*In general, this model does seem to hold potential, and I hope that the authors continue to move forward with it. Again, this ms. does not claim to definitively resolve outstanding problems of eNd-Nd in the oceans, so there is not too much to critique beyond the model itself. For this, I have a few comments:*

*L22-24: the Sm:Nd of the rock type plays a critical role in the eNd that results; equal to, if not more than, the age of the rocks.*

We thank Referee #3 for this comment. We agree that this needed clarification. It is correct that some rock types, such as basalts, have different Sm:Nd ratios than the ones we are using, which reflect continental values. In this context, a unique feature of the Sm–Nd system is that most continental rocks have the same Sm:Nd ratio, as described in DePaolo and Wasserburg (1976), McCulloch and Wasserburg (1978), and Goldstein et al. (1984).

In response, we have added the additional text below (L23–26):
"While the Sm:Nd ratio varies within the earth, these ratios are remarkably similar in most rocks in the continental crust and across the geological time scale, and about 40% lower than the bulk earth Sm:Nd (DePaolo and Wasserburg, 1976; McCulloch and Wasserburg, 1978; Goldstein et al., 1984), and as a result the $\varepsilon_{Nd}$ values in continental rocks generally directly reflect the average crustal age."

*L85-90: note that the internal cycling of eNd/Nd can be argued to be of secondary importance, when the budget of the entire ocean system is not yet resolved. That is, as described in Du's work, water column processes can only redistribute eNd/Nd, they cannot explain the source functions that are currently wanting.*

We are a little confused about this comment, but we have tried to add additional clarification to the text. The intention of this paragraph is to introduce the reader to another type of model that has been used to simulate Nd isotopes in the ocean. By definition this type of model (a "boundary propagation model") cannot provide information about the budget of the entire ocean system, as it cannot explicitly model external sources of Nd, nor internal cycling processes (such as reversible scavenging). However, when used as a tool to explicitly test the hypothesis that Nd isotope ratios are conservatively transported into the ocean interior, this is not a problem.

We have tried to make that point more clear in this paragraph by adding some additional text (L91–93):
"These models have been used to explicitly test the "conservativeness" of Nd isotopes as a tracer, since they do not incorporate external fluxes of Nd or internal cycling processes, and can thus only be used to simulate conservative transport."

*L104: what is a green function based diagnostic?*

Green functions (sometimes spelled Green's functions) can be used for solving ordinary differential equations with an initial condition and/or boundary values. In our case, they can be thought of as the [Nd] responses to unit local sources of Nd. The general solution (the global [Nd] field) is then equal to the sum of all the Green functions for each local source (see, e.g., Morse and Fechbach, 1953). Conversely, the linearity allows us to partition the solution into components of [Nd] or $\varepsilon_{Nd}$ of interest. These diagnostic partitions are based on Green functions. For example, this is how we calculate the global [Nd] field coming from a particular source type (hydrothermal, aeolian, etc.) using Eq. (19). Similarly, the diagnostics examples in Sections 3.3.3 and 3.3.4 are also Green-function based.

In response, we have added (L109–112):
"Green functions (sometimes spelled Green's functions) can be used for solving ordinary differential equations with an initial condition and/or boundary values (see, e.g., Morse et al., 1953). In our case, they can be thought of as the [Nd] responses to unit local sources of Nd and allow us to partition [Nd] or $\varepsilon_{Nd}$ into components of interest, such as Nd from a particular source or location."

*L135: why is circulation (Tcirc) a sparse matrix?*

The circulation matrix is sparse because water can only move between grid boxes that are adjacent, therefore there are many zeros in the matrix, which represent the inability of water to move directly from non-adjacent boxes (i.e. the surface of the Atlantic to the bottom of the Pacific).

We have added a brief explanation to the text (L145):
"(Most of the entries of $\mathbf{T}_{circ}$ are zero because water can only travel directly between neighboring grid cells.)"

*L275: hydrothermal vents are known to be fundamentally a sink term for Nd and also eNd. From early work (Klinkhammer, German) to recent (Chavagnac), the magnitudes of fluxes related to hydrothermalism have been found to be generally zero to negative. I do not understand why this is a source (it would make their model more simple too).*

We know that hydrothermal vents are not considered a substantial source of Nd to the ocean, primarily because of scavenging onto hydrothermal particulates (which we haven't modeled). Indeed the optimized model has a very small contribution from hydrothermal vents (~1%). However, vents are clearly part of the system and our aim with this model is to include a wide range of possible sources so that future model users can make their own decisions, including removing the hydrothermal source or explicitly representing hydrothermal particles so that dynamics at vents are modeled more realistically.

In response, we now say (L309–312):
"Arguably, the hydrothermal system as a whole acts as a net sink of Nd in the ocean (Stichel et al., 2018). As described in Section 2.3, the GNOM v1.0 does not include a parameterization of scavenging due to hydrothermal particles. Future versions of the GNOM should attempt to include such a removal process in order to properly balance the hydrothermal source and allow the $\varepsilon_{Nd}$ signature to be modified along hydrothermal vents without increasing the [Nd] concentration at the same time."

*L298-302: I did not understand this. It is important though.*

We have tried to clarify this paragraph by adding additional non-technical language that summarizes what we have done conceptually and describes the equations dictating particle scavenging in the model. We recognize that the implicit formulation is more difficult to understand than if we had explicitly modeled scavenged Nd.

In response, we have revised the paragraph which now reads (L326–333):
"However, we avoid the explicit simulation of scavenged neodymium, $X$Nd, by having a fraction of the dissolved neodymium pool sink to the box below as if it were adsorbed onto a falling particle. To do this in practice, we take advantage of the direct relationship between free and scavenged Nd, Eq. (15), assuming (R1) operates on shorter timescales than either vertical particulate transport or ocean transport. (This assumption is common in models that include scavenging and simpler than resolving the adsorption/desorption rates dynamically (e.g., van Hulten et al., 2018).) Since dissolved and scavenged Nd are in equilibrium, Eq. (15) uniquely determines [$X$Nd] = $K_X$ [$X$] [Nd] given the modelled [Nd] and the prescribed particle concentration [$X$] (from the four particle fields included in the GNOM v1.0, described below). Consequently, the corresponding partial downward flux of dissolved Nd is given by $w_X$ [$X$Nd] where $w_X$ is the settling velocity of particle $X$."

*Fig.5: the scavenging of Nd in the surface seems exceptionally high in all cases. Nd is low in the surface, certainly, but not so low as might be suggested by these implied removal rates.*

The apparently high scavenging at the surface is partly an artifact of the fact that scavenging always transports material downward–for grid boxes below the surface, they receive material from the box above and export material to the box below. This is not the case for the surface box, so the removal rate seems very large in comparison. Indeed, despite these apparently large export fluxes from the topmost box, the concentration of Nd at the surface is not zero (see Figure 8a-d). As an additional note, scavenging rates are not often presented as horizontal integrals (and these horizontally integrated values are difficult to compare with measured values at a given location), but we think it is still useful to present them in this way to illustrate subtle differences between particle types.

We have added a sentence to the caption of Fig. 5 that clarifies this point:
"Note that in (e)–(f), net scavenging rates in the top layer are positive and largest by construction because Nd can only be removed there, as opposed to all the layers below which can receive Nd from superjacent layers."

*L350-375, L391-394: i did not understand much of this, but a modeler might.*

These sections explain how we define the acceptable parameter values and apply a penalty if the model pushes a parameter into an unrealistic place. The optimization algorithm tries to minimize Equation 17, so if any parameters go to unrealistic values, the penalty grows and the algorithm will tend to push those values back toward realistic numbers. The first sentence of the paragraph is our attempt to say this in plain english before adding the technical details that are necessary for this type of model description paper.

We have tried to make this text more clear (L384–385): "If any parameter reaches a value close to the limits we impose in the model, this third term will grow large; since the algorithm tries to minimize Eq. (17), it will push that parameter back to a more acceptable value."

*L415: perfectly compensate each other? what does this mean?*

There are two parameters that control the scavenging strength: $K_x$, the equilibrium constant for scavenging, and $w_x$, the sinking speed. In the scavenging matrix, these terms are always multiplied together (i.e. they never appear alone). If we optimized both of them, it would be easy for the model to make one parameter large and the other one small, with the end result being that both of these terms compensate for each other in the final equation.

We have added some text to clarify this point (L452–455):
"(…) because $K_X$ and $w_x$ can perfectly compensate each other. For example, doubling $K_X$ while halving $w_X$ has no effect on Nd distributions and the objective function. Only their product, $K_X w_x$, which sets the strength of the "scavenging pump'" through the operator matrix

$\mathbf{T}_{scav}$, appears in the tracer equations (see Eq. (16) or e.g., John et al., 2020), such that these parameters cannot be easily optimized independently."

*Table 2: why are Criver and Cgw concentrations and not fluxes? The dust solubilities are indeed very high.*

The parameters $c_{river}$ and $c_{gw}$ are concentrations so that they give the Nd source when multiplied by known river and groundwater volumetric flow rates. The water discharge ($m^3 \, s^{-1}$) times the concentration of Nd ($mol \, m^{-3}$) in that water gives the source of Nd ($mol \, s^{-1}$) to the ocean (as in Eq. (11)). Note that this is an "effective" concentration rather than a concentration that you could measure in a river or aquifer because it only represents Nd that makes it into the ocean (and omits Nd that is sequestered in an estuary, for example). The resulting fluxes of Nd to the ocean are given in Table 3.

Regarding the dust solubilities, we acknowledge that these numbers are high and we aren't trying to say that the final optimized values are correct. The solubility parameter was allowed to span the full range (0-100%), although lower solubilities were favored. Future model users are welcome to tighten the range restrictions so the solubilities achieve more accurate values.

No changes to the manuscript in response to this comment.

*Table 3: this is an interesting table; the main sources to the oceans are dust and sediment. However, the subsequent section 3.3.1 needs to be far better referenced: first off by adding references to the model comparisons. Secondly, the authors should compare their flux estimates to observational flux estimates, and reference these papers.*

The references for the model comparisons are all listed in Table 1. The goal of this section is to provide some reassurance that our optimized model values are reasonable given previous modeling studies. We have also compiled some estimates of global sources based on measurements and added them to this section.

We have added more citations of Table 1 and estimates from field studies to this section in response to this comment:
- L520–521: "our aeolian sources are an order of magnitude larger than previous GCM-based modelling studies (0.69–3.5 Mmol $yr^{-1}$) (Table 1) and than the 4.4 Mmol $yr^{-1}$ estimate of Greaves et al. (1994)."
- L532–533: "At 32 Mmol $yr^{-1}$, the GNOM sedimentary source falls right within the 0–78 Mmol $yr^{-1}$ range of previous models (28–55 Mmol $yr^{-1}$ for GCM-based studies) (Table 1) and in agreement with the 18–110 Mmol $yr^{-1}$ range of Abbott et al. (2015b)."
- L534–536: "The third largest source is riverine, with about 10 Mmol $yr^{-1}$, also in accord with the published 1.8–12.4 Mmol $yr^{-1}$ range in previous models (Table 1) and similar to the 4.6–12 Mmol $yr^{-1}$ values from Goldstein and Jacobsen (1987) and Greaves et al. (1994)."
- L536–538: "The GNOM optimization did not favor a large source from submarine

groundwater discharge, which has been estimated between 29–81 Mmol yr$^{-1}$ by Johannesson and Burdige (2007), but this source was not included in any other modelling studies so it is difficult to compare with other global model estimates."

*L.492: I don't understand this section. Also, a molecule of Nd in the oceans is the same as any other; just to make clear that other (non-Nd) parameters are critical. For example, (L 507:) why should sediment sourced Nd be removed quicker from the water column? Unless there are significant nepheloid interactions, there are as many particles to scavenge 10 m off the seafloor as there are 1000m off the seafloor (in the model and largely in the ocean).*

We completely agree that a molecule of Nd in the ocean is the same regardless of its source, however the beauty of a model is that we can keep track of these molecules *as if* they were different. The reason sedimentary sourced Nd has a shorter residence time is that it enters the ocean near the bottom. Therefore, once it is scavenged it only needs to fall a few meters before it leaves the ocean domain. A molecule sourced from dust, for instance, that enters at the surface needs to fall 1000s of meters once it gets scavenged. Since particles sink at similar speeds, it takes that dust-sourced molecule longer to reach the sediments than the sedimentary-sourced molecule. This happens regardless of particle concentration.

We have expanded this paragraph to clarify this point (L556–561):
"Unlike in the real ocean, where each molecule of Nd is indistinguishable from the next (with no information about its initial source), in a model we can track Nd coming from different sources and calculate source-partitioned residence times. We find that sedimentary-sourced Nd has the shortest residence time at 370 yr. This is because it is injected just above the seafloor and is thus buried more quickly (i.e., a molecule of Nd sourced from the sediments at 5000 m, which only has to fall a few meters to be scavenged back to the sediments, leaves the ocean quicker than a molecule near the surface sourced from dust, which has to fall 1000s of meters)."

*Section 3.3.4: i understand the goal for this section, and it seems like a good idea. But isn't the entire thing borne out in eNd in the model? Can't this section be shortened?*

We believe that it is important to keep the part about Nd as well as eNd in this section. While it is generally accepted that Nd is not conservative (even though eNd may be conservative or quasi-conservative), this has not been thoroughly investigated in a quantitative sense. We believe Figure 13 is actually quite informative in giving a quantitative picture of how much Nd is transported by the circulation versus added to the system. This also sets us up to trace Nd isotopes.

We have changed the title of this section to "Tracking Nd to investigate $\varepsilon_{Nd}$ conservativeness" so it is clearer what we are trying to explore in this section (L564).

*L688: "x x x"?*

We thank Referee #3 for spotting this omission.

In response, we have added the missing root-mean-square value for the simple Si-cycling model and indicate it in Fig. A1. We have also added the root-mean-square values for [Nd] and $\varepsilon_{Nd}$ in Section 3.1 and Fig. 7.

*General comments:*

*The model output does capture a lot of the observed data, but isn't the model trained to do this? through use of 43 free variables? A modeler should offer a critical evaluation of the model skill.*

Yes, the model is trained to get the right answer, and it is always possible to get the right answer for the wrong reason (i.e., the wrong parameter values). As we said before, the goal with this model description paper is to include all (or as many as possible) of the sources and processes that we think are important for Nd cycling in the ocean. In future work, we plan to explore whether we can get similarly good matches between the model and observational data by eliminating some of the free parameters or constraining them more tightly. This is also something that a future GNOM user could do.

We have modified a sentence in the conclusion to emphasize this point (L675–676): "Parameter values and acceptable ranges can be tuned, entire mechanisms can be turned off, eliminating free parameters, or added with a few changes of simple lines of code."

*I appreciate the attempts to define input/output functions, but it would be interesting to see how close their approximations are to experimental and geochemical modeling observations of these processes.*

As the Referee #3 clearly understands, the goal of this paper is to describe a model of the marine Nd cycle and provide some initial estimates of optimized parameter values. We plan to improve upon this model in the future, including more detailed comparisons between modeled sources/processes and real-world measurements. This detailed investigation is outside the scope of this paper, however. We have tried to add some additional comparisons between our estimates and measurements (as well as previous models), however this comparison can be difficult given the global nature of some sources, i.e., discrete measurements need to be extrapolated in order to compare with a global model.

---

## Author Response (AR2)

**Referee #1**

Below are Referee #1 comments reproduced in grey italic and our point-by-point replies in black. Line numbers in our response refer to the revised manuscript with changes accepted.

*I confirm that Pasquier's modelling approach is very interesting, efficient and full of promises. The authors answered to my comments regarding the lack of references, and clarified some of the issues I raised. However, I still consider that there are still unclear sections.*

We would like to thank Referee #1 for their positive overall opinion of our work. We hope that our responses to the comments below help to clarify the issues raised.

The goal of the manuscript is still unclear, mostly blurred by the conservativity test in the end, as well as a poor discussion of the hypothesis vs simulated parameters. The authors should clearly state in the abstract that the goal of their study is to present the model and not to establish parameter values. This is still not clear, and the last section (application to test the conservativity) is more confusing than clarifying.

The main goal of this manuscript is to describe the Global Neodymium Ocean Model (GNOM) v1.0, and highlight some of the unique attributes of this model compared with previous models of the modern ocean Nd cycle. Thus, this paper fits the GMD "Model Description" type paper.

In the abstract of the manuscript, we state that: "... in this model description paper we [are] present[ing] and describ[ing] the Global Neodymium Ocean Model (GNOM) v1.0...". We also state that our "...systematic objective optimization allow[s] us to make **preliminary** estimates of biogeochemical parameters." The results we present are intended to highlight the potential of this new model, but we are in agreement that some parameter values from our preliminary estimation are unrealistic. We plan to make further refinements in the future. Our hope is that by publishing the *model description* paper separately from a paper focusing on the *scientific implications* of finalized parameter values, other members of the community can, in the interim, also use this new tool and make further improvements as well.

As part of the Model Description type paper for GMD, there is a requirement that "[t]he model description should be contextualised appropriately. For example, the inclusion of discussion of the scope of applicability and limitations of the approach adopted is expected." In light of this, we feel that it is crucial to include the section on unique diagnostics that are possible with the GNOM model, such as the test of conservativeness that we present in Section 3.3. These diagnostics, which are made possible by the formulation of the model, are also mentioned in the abstract.

**We have edited the abstract in an attempt to further clarify our goals in this paper.**

My understanding of the hypotheses is still very confused. I would appreciate if the authors would make the effort to improve their model description and tests. For example, they are going to tinker with sediment flux parameters (let's say in Greenland) but actually, they don't optimize the sources but only the parameters associated with the sources. Particle fall velocities are considered as constant while Dutay et al showed that this parameter can strongly influence the simulation results. On how much time the velocity fields are averaged is not given, and if the AWESOME OCIM particle fields are consistent with the same dynamics is not given either. We are somewhat confused by this comment. First, it is unclear which part of the model description Referee #1 believes is missing.

The comment on optimizing the parameters rather than the sources themselves is also confusing. If Referee #1 is suggesting that each pixel of the model should be its own parameter for the sediment source, then that would require the optimization of an additional 10,441 parameters, which is an unreasonable request. An important trait of the GNOM is that its source and sink parameterizations are based on simple mechanistic representations. For instance, the sedimentary-source parameterization of the GNOM can be boiled down to three reasonable assumptions: it varies with depth, composition (via the  $\varepsilon_{Nd}$  enhancement), and in some cases with location (Greenland enhancement).

Regarding particle sinking velocities, we are unsure which Dutay et al. study Referee #1 is referring to, but we do not dispute that particle sinking velocities can vary. As we describe on L455, our assumption of constant settling velocity (and having this not be an optimizable parameter) is a simplification. Due to the parameterization of reversible exchange itself, there is no point in independently optimizing both the settling velocity and the scavenging equilibrium constant:

"...we do not optimize the corresponding settling velocities ... because  $K_x$  and  $w_x$  can perfectly compensate each other. For example, doubling  $K_x$  while halving  $w_x$  has no effect on Nd distributions and the objective function. Only their product,  $K_x w_x$ , which sets the strength of the "scavenging pump" through the operator matrix  $\mathbf{T}_{scav}$ , appears in the tracer equations (see Eq. (16) or, e.g., John et al., 2020), such that these parameters cannot be easily optimized independently."

Finally, we note that in a steady-state model such as the GNOM described here, variables and parameters are not averaged over specific periods of time (steady-state = no variation in time) although one could think of them as representing the climatological mean.

**No changes to the manuscript were made in response to this comment.**

Regarding the conservativity test. In this section, the authors declare that they will model Nd parameters while this part presents the modeling of a new tracer in itself (certainly related to Nd). Actually, a very fast passive tracer modeling tool allowing to do a lot of sensitivity tests is presented, in which they have implemented Nd and its sources. Again, it's a nice tool, but this part leaves the reader with the same feeling as for the inverse modeling above, with this "catalog aspect" where they present a rather sloppy (or at least not finished) application saying that they will do it seriously later.

As reiterated multiple times in our reply to the first reviews, our manuscript is a "model **description** paper", which aims to **describe** the GNOM framework. We respectfully reject Referee #1's claim that showcasing an advanced (and efficient) Green-function diagnosis that is seldom easily obtainable from GCMs in our model is somehow "sloppy".

It seems that Referee #1 is expecting a full geochemical study such as would be required for publication in a Geosciences journal, rather than the description of a new model as is required by GMD for a "Model Description" paper (see response above).

**No changes to the manuscript were made in response to this comment.**

**Detailed comments**

Line 34: the CHUR value has bee actualized to 0.512630 (see Bouvier et al, EPSL, 2008)

In order for  $\varepsilon_{Nd}$  measurements to be consistent with those published throughout time, it is crucial that they are all referenced to the same value. Therefore, while it is true that the CHUR value has been updated since the work of Jacobsen and Wasserburg (1980), it is preferable for us to use the Jacobsen and Wasserburg (1980) value of 0.512638 for consistency with observational data.

We also note that the Jacobsen and Wasserburg (1980) CHUR value differs from that of Bouvier et al. (2008) only in the sixth significant digit, which is within their reported uncertainty of  $\pm 0.000011$ , and that swapping one for the other results in a mere 0.15  $\frac{1}{2000}$  change in  $\varepsilon_{Nd}$ , which is within error of most measurements (and within the  $\pm 0.2$   $\frac{1}{2000}$  uncertainty as propagated from the Bouvier et al. (2008) uncertainty).

We have updated L34 to read: "For consistency with previously published data, we use  $R_{CHUR} = 0.512638$  from Jacobsen and Wasserburg (1980), rather than the updated value from Bouvier et al. (2008)."

*Line 44: the interpretation "powerful palaeoceanographic tracer" is challenged by all the studies trying to tackle the "missing parameter". This should also be mentioned here (for reasons of fairness and rigor).*

Referee #1 ignored the adjective "potentially" in their quote of our manuscript, which reads: "**potentially** powerful paleoceanographic tracer". We strongly disagree with Referee #1's views here. There have been clear paleoceanographic advances based on Nd, as demonstrated by the plethora of scientific discoveries in the literature. Every paleo proxy has complications and limits. In fact, in our view, although we did not do so, it would have been legitimate to forgo the modifier "potentially" and simply call Nd isotopes a powerful paleoceanographic tracer.

No changes to the manuscript were made in response to this comment.

*Line 49: the Geotraces IDP 2021 is published yet. Although not used in this work, could be mentioned.*

The Geotraces IDP 2021 was published after our original submission and model runs. In future versions of the model, we will certainly include this updated dataset. We would like to note that we took extreme care to include all other *published*  $\varepsilon_{Nd}$  data that came out after the IDP 2017 but before our manuscript submission.

No changes to the manuscript were made in response to this comment.

*Line 78: to my knowledge, only Gu et al (2019,2020) attempted to mathematically optimize a Nd-Cycling. Rempfer as Arsouze (or others) only made artisanal tests. The authors could moderate this sentence.*

Re-examining the text and supplemental material of Gu et al. (2019), we see that they performed a very similar optimization to Rempfer et al. (2011) (two parameters were varied in each study).

**We have updated L78 to read:**

"To our knowledge, only Rempfer et al. (2011) and Gu et al. (2019) have attempted to optimize a Nd-cycling model, using the low-resolution Bern3D OGCM and 3° resolution Community Earth System Model (CESM1.3), respectively, and each optimizing only two parameters."

Line 117: I disagree with the following sentence, for 2 reasons:

"These preliminary diagnostics already reveal important information. They help quantify the conservativeness of epsNd along water pathways and unveil the underlying mechanisms by evaluating the effect of local sources and sinks."

The first reason is that it is contradictory with the initial goal which is (e.g. line 530) "we emphasize that it is not the goal of this manuscript to establish estimates of the GNOM parameters and that we welcome future GNOM users to apply narrower ranges for those parameters for which they have better constraints (for example, restricting dust Nd solubilities to values below 10 %)"

The whole manuscript would benefit if the goals are better and clearer written (as I suggest, starting in the abstract). The authors should not declare that they "unveil" source parameters having dust solubilities of 80% ...which is so far from the observations!

The second reason is that conservativity is not tested with exactly the same model (see above). Actually, I'd strongly suggest to remove the conservativity section (3.3.4) which will have many advantages 1) make the whole work clearer and explicitly dedicated to the presentation of GNOM; 2) inverse model are developed to constrain the sources that explain the distribution of any parameter in a given reservoir, which is discussed in the other sections (no problem with that) but testing conservativity would gain with a Lagrangian modelling, and not with a wobbly inversion and 3) this will shorten the manuscript, which is very long.

*Thus, I'd replace the sentence line 117 by another one underlining again that the goal of this work is not to establish the GNOM parameters..."*

Referee #1 seems confused about a couple of points and misrepresents some of our claims.

Firstly, we want to make clear that the diagnostics are tested in exactly the same model as is used for all the work presented in this manuscript. See L10 of the abstract.

**We have edited L580 of the main text to reiterate that the diagnostics are implemented within the GNOM model.**

We feel the line quoted about our preliminary diagnostics is already mild in tone, but perhaps the reviewer is suggesting that our model diagnostics cannot reveal *any* information about the system if the optimized parameters are not finalized. We disagree. Again, the main point of this paper is to highlight the capabilities of our new model and the diagnostics are an important part of the model. Indeed, we *specifically do not* include much interpretation of the diagnostics we present, as that would require more finalized parameter values. In future science-focused papers, with finalized parameter values, we plan to discuss the scientific implications of the parameter values themselves and the Green-function-based diagnostics.

**We have tried to make this point clearer on L119:**

"Detailed investigations of these diagnostics are out of the scope of this study and will be carried out in future work using a subsequent version of the GNOM with more finalized parameter values."

Again, our manuscript is quoted here without context. Nowhere do we claim that we "unveil source parameters". We merely say that the partitions allowed by our diagnostics "…help quantify the conservativeness of  $\varepsilon_{Nd}$  along water pathways and **unveil the underlying mechanisms** by evaluating the effect of local sources and sinks".

**We have edited this sentence (L118) to now read:**

"...help quantify the conservativeness of ɛNd along water pathways and unveil underlying mechanisms by evaluating the effects of local sources and sinks".

All of the co-authors agree that the suggestion by Referee #1 to remove Section 3.3.4 is unreasonable, given that it describes a key feature of the GNOM. Running such diagnostics in more traditional GCMs is not a trivial task, but here, owing to the matrix and steady-state representation, these diagnostics are easily set up and take roughly 10 seconds to run on a modern laptop.

Referee #1 is certainly entitled to their opinion that a Lagrangian model would be suited to such a study, but here again, we disagree. It would actually be quite difficult for Lagrangian models (i.e., particle-tracking models) to investigate the question of  $\varepsilon_{Nd}$  conservativeness because they would not capture the effects of eddy-diffusive mixing, which is *very important* for quantifying conservativeness. Dissolved tracers can diffuse and mix in a way that particle-tracking Lagrangian models can't represent directly.

Referee #1 also fails to characterize what makes our inversion "wobbly" and shortening a manuscript for the sake of shortening a manuscript seems ill-guided.

No changes have been made in response to this portion of the comment.

Line 314: Arsouze and others are making the same first order assumption

Agreed.

In response, we have added a citation of Arsouze et al. (2009): "We follow, e.g., Bacon and Anderson (1982), Siddall et al. (2008), and Arsouze et al. (2009), and assume that dissolved and scavenged Nd are exchanged via a first-order kinetic reaction"

*Line 350: Lagarde et al recently published a well documented section of particulate REE (in the North Atlantic) that could be considered here too (Biogeosciences, 2020)*

It seems like the Lagarde et al. paper mostly focuses on scavenging onto biological particles (e.g., biogenic silica), but they do discuss scavenging onto Fe-Mn (hydr)oxides. We have therefore added this citation to the sentence mentioning Fe-Mn (hydr)oxides, since our model already incorporates biogenic particles such as silica, which are known to scavenge Nd.

We have added this citation to L353.

*Line 414: I would suggest to precise that the authors are tackling the dissolved Nd parameters (in the title for example)*

OK. We have edited this section title to specify that we are focused on dissolved Nd.

"2.4.2 Dissolved neodymium and ENd data"

**Referee #2**

Below are Referee #2 comments reproduced in grey italic and our point-by-point replies in black. Line numbers in our response refer to the revised manuscript with changes accepted.

**Dear Editor, dear authors,**

After reading the authors' reply I believe my disagreement with the authors is about the model philosophy, and what does this paper want to achieve.

In the initial draft, the manuscript seems not only a model description paper, but at the same time, the authors' affirmative languages make it look like a thorough study of the modern ocean Nd cycle. As stated in my previous comments, I have little criticism on this manuscript as a model description paper, which I welcome STRONGLY (except for some problems with the optimization approach). My criticism is mainly about this manuscript as a "proper" geochemical study of the ocean Nd cycle. The authors insisted in the reply that this manuscript is mainly a model description paper, and the results are "test examples" that will be redone with better care later, and thus largely deferred my criticism on the Nd cycle to future studies. If so, they should make is clear that the results should NOT be considered "final" as they themselves believe so. My worry is that, these test results will confuse readers and lead to incorrect citation without proper warning.

The authors are trying to purse a "truly global" parameterization of every aspect of the Nd cycle. This is highly commendable and ambitious, and it is also the ultimate goal of all scientists studying Nd. But my opinion remains that this is NOT achievable in practice at the moment, because of sever data limitation and poor mechanistic understanding. My model philosophy is thus conservative: the modeler should follow measured data whenever they are available even if imperfect, rather than use sweeping global generalizations and optimizations that appear superior but has no mechanistic foundation. This is clearly my bias as a data scientist which I acknowledge fully.

We would like to thank the reviewer for their comments and their favorable view of our paper as a model description paper.

In our latest revision of the manuscript, we have included more language throughout that clarifies the manuscript goals (to describe a new model of the marine Nd cycle) and reiterates that we are not presenting final parameter values (which we hope to do in a future science-focused paper).

We also appreciate the reviewer's perspective about implementing global models of the Nd cycle despite yet incomplete data coverage. We feel that it is still a worthwhile endeavor at this point, but we also hope that this model will only become more useful and accurate in the future as additional data are published.

**Referee #3 (Brian Haley)**

Below are Referee #3 comments reproduced in grey italic and our point-by-point replies in black. Line numbers in our response refer to the revised manuscript with changes accepted.

*I* appreciate the efforts taken by the authors in response to the (many) comments made in review. I can fully endorse publication of this work at this time.

(However, I just want to stress to the authors that (1) the comments on the Sm/Nd ratio difference between ALL rocks is as critical as how old they are for eNd. Yes, basalts are \*really\* different, and this their eNd is so different. But the age of a rock is only a part of the causality of a given eNd. Also (2) hydrothermalism is a sink, and does not add to nor modify water column eNd (also see work by Chavagnac) - at least not directly. But then i dont know how your model deals with the precipitate once it forms...)

We thank Referee #3 for their detailed initial review, which substantially improved the manuscript. In future iterations of this model, we hope to include a more detailed (and accurate) description of neodymium cycling at hydrothermal vents.

The Acknowledgements section now includes this sentence: "The authors would like to thank the editor, Brian Haley, and two other anonymous reviewers for their helpful comments, which substantially improved this paper."

---

## Author Response (AR3)

Dear Dr. Yool,

We are very pleased to hear that our paper will be accepted with corrections—thank you for all the assistance throughout this process. We have re-formatted the Code and Data Availability section of the manuscript, so it is easier to follow (including a bulleted list of the datasets that are programmatically downloaded with the GNOM code). We have also re-worded the Acknowledgements section as suggested.

Sophia Hines, on behalf of all authors